# Circulating miRNAs as Diagnostic and Prognostic Biomarkers in Common Solid Tumors: Focus on Lung, Breast, Prostate Cancers, and Osteosarcoma

**DOI:** 10.3390/jcm8101661

**Published:** 2019-10-11

**Authors:** Michela Bottani, Giuseppe Banfi, Giovanni Lombardi

**Affiliations:** 1IRCCS Istituto Ortopedico Galeazzi, Laboratory of Experimental Biochemistry and Molecular Biology, Via Riccardo Galeazzi 4, 20161 Milano, Italy; michela.bottani@grupposandonato.it (M.B.); banfi.giuseppe@fondazionesanraffaele.it (G.B.); 2Vita-Salute San Raffaele University, 20132 Milano, Italy; 3Dept. of Physiology and Pharmacology, Gdańsk University of Physical Education and Sport, Gdańsk, ul. Kazimierza Górskiego 1, 80-336 Pomorskie, Poland

**Keywords:** biomarkers, circulating miRNAs, miRNA signature, extraanalytical variability, sensitivity and specificity, lung cancer, prostate cancer, breast cancer, osteosarcoma

## Abstract

An early cancer diagnosis is essential to treat and manage patients, but it is difficult to achieve this goal due to the still too low specificity and sensitivity of classical methods (imaging, actual biomarkers), together with the high invasiveness of tissue biopsies. The discovery of novel, reliable, and easily collectable cancer markers is a topic of interest, with human biofluids, especially blood, as important sources of minimal invasive biomarkers such as circulating microRNAs (miRNAs), the most promising. MiRNAs are small non-coding RNAs and known epigenetic modulators of gene expression, with specific roles in cancer development/progression, which are next to be implemented in the clinical routine as biomarkers for early diagnosis and the efficient monitoring of tumor progression and treatment response. Unfortunately, several issues regarding their validation process are still to be resolved. In this review, updated findings specifically focused on the clinical relevance of circulating miRNAs as prognostic and diagnostic biomarkers for the most prevalent cancer types (breast, lung, and prostate cancers in adults, and osteosarcoma in children) are described. In addition, deep analysis of pre-analytical, analytical, and post-analytical issues still affecting the circulation of miRNAs’ validation process and routine implementation is included.

## 1. Introduction

Cancer represents a significant global cause of death, so early diagnosis with a constant checkup of disease progression and patient response to chemotherapy and surgery are essential to efficiently manage subjects with cancer. Up to now, tissue biopsies have been considered one of the most valuable methods for cancer diagnosis, but they are an invasive procedure. To overcome these limits, great efforts have been made in the past few years to find novel, minimally invasive, fast, reproducible, and low-cost techniques for cancer management, such as circulating biomarkers in human biofluids (i.e., blood and derived plasma or serum). Currently, circulating microRNAs (miRNAs) are promising markers for cancer diagnosis, prognosis, for monitoring the treatment response, and as powerful tools for personalized approaches.

### 1.1. MiRNA Biogenesis and Biological Functions

MiRNAs, a category of non-coding small RNA (~22 nucleotides) [1,2], act as modulators of gene expression in human cells and tissue, and their potential as biomarkers for human pathologies has been highlighted by their detection in 12 types of human fluid [3,4,5]. In addition, miRNA expression has been characterized in 61 types of tissue [6]. MiRNA genes can be sited within both protein-coding or non-coding gene sequences in all human chromosomes, and their biogenesis includes multiple stages [7,8,9,10,11]. First, in the nucleus, miRNAs are transcribed and processed into a hairpin precursor (pre-miRNA), by the DROSHA–DGCR8 complex, which is translocated into the cytoplasm and cleaved by DICER obtaining, a double-stranded miRNA. Both strands are, or only one strand is, selected for incorporation into the RNA-induced silencing complex (RISC) to act as mature miRNAs and, thus, to bind the “seed” region of the target mRNA. MiRNAs are involved in a wide range of regulatory functions in both biological and pathological processes: cell/tissue development, differentiation, metabolism, homeostasis, proliferation and apoptosis, immune responses, and cancer development, progression, and response to therapy [12]. MiRNA biogenesis and functions can be affected by: (i) epigenetic modifications, (ii) alteration in protein activity involved in this process [13,14,15,16], (iii) single nucleotide polymorphisms (SNPs) in miRNA genes [17,18], (iv) hereditary mutations in the miRNA seed region [19,20,21]; (v) somatic mutation in miRNA genes [22,23], and (vi) somatic copy-number variation of miRNA genes [24].

Information about all the discovered miRNAs can be found in databases such as miRBase (www.mirbase.org) [25] and MirGeneDB (http://mirgenedb.org) [26].

### 1.2. Circulating miRNAs as Biomarkers

In cancer, a biomarker can give diagnostic, prognostic, or predictive information about the considered pathological condition, and it must be highly specific, sensitive, and easily measurable. In addition, a biomarker must be validated by clinical studies and should be useful for patient management [27]. Circulating miRNAs show both strengths and weaknesses as biomarkers in each stage of their validation process (preanalytical, analytical, and postanalytical). Considering the preanalytical phase, miRNAs are detectable in human biofluids, collected with a minimally invasive procedure, and their transport into extracellular vesicles or binding to proteins makes them stable in several conditions (extreme pH values, repetitive freeze and thaw, up to 24 h at room temperature, or for decades at <−70 °C) [4,5,28,29,30,31,32,33]. On the other hand, this stage of validation can be affected by several biases concerning patients’ daily habits (smoking, physical activity, kidney pathologies, diet, and circadian rhythm) [34,35,36,37,38], sample collection (type of tubes and anticoagulant additive as well as the phlebotomy technique), and handling (sample centrifugation force and time) [39,40,41,42]. For instance, our newest results demonstrated that plasma from ethylendiaminotetraacetate (EDTA) salts-anticoagulated tubes with a gel separator resulted in higher detectability and stability of a wide panel of miRNA than standard EDTA plasma and platelet-poor EDTA plasma [43]. In the analytical phase, reliable methods for miRNA measurement are available, with real-time quantitative polymerase chain reaction (RT-qPCR) considered as the gold-standard technique, but significant interplatforms (PCR-based, microarray, and next-generation sequencing (NGS)) differences have been highlighted [44,45]. In this phase, circulating miRNA validation can be affected by non-circulating miRNA contamination (skin and blood cells, and activated platelets) and by hemolysis [28,33,39,42,46,47,48,49]. Finally, issues regarding the postanalytical phase mainly concern the absence of a standardized data-normalization process (single or multiple exogenous or endogenous genes, global mean of all evaluated miRNAs) [32,50,51]. Therefore, the use of circulating miRNAs as biomarkers in patient management requires standardized protocols and procedures for each phase of their validation [28,42,50].

### 1.3. Aim

Based on the potentialities of miRNAs as biomarkers, enormous efforts have been made in studying and in defining the existing relationships between miRNA-altered expression and human pathological conditions, particularly in oncology, as demonstrated by the huge number of articles published in the last few years on this topic. Due to this growing interest, the aim of this review was to comprehensively collect available data (published until the end of 2018) about current and potential next uses of circulating miRNAs as biomarkers in the diagnosis and prognosis of a subset of solid tumors. Specifically, we focused our attention on high-incidence tumors and on those tumors in which miRNA-based diagnostics has gained the most promising results. In particular: lung cancer, due to high mortality in both genders and its increasing incidence in women; prostate and breast cancers, due to their high incidence in males and females, respectively; and osteosarcoma, since it represents one of the most common solid tumors in children. Information was obtained from each paper claiming diagnostic, prognostic, and/or predictive potential of circulating miRNAs, also including any available information about pre-analytical precautions adopted, quantification platforms used, and normalization strategy applied. Where available, we also collected information about the sensitivity and specificity of specific circulating miRNA signatures to evaluate their clinical potential as biomarkers to assess the presence of a disease and, at the same time, the absence of the same pathology in healthy individuals. In addition, the plot of specificity vs. sensitivity, namely, the receiver-operating-characteristic (ROC) curve, and the respective area under the curve (AUC), which represents the degree of separability and is an effective accuracy measure, were considered [52]. An AUC of 1 refers to perfect separability, whereas a value of 0.5 shows no capacity to discriminate the datasets. Articles with methodological deficiencies were excluded from the review.

Besides the described distinction between free (mainly protein-associated) and exosome-/microvescicle-/LDL-associated circulating miRNAs and their distinct functions, from biomarker-like (free circulating miRNAs) vs. hormone-like (encapsulated circulating miRNAs) [53], based on the aim of this review, only free-circulating miRNAs were considered.

## 2. MiRNAs as Tumor Biomarkers

Aberrant expression of circulating miRNAs as diagnostic and prognostic biomarkers in several types of cancer is a topic of interest. Altered levels of circulating miRNAs have been associated with tumor growth, progression, metastasis, and drug resistance, suggesting their potential use as a tool to optimize patients’ therapies. As a general rule, oncogene miRNAs are upregulated in tumor samples, and consequently in patients’ blood, whereas tumor-suppressor miRNAs are downregulated.

### 2.1. Circulating miRNAs and Lung Cancer (LC)

Worldwide, lung cancer (LC) is a leading cause of cancer mortality in both males and females. LC can be ascribed at two main types: small-cell lung cancer (SCLC) accounting for 15% of cases, and 85% are represented by non-small-cell lung cancer (NSCLC). NSCLC is characterized by two predominant subtypes: adenocarcinoma (ADC) and squamous cell carcinoma (SCC). To date, methods for early LC diagnosis include chest X-rays, low density computed tomography (LDCT), and other imaging techniques. In addition, several potential circulating biomarkers have been identified: the carcinoembryonic antigen (CEA), cytokeratin-19 fragment (CYFRA21-1), neuron-specific enolase (NSE), cancer-associated antigens CA125 and CA19-9, and chromogranin A, together with molecular signatures as mutations in *KRAS* and *TP53* genes. However, these methods have shown limited specificity, sensitivity, and reproducibility for LC detection.

#### 2.1.1. Circulating miRNAs as Biomarkers for LC Diagnosis and Prognosis

One of the first papers aimed at discovering novel circulating miRNAs as biomarkers for LC, starting from a screening of 880 mature miRNAs, identified miR-1254 and miR-574-5p as potential serum biomarkers for early-stage NSCLC due to their upregulation in NSCLC compared to healthy subjects. Their combination gave an AUC of 0.75 with 73% sensibility and 71% specificity for early-stage NSCLC [54]. MiR-21, previously reported as an oncogene in several human cancers, was identified by microarray analysis out of 427 miRNAs as the most upregulated miRNA in serum from NSCLC patients compared to healthy volunteers. Its levels are associated with worse prognosis and adverse clinicopathological features (grade and metastasis) [55]. MiR-21 was found in NSCLC tumor samples, other than in association with the tissue upregulation of miR-141 and miR-200c [56], and the downregulation of miR-486 in both tissue and plasma [57]. MiR-21 tissue expression is correlated with tumor size, while its circulating levels are correlated with stage and lymph-node metastasis. ADC is associated with higher miR-21 and lower miR-200c tissue expression than SCC. Moreover, tissue expression levels of miR-21, miR-141, and miR-200c as well as serum miR-21 levels, are inversely correlated with overall survival (OS) [56]. In addition, plasma miR-21 increase and miR-486 decrease gave an AUC of 0.740 and 0.857, respectively, in LC that increased to 0.901 if the miR-21-to-miR-486 ratio was considered [57]. MiR-21 upregulation in NSCLC samples is also correlated with a poor response to platinum-based chemotherapy after tumor resection [58].

Three members of the miR-183 family, miR-183, miR-182, and miR-96, could play crucial roles in NSCLC development. They are downregulated in both NSCLC tissue and serum, although only miR-96 tissue expression is correlated with serum levels. Tissue overexpression of miR-183 is correlated with lymph-node metastasis, lung-membrane invasion, and advanced clinical stages, while tissue and serum upregulation of miR-182 is strongly associated with lung-membrane invasion and >3 cm tumor size. In addition, miR-96 serum levels and miR-183, miR-182, and miR-96 expression in cancer tissue are higher in SCC than in ADC and are inversely correlated with OS, thus indicating prognostic potential [59]. Two pairs of serum miRNA signatures (miR-15b/miR-27b and miR-15a/miR-27b) were identified as potentially discriminating NSCLC patients from healthy subjects. ROC analysis revealed an AUC of 0.98 with 100% sensitivity and 84%–100% specificity for miR-15b/miR-27b, 87%–94% sensitivity, and 75%–93% specificity for miR-15a/miR-27b [60]. Low serum levels of miR-625* and miR-361-3p, identified from 1158 screened miRNAs, discriminated NSCLC patients from both benign lung disease and healthy individuals (AUC: 0.86 for miR-361-3p and 0.77 for miR-625*). Moreover, miR-625* levels were significantly lower in SCC and smoking patients than in ADC and non-smoking patients. After tumor-ablation serum, miR-625* and miR-361-3p levels were significantly raised to values comparable to those of benign lung disease patients or healthy controls [61]. The downregulation of plasma miR-204 is also associated with NSCLC patients, with accuracy higher than commonly used CEA and CA19-9 markers (AUC of 0.81, 0.72, and 0.69, respectively), and is correlated with tumor stage, distant metastasis, and shorter survival [62]. NGS strategy revealed that miR-181b-5p and miR-21-5p are upregulated in serum from SCC patients compared to healthy controls, while miR-103a-3p and miR-21-5p are upregulated, and miR-486-5p downregulated in ADC. MiR-486-5p and miR-181b-5p were further validated in tumor tissue compared to adjacent normal tissue [63]. By contrast, increased levels of miR-486, together with miR-150, were found in NSCLC plasma samples compare to healthy controls [64]. Two other validated microarray analyses identified plasma miR-30a and serum miR-1244 as independent diagnostic markers for discriminating NSCLC from both benign lung lesion and healthy controls. MiR-30a has AUC, sensitivity, and specificity values of 0.727, 61%, and 84% for NSCLC, and 0.727, 55%, and 94% for benign lesions [65]. MiR-1244 discriminates all-stage and early-stage NSCLC patients both from healthy controls (AUC, specificity, sensitivity: 0.832, 82%, 80%, and 0.80, 54%, 100%, respectively) and from benign pulmonary nodules (0.861, 78%, 92%, and 0.848, 73%, 92%) [66].

Many papers aimed at validating miRNAs, previously associated with LC tissue samples or found dysregulated in other tumor types, as circulating biomarkers for LC. Starting from a microarray screening that identified 12 miRNAs aberrantly expressed in Stage I NSCLC primary tumor tissue, the potential diagnostic and prognostic ability of these miRNAs for NSCLC was also validated in plasma samples. Results showed that five of the 12 miRNAs are differently regulated in cases compared to healthy subjects (miR-210, miR-182, and miR-21 are upregulated, while miR-486-5p and miR-126 are downregulated), and are able to discriminate NSCLC with the following AUC, sensitivity, and specificity: 0.88, 85%, 69% for miR-486-5p, 0.75, 74%, 69% for miR-210, 0.66, 52%, 76% for miR-182, 0.76, 69%, 83% for miR-126, and 0.82, 79%, 66% for miR-21. In addition, the combination of miR-486-5p + miR-210 + miR-126 + miR-21 displayed higher diagnostic potential than single miRNAs (AUC, sensitivity, specificity: 0.93, 86%, 97%) [67]. The same authors later demonstrated that a three-plasma miRNA signature (miR-210 and miR-21 upregulation, and miR-486-5p downregulation) discriminated computed tomography (CT)-identified LC from benign solitary pulmonary nodules (SPNs) with an AUC of 0.855, 75%–76% sensitivity, and 85%–85% specificity [68]. Of the 15 LC-associated miRNAs selected from the literature (miR-221, miR-210, miR-205, miR-203, miR-199b, miR-197, miR-183, miR-182, miR-155, miR-128, miR-125b, miR-106a, miR-24, miR-21, and miR-17), after validation, only miR-197, miR-182, and miR-155 were significantly upregulated in LC patients, at all tumor stages, compared to healthy subjects. Single and combined miRNAs displayed elevated accuracy in identifying cases (AUC: 0.88 for miR-197, 0.71 for miR-182, 0.87 for miR-155, and 0.90 for miR-155 + miR-197 + miR-182). In addition, their levels were decreased by chemotherapy in responsive patients, while metastatic patients experienced higher plasma miR-197 and miR-155 than their non-metastatic counterparts [69]. Being aberrantly expressed in NSCLC tissue, let-7a was studied in blood. Its downregulation in the whole blood (WB) of NSCLC patients, compared to healthy subjects, was associated with an AUC of 0.95, with 90% specificity and 90% sensitivity [70]. MiR-21, miR-205, miR-30d, and miR-24 were upregulated in preoperative LC patients’ sera compared to healthy volunteers 10 days post-surgery, and only miR-21 returned to the healthy-control level. However, serum miR-21 and miR-24 significantly were decreased in the postsurgery phase. Presurgery ROC analysis in early- and advanced-stage LC revealed the following AUC, sensitivity, specificity values: 0.69, 46%, 92% and 0.70, 48%, 88% for miR-21, 0.78, 85%, 72% and 0.81, 100%, 64% for miR-205, 0.74, 76%, 80% and 0.76, 83%, 64% for miR-30d, 0.83, 76%, 64% and 0.86, 83%, 80% for miR-24. In addition, when combined with serum CEA concentrations, the AUC was 0.70 for miR-21, 0.66 for miR-205, 0.83 for miR-30d, and 0.75 for miR-24, which are all significantly higher than the AUC for CEA alone (0.56). Moreover, high preoperative serum miR-21 and miR-30d levels independently predict reduced OS [71]. A validation study defined AUC, sensitivity, and specificity of 0.81, 74%, and 72% for serum miR-21 in NSCLC, with a correlation with worse prognosis and negative clinicopathological features [72]. MiR-125b levels have been associated with NSCLC tumor stage (I–II, III, IV), discriminating them from healthy controls (AUC: 0.66, 0.84, 0.90; sensitivity: 96%, 93%, 95%; specificity: 38%, 66%, 67%) as well as early and advanced tumor stages (AUC, sensitivity, specificity: 0.78, 82%, 68%, for Stages I–II vs. Stage III; 0.67, 41%, 96% for Stage III vs. Stage IV). In addition, elevated serum miR-125b independently predicts negative prognosis [73]. High diagnostic accuracy in NSCLC has also been determined for miR-210 (AUC of 0.78), which is significantly increased in Stages III–IV than in Stages I–II, and decreased in patients responsive to cisplatin-based chemotherapy [74]. In WB, upregulated miR-339 and miR-328 gave the highest diagnostic accuracy for early and advanced NSCLC stage compared to healthy controls (AUC of 0.80 and 0.82, respectively), that is improved when combined with other miRNAs (AUC of 0.84 for miR-328 + miR-361 and 0.82 for miR-339 + miR-140) [75]. In two separate cohorts, the serum-expression levels of five miRNAs displayed elevated ranges of diagnostic accuracy (AUC) for NSCLC: 0.94–0.96 for miR-223, 0.92–0.94 for miR-155, 0.77–0.82 for miR-145, 0.77–0.79 for miR-21, and 0.89–0.91 for miR-20a. In addition, further subgroup analysis revealed that miR-155 has better diagnostic performance in non-smokers than in smokers, while miR-223, miR-145, miR-21, and miR-20a accuracy was higher in smokers, and all better differentiated NSCLC patients from healthy subjects in the advanced-stage and SCC subgroups [76]. Serum miR-145 has been identified in another NSCLC cohort (AUC 0.84), together with the upregulation of miR-146a and miR-125a-5p (AUC: 0.78 and 0.71, respectively) [77]. Compared to healthy controls, ADC is associated with serum miR-155 elevation, showing sensitivity (72%) higher than that of common biomarkers for LC: CA-125 or CEA (67% and 58%, respectively). By combining serum miR-155 and CA-125, sensitivity and specificity for ADC diagnosis were increased to 89% and 69%, respectively [78]. The downregulation of serum miR-499 accurately discriminates NSCLC patients from healthy subjects with an AUC of 0.91 [79]. Similarly, the downregulation of serum miR-152a and let-7c, and plasma miR-195 reveal elevated accuracy in distinguishing NSCLC patients from healthy controls (AUC, sensitivity, specificity: 0.85, 86%, 81%; 0.71, 72%, 78%; 0.89, 78%, 86%, respectively). MiR-152 and let-7c are associated with clinical staging and differentiation status [80], while miR-195 is associated with lymph-node metastasis, advanced clinical stages, and shorter OS [81]. Serum miR-494 and miR-411 (upregulated), and miR-770 and miR-98 (downregulated) are associated with NSCLC (AUC: 0.85, 0.84, 0.84, and 0.86, respectively) and are correlated with metastasis, clinical staging, differentiation, and worse prognosis [82,83,84,85]. MiR-141 has been suggested as useful in early NSCLC detection [86]. A valuable study identified serum miR-128-3p and miR-33a-5p, downregulated in LC, singularly or in combination, as accurately diagnosing all-stage and early-stage LC (AUC: 0.93 and 0.93 for miR-128-3p, 0.87 and 0.85 for miR-33a-5p, 0.95 and 0.96 for their combination) with better performances than the more common CYFR21-1, NSE, and CA-72-4 (AUC: 0.59, 0.62, and 0.52) [87].

The correlation of circulating miRNAs with LC patient prognosis has also been analyzed. Tissue and serum miR-19a and miR-17-5p are upregulated in LC, and their expression levels are inversely correlated with OS, and directly with stage lymph-node metastasis (miR-19a) [88,89]. Plasma miR-375, and serum miR-365 and miR-138, which are downregulated in NSCLC, are inversely correlated with distant or lymph-node metastasis and tumor grade (miR-365 and miR-138), and directly with OS [90,91,92]. The combination of low miR-146a and high miR-19b expression levels in both NSCLC tissue and serum, compared to their healthy counterparts, is associated with advanced tumor stage, lymph-node metastasis, worse response to chemotherapy, and OS [93]. In sera from NSCLC, miR-486-5p downregulation is associated with worse disease progression in a prognostic way [94]. More recently, a 7-miRNA ratio panel (miR-18a/328, miR-142-3p/342-3p, miR-186/342-3p, miR-18a/93, miR-19b/339-3p, miR-17/486-5p, and miR-151-3p/423-5p) and a 5-miRNA ratio panel (miR-24/142-3p, miR-27a/150, miR-27a/339-3p, miR-27a/532-3p, and miR-106b/532-3p) in NSCLC serum were strongly correlated with high-risk recurrence and worse survival (AUC of 0.859 for recurrence and 0.716 for shorter survival). The predictive accuracy of these two signatures was improved when combined with clinical features (AUC of 0.88 and 0.82, respectively) [95]. High-plasma miR-18a, miR-20a, miR-92a, miR-126, miR-210, and miR-19a are associated with reduced disease-free survival (DFS) and, similarly, high-plasma miR-18a, miR-20a, miR-92a, miR-210, and miR-126 are correlated with shorter OS [96]. Additionally, five serum miRNAs (miR-328, miR-191, miR-145, miR-28-3p, miR-18a) are significantly correlated with three-year OS in NSCLC patients [97].

In many cases, circulating miRNA panels were identified as accurate markers for LC diagnosis and prognosis. An 11-plasma miRNA signature discriminated between NSCLC patients and healthy subjects (AUC = 0.88), while a 6-plasma miRNA panel discriminates, in the same cohort, NSCLC patients from chronic obstructive pulmonary disease (COPD) patients (AUC = 0.94). AUC, sensitivity, and specificity of the 11-plasma miRNA panel values overcame those of common biomarkers (CYFRA 21-1 and tissue polypeptide-specific antigen). In addition, two different 3-plasma miRNA signatures are associated with increased ADC progression (high miR-155-5p and miR-223-3p, low miR-126-3p) and with reduced survival for SCC patients (high miR-20a-5p, low miR-152-3p and miR-199a-5p) [98]. Plasma miR-145 and miR-21 (upregulated), and miR-155 (downregulated) accurately discriminate early-stage LC from healthy smokers (AUC = 0.85–0.87, sensitivity: 69%–77%, specificity: 78%–81%) [99]. In ADC patients, a panel of six upregulated miRNAs (miR-939, miR-616*, miR-566, miR-550, miR-146b-3p, miR-30c-1*) and a signature of two downregulated miRNAs (miR-656, miR-339-5p) were identified as potential diagnostic serum markers for distinguishing patients from healthy subjects (AUC of 0.70 and 0.60, respectively) [100]. The combination of plasma miR-125a-5p, let-7e (downregulated), and CEA increased diagnostic accuracy (AUC: 0.66 for miR-125a-5p + CEA, 1.00 for let-7e + CEA, and 1.00 for miR-125a-5p + let-7e + CEA vs. 0.534 for CEA) for early-stage NSCLC vs. healthy controls [101]. A serum 6-miRNA panel combined with CEA increased discrimination accuracy (AUC) for all-stage and Stage I NSCLC patients from healthy subjects compared to CEA (0.58, 0.53): 0.71 for miR-29c + CEA, 0.71 for miR-429 + CEA, and 0.80 for miR-29c + miR-429 + CEA in all-stage NSCLC; 0.76 for miR-29c + CEA, 0.66 for miR-429 + CEA, and 0.83 for miR-29c + miR-429 + CEA in Stage I NSCLC [102]. Similarly, the panel composed of serum miR-210, miR-183, miR-182, miR-126, and CEA has higher diagnostic potential for all-stage and early-stage NSCLC (AUC, sensitivity, specificity: 0.97, 81%, 100%, and 0.98, 89%, 93%, respectively) than CEA alone [103]. The serum miR-652 + miR-660 combination, that is upregulated in NSCLC, more accurately discriminates patients from controls than Cyfra21-1 (AUC: 0.86–0.90 and 0.82–0.83, respectively), and even more when miR-660, miR-652, and Cyfra21-1 are used together (AUC: 0.94–0.95) [104]. In addition, combining serum levels of Cyfra21-1 and either miR-21 or miR-486 and miR-210, increases diagnostic potential in early-stage NSCLC (AUC: 0.91 for Cyfra21-1 + miR-21, 0.92 for Cyfra21-1 + miR-486 + miR-210) [105,106]. A 4-serum miRNA signature, composed by upregulated miR-301, miR-200b, miR-193b, and miR-141 has elevated accuracy in discriminating all-stage and Stage I NSCLC patients from healthy controls with AUC of 0.99–0.99 and 0.99–0.99, respectively [107]. Another validated panel of serum miRNAs for early-stage NSCLC consists of miR-126, miR-125a-5p, and miR-25 [108]. Their combination has diagnostic accuracy in the range of 0.93–0.94. MiRNAs belonging to the miR-148/152 family (miR-148a, miR-148b, and miR-152) exert a role in NSCLC: their expression is lower in NSCLC than in benign pulmonary diseases and is associated with lymphatic metastasis, tumor stage, and size. When combined, their diagnostic potential overcomes CEA (AUC, sensitivity, specificity: 0.79, 72%, 90% vs. 0.51, 44%, 89%) [109]. Even higher accuracy is gained by combining serum miR-148-152 family members and miR-21, as determined in NSCLC patients compared to healthy controls (AUC, sensitivity, specificity: 0.98, 96%, 91%) [110]. In NSCLC, miR-3662 and miR-499 are upregulated and their matched serum levels (or those of their precursors pri-miR-3662 and pri-miR-499) effectively discriminate patients from healthy controls (AUC, sensitivity, specificity: 0.91, 82%, 92% for miRNAs, 0.90, 76%, 82% for pri-miRNAs) [111,112].

In some cases, diagnostic panels are composed of miRNAs and other non-coding RNA molecules. For example, the miR-1254 + miR-574-5p + miR-485-5p + MALAT1 serum panel accurately discriminates NSCLC patients from healthy subjects (AUC = 0.84–0.86; sensitivity: 86%–93%; specificity: 73%–81%) [113].

A 10-serum miRNA signature (miR-320, miR-223, miR-222, miR-221, miR-199a-5p, miR-152, miR-145, miR-25, miR-24, miR-20a), was identified and validated in two large independent cohorts as a valuable biomarker for early diagnosis of NSCLC [114]. A unique set of six upregulated serum miRNAs (miR-429, miR-205, miR-203, miR-200b, miR-125b, and miR-34b) was found able to detect NSCLC early, with an AUC of 0.89, 85% sensitivity, and 74% specificity [115]. Another plasma miRNA panel, composed of upregulated miR-4478 and miR-448, previously found aberrantly expressed in different cancer types, diagnoses LCs (AUC, specificity, sensitivity: 0.90, 89%, 79%) and, more specifically, early NSCLC with elevated accuracy (AUC, specificity, sensitivity: 0.90, 90%, 76%) [116]. A multiphase case-control study identified the plasma panel composed of miR-628-3p, miR-532, and miR-425-3p as accurately discriminating early NSCLC patients from healthy controls (AUC, sensitivity, specificity: 0.97, 92%–97%, 95%–98%). This panel distinguishes NSLCL from benign lung diseases and discriminates different LC subtypes. MiR-425-3p and miR-628-3p are responsive to surgery (downregulation) [117]. Using a fluorescence quantum dot liquid bead array, the serum panel composed of downregulated miR-20a-5p and miR-16-5p, and upregulated miR-15b-5p accurately discriminated NSCLC patients from healthy controls with an 86%–94% sensitivity range and 91%–94% specificity rage [118]. The combination of plasma miR-19b (upregulated) and miR-183 (downregulated) expression levels provided LC detection, with 95% specificity and 95% sensitivity (AUC = 0.99). MiR-19b has the greatest diagnostic potential in discriminating SCC, while miR-183 for ADC from healthy controls (AUC, sensitivity, specificity: 0.83, 75%, 91%, and 0.92, 100%, 81%, respectively) [119]. A 4-plasma miRNA signature (miR-205-5p, miR-210, miR-145, and miR-126) discriminates NSCLC patients with 92% sensitivity and 96% specificity [120]. A 10-serum miRNA signature (miR-423-3p, miR-221, miR-199a-3p, miR-151-5p, miR-148b, miR-126-3p, miR-27b, miR-26a, miR-23b, and let-7f, all upregulated) was validated for distinguishing LC from healthy subjects together with a 4-miRNA panel (miR-423-3p, miR-221, miR-148b, miR-23b) that showed an elevated discriminatory power for LC (AUC: 0.89 and 0.89, respectively) [121]. The combination of plasma miR-223, miR-145, miR-21, and miR-20a (all upregulated) worked as a sensitive biomarker for early-stage NSCLC patients compared to healthy controls (AUC, specificity, sensitivity: 0.90, 82%, 90%) [122]. Serum miR-15b-5p/miR-146b-3p, miR-20a-5p/miR-146b-3p, and miR-19a-3p/miR-146b-3p ratios were validated as independent diagnostic biomarkers for discriminating between NSCLC and benign pulmonary nodules (AUC ranging from 0.72 to 0.85), whereas miR-15b-5p/miR-146b-3p, miR-20a-5p/miR-146b-3p, miR-20a-5p/miR-17-5p, miR-92a-3p/miR-20a-5p, miR-16-5p/miR-20a-5p, and miR-16-5p/miR-19a-3p efficiently discriminated NSCLC from pulmonary inflammation diseases (AUC ranging from 0.54 to 0.80) [123]. Similarly, a 10-miRNA ratio panel, composed of 14 plasma miRNAs, accurately discriminated between LC patients and healthy controls with an AUC of 0.98 [124]. The serum miR-223 + miR-145 + miR-20a panel also accurately discriminated NSCLC patients from healthy controls with an AUC of 0.88. In particular, miR-145 was downregulated in all-stage NSCLC patients compared to healthy controls, miR-20a downregulation differentiated Stage I–II and Stage III NSCLC patients from healthy controls, while miR-223 upregulation discriminated Stage IV NSCLC patients from healthy controls [125]. An additional 2-miRNA panel (miR-139-5p+miR-152-3p) performed well with AUC, specificity, and sensitivity of 0.80, 88%, and 95% [126].

CT is used as the primary screening tool for early detection of LC in heavy smokers. Ten miRNAs are deregulated in LC tissue from patients identified by CT compared to nontumor tissue: miR-7, miR-21, miR-200b, miR-210, miR-219-1, miR-324 are upregulated, while miR-126, miR-451, miR-30a, and miR-486 are downregulated. In addition, miR-205 and miR-21 accurately discriminate ADC from SCC, miR-518e and miR-144 are significantly downregulated in rapidly growing tumors, and miR-429 is inversely correlated with DFS. The same miRNA expression analysis, performed on normal lung tissue, identified miR-126*, miR-126, let-7c, miR-222, miR-30e, miR-1-2, miR-29b-1, miR-30d-prec, miR-15a, and miR-16 that are associated with early detection. MiR-379 and miR-29-1* are correlated with reduced forced expiratory volume (FEV), miR-30d* with increased growth rate, and miR-34b with worse DFS [127]. Microarray analysis revealed that a signature of 16 plasma miRNA ratios, involving 15 upregulated miRNAs, identified subjects that were developing cancer (AUC 0.85, 80% sensitivity, and 90% specificity), with miR-660, miR-451, miR-140-5p, miR-92a, miR-30c, and miR-28-3p as the most frequently upregulated miRNAs. In addition, a signature composed of 16 miRNAs ratios, involving 13 deregulated miRNAs, discriminated patients from healthy subjects (AUC 0.88, 75% sensitivity, and 100% specificity). Furthermore, two other signatures of 10 miRNA ratios each, composed of nine and 11 miRNAs, respectively, identified patients with poor prognosis (80% sensitivity and 100% specificity) or aggressive disease (88% sensitivity and 100% specificity), with miR-660, miR-486-5p, miR-451, miR-221, miR-197, miR-140-5p, miR-106a, miR-28-3p, and miR-16 as the most deregulated miRNAs. These four ratio signatures were further validated in a larger cohort of samples, prospectively collected within the randomized Multicenter Italian Lung Detection (MILD) clinical trial of low-dose-CT observation. Results again underlined the significant diagnostic and prognostic performance of plasma miRNA signatures complementary to CT imaging, in order to reduce the false-positive rate [128].

#### 2.1.2. Circulating miRNAs as Markers for Metastatic LC, Tumor Recurrence, and Response to Adjuvant Therapies

Some papers specifically focused their attention on the identification and validation of circulating miRNAs as biomarkers for metastatic NSCLC. miR-222, miR-183, and miR-126 were significantly downregulated in serum samples of Stage IV NSCLC patients compared to healthy controls, but only miR-183 and miR-126 maintained this trend when compared to Stage I/II patients, thus indicating that these two miRNAs could be useful as serum biomarkers for more advanced NSCLC [129]. Conversely, plasma miR-422a upregulation discriminated LC patients with lymphatic metastasis from non-metastatic patients (AUC = 0.79) [130].

Half of the NSCLC patients that undergo complete tumor resection may suffer local or distant recurrence. Consequently, to reduce this risk, patients are treated with adjuvant therapies. The whole miRNome of NSCLC patients monitored after tumor resection could be useful to reveal metastasis occurrence [131]. Early-stage ADC patients with tumor recurrence, metastasis, and not submitted to adjuvant therapy showed increased serum levels of miR-142-3p and miR-29b compared to the non-recurrence group that is associated with reduced OS [132]. Of these, only miR-142-3p, alone or in combination with the tumor stage, remained an accurate serum biomarker for recurrence risk in patients with poor outcome that received adjuvant therapy (AUC of 0.64 and 0.78, respectively). In addition, miR-486, miR-221, miR-30d, and miR-1 altogether are correlated with shorter DFS in NSCLC patients after tumor resection [133].

Platinum-based chemotherapy combined with radiotherapy represents the first-line treatment for locally advanced NSCLC. Circulating miRNAs have been investigated over their predictive potential of response to chemotherapy. High levels of miR-125b have been found in the serum of NSCLC patients resistant to cisplatin-based chemotherapy compared to nonresistant, and its levels have been associated with cancer stage and differentiation status [134]. Plasma miR-150 and miR-29a levels are inversely correlated with administered radiotherapy dose in Stage IIIA NSCLC patients [135]. Dynamic changes of plasma miR-125b and miR-19b expression levels, and miR-125b/miR-19b ratio are valuable markers for discriminating NSCLC patients responsive to chemotherapy [136]. Another study suggested that elevated plasma miR-613, miR-495-3p, miR-302e, and miR-98-5p levels might mark sensitivity to radiotherapy in NSCLC [137]. Serum let-7 levels change in response to radiotherapy in NSCLC patients, and up- or downregulation of this miRNA is correlated with cancer-cell proliferation rate during therapy [138]. Eleven serum miRNAs, miR-205-5p, miR-200b-3p, miR-155-5p, miR-145-5p, miR-134miR-126-3p, miR-125b-5p, miR-92a-3p, miR-34a-5p, miR-22-3p, and miR-10b-5p, combined with clinical factors, have been used to create a predictive dose response score (DRS) for NSCLC patients treated with radiotherapy [139]. Low DRS and high-dose radiation therapy are associated with longer OS, and reduced risk of distant metastasis and tumor progression, compared to patients treated with a standard dose of radiation.

Among cancer therapies, immunotherapy is considered a promising strategy. A 7-serum miRNA signature (miR-548j-5p, miR-495-3p, miR-494-3p, miR-493-5p, miR-411-3p, miR-215-5p, miR-93-3p) is significantly associated with >6 month OS in LC patients treated with immune checkpoint inhibitor nivolumab (AUC, sensitivity, specificity: 0.81-71%-90%) [140]. Advanced NSCLC patients, clinically improved following nivolumab treatment, had lower plasma miR-375 and miR-320b levels than patients with early progression [141].

#### 2.1.3. LC miRNome

A valuable retrospective study investigated the potential of serum miRNome as a prognostic and diagnostic tool for LC. The authors compared the serum miRNA profile of LC patients to those obtained immediately or many years before diagnosis. The serum miRNA profile of samples taken after LC diagnosis was more comparable to that from samples collected closest prior to diagnosis than that collected years before. Consequently, the serum miRNA profile seemed to change with the development of the tumor. Moreover, the serum miRNA signature of healthy controls was more similar to that of patient samples collected many years before diagnosis (four deregulated miRNAs) compared to that closer to or after diagnosis (19 and 29 deregulated miRNAs, respectively) [142]. Through NGS, 32 known miRNAs and 7 novel miRNAs were identified as significantly altered in WB from LC patients compared to healthy subjects [143]. Using a similar approach, a 34-serum miRNA signature discriminated both asymptomatic early-stage and advanced-stage NSCLC patients from healthy smokers (AUC: 0.89 for Stage I NSCLC, and 0.88 for Stage II–IV NSCLC) [144]. A 24-plasma miRNA panel able to discriminate early-stage NSCLC patients from healthy controls was identified and validated (AUC, sensibility, sensitivity: 0.92, 83%, 80%). The diagnostic accuracy of this panel increased when considered in combination with age, sex, and smoking habits (0.94, 83%, 84%) [145]. Five, 10, and 50 blood-based miRNA sets discriminated with elevated accuracy, specificity, and sensitivity NCSLC patients from healthy controls (95% accuracy and AUC of 0.98 for five markers; 95% accuracy and AUC of 1.00 for 10 markers; 98% accuracy and AUC of 0.99 for 50 markers) [146]. In addition, the whole miRNome of NSCLC patients monitored after tumor resection could be useful to reveal metastasis occurrence [131].

In conclusion, the 10 miRNA ratios [124] and the miR-183 + miR-19b panel identified LC patients with the highest accuracy [119], while the miR-152 + miR-148a + miR-148b + miR-21 panel [110] showed the highest ability in discriminating NSCLC patients from healthy controls. Notably, miR-125b was able to accurately discriminate among all NSCLC stages [73], whereas the miR-301 + miR-200b + miR-193b + miR-141 [107] and miR-125a-5p + let-7e + CEA panels [101] specifically distinguished early-stage NSCLC patients. Additionally, the miR-199a-5p + miR-152-3p + miR-145-5p + miR-25-3p + miR-24-3p + miR-20a-5p panel could efficiently discriminate between NSCLC patients and COPD [98]. The miR-486 + miR-221 + miR-30d + miR-1 combination identified high-risk recurrence NSCLC patients [133], the panel composed of miR-548j-5p, miR-495-3p, miR-494-3p, miR-493-5p, miR-411-3p, miR-215-5p, and miR-93-3p strongly predicted >6 months OS in LC patients after nivolumab treatment [140], miR-422a levels revealed metastasis presence [130], and miR-613, miR-495-5p, miR-302e, miR-98-5p were correlated with sensitivity to radiotherapy [137]. Finally, the miR-3135b + miR-550a-3p + miR-151a-3p + miR-151a-5p + miR-151b + miR-139-5p panel could discriminate SCC cases [126], the combination of miR-155 with CEA and CA-125 revealed ADC cases [78], and the miR-155 + miR-145 + miR-21 and miR-486-5p + miR-210 + miR-21 panels showed the highest discriminatory power for differentiating tumors from benign pulmonary nodules [68,99].

For LC, one of the most studied circulating miRNAs as diagnostic and prognostic biomarker is miR-21. Interestingly, all studies that evaluated the circulating levels of this miRNA showed increased levels of miR-21 in LC patients vs. the control group, with AUC from 0.59 to 0.87, despite the different employed preanalytical protocols (matrix: plasma vs. serum; centrifugation speed and length) and analytical methods (RT-qPCR, NGS, microarray followed by RT-qPCR).

All information about the circulating miRNAs associated with LC are summarized in Table 1.

### 2.2. Circulating miRNAs and Breast Cancer (BC)

Breast cancer (BC) is the most prevalent tumor in women, and mammography is considered the gold-standard method for its diagnosis; however, mammography presents some limitations, including the use of ionizing radiation and the possibility of false-positive results. Clinical decisions about BC treatment depend on its molecular features, such as the estrogen receptor (ER), the progesterone receptor (PR), and the human epidermal growth factor receptor 2 (HER2). To date, the most used marker in clinical routine for detecting BC is CA 15-3, but it presents low sensitivity. Other serum markers, such as CEA and TPS, are even less sensitive than CA 15-3. Based on these facts, novel biomarkers for BC diagnosis and prognosis are desirable.

#### 2.2.1. Circulating miRNAs as Biomarker for BC Diagnosis and Prognosis

The first report comprehensively screening circulating miRNAs in BC identified the downregulation of let-7c and the upregulation of miR-589 in Caucasian American (CA) cases as potentially discriminating BC from healthy controls (AUC: 0.78–0.84 and 0.62–0.85, respectively). Similarly, let-7d* and miR-425* are useful for African American (AA) cases (AUC: 0.73–0.99 and 0.79–0.83, respectively) [147]. Microarray screening identified miR-202 (upregulated in WB) and miR-484 (upregulated in serum) as potential diagnostic markers [148,149]. With a similar approach, it was found that, in BC tissue compared to the surrounding normal tissue, miR-200a-3p, miR-183-3p, miR-142-3p, miR-141-3p, miR-96-5p, and miR-21-5p are upregulated, whereas miR-3656, miR-638, miR-505-5p, and miR-125b-5p are downregulated. After validation, miR-505-5p, miR-125b-5p, miR-96-5p, and miR-21-5p are deregulated in plasma from BC patients compared to healthy controls. Of these, miR-505-5p and miR-96-5p have turned out to be the most valuable markers (AUC, sensitivity, specificity: 0.72, 75%, 60%, and 0.72, 73%, 66%, respectively) [150]. MiRNA expression profiling identified miR-4270, miR-1290, miR-1225-5p, miR-1207, and miR-1202, whose expression is upregulated in early-stage compared to advanced BC [151]. Five other circulating miRNAs, found upregulated in WB from BC patients compared to healthy controls, have high discriminating power (AUC, sensitivity, specificity: 0.93, 80%, 100% for miR-30b-5p; 0.77, 53.3%, 100% for miR-96-5p; 0.76, 53.3%, 92.3% for miR-182-5p; 0.83, 86.7%, 69.2% for miR-374b-5p; 0.81, 66.7%, 100% for miR-942-5p). Of these, miR-942-5p, miR-374b-5p, and miR-182-5p are inversely correlated with survival [152]. MiR-1915-3p (upregulated) and miR-455-3p (downregulated) have also acquired diagnostic value in identifying BC patients from healthy controls since ROC analysis gives AUC values of 0.88 and 0.79 [153].

Previous evidence about deregulated miRNAs found in other cancer types, in BC tumor tissue, and BC-derived cell lines has given the basis for novel investigations in the field of BC diagnosis and prognosis. One of the first reports validated the upregulation of miR-195 and let-7a as discriminating BC cases from healthy controls. However, only miR-195 showed the same altered regulation in BC tissue compared to the non-cancerous surroundings. Moreover, blood miR-195 is correlated with tumor stage and lymph-node metastasis. Higher circulating levels of miR-10b and miR-21 have been observed in ER^-^ compared with ER^+^ BC patients [154]. BC tissue and serum are characterized by the upregulation of miR-155, miR-106a, and miR-21, and the downregulation of miR-335, miR-199a, and miR-126. Their relative expression, but not miR-106a, is strictly associated with tumor grade and ER/PR expression [155]. MiR-155 is highly accurate in discriminating BC patients from healthy controls, with AUC, sensitivity, and specificity of 0.80, 65%, and 81% [156].

One of the most studied miRNAs due to its established role in several cancers is miR-21. Circulating miR-21 is also upregulated in BC, and a first report has revealed its elevated accuracy in discriminating both BC patients from healthy controls and Stage IV from early-stage BC (AUC, sensitivity, specificity: 0.72, 75%, 67%, and 0.833, 86%, 70%, respectively) [157]. Genome-wide NGS screening found coregulation of miR-21 in both tissue and serum from BC [158]. This analysis also identified miR-29a [158] and miR-222 [159] as upregulated in BC patients compared to control subjects. MiR-92a, found downregulated in BC, has accuracy as a BC biomarker comparable to miR-21 (AUC: 0.92 vs. 0.93). As miR-21, serum miR-92a is correlated with BC tissue expression, lymph-node metastasis, and tumor size [160]. Serum miR-195, miR-181a, and miR-30a are also downregulated in BC, and their diagnostic accuracy in all-stage and early-stage BC showed sensitivity values (59%–70.7% for miR-181a; 74% for miR-30a; 68%–69% for miR-195) higher than those of common BC diagnostic markers CEA and CA153 (sensitivity: 8.4%–15.0% for CEA, 10.5%–21.0% for CA153) [161,162,163]. MiR-195 is also correlated with high sensitivity (52%) to neoadjuvant chemotherapy (NCT) response. Compared to healthy controls, oncogene miR-182 is downregulated in both BC tissue and serum, and is inversely correlated with ER^+^ and PR^+^ status [164]. MiR-200c downregulation in WB discriminates all-stage and early-stage BC patients from healthy subjects (AUC, sensitivity, specificity: 0.79, 90%, 70.2%, and 0.82, 90%, 75%, respectively), and it is considered a negative prognostic marker [165]. BC patients also show altered serum levels of miR-652-3p, miR-425-5p, miR-148b-3p, and miR-145-5p (downregulated), and miR-10b (upregulated) compared to healthy controls. Of these, miR-652-3p and miR-148b-3p display efficient discriminatory ability, with AUC of 0.75 and 0.70, respectively, while miR-10b is strongly correlated with poor patient prognosis [166]. Upregulation of miR-598-3p and miR-382-3p, and downregulation of miR-1246 and miR-184 are also associated with BC and can discriminate patients from healthy subjects (AUC, sensitivity, specificity: 0.94, 95.0%, 85.0% for miR-598-3p; 0.74, 52.0%, 92.5% for miR-382-3p; 0.90, 93.0%, 75.0% for miR-1246; 0.74, 87.5%, 71.0% for miR-184) [167]. More recently, other circulating miRNAs have displayed diagnostic potential in BC: miR-103a and miR-24 are downregulated in serum, while serum miR-140-3p and plasma miR-520g are upregulated [168,169,170]. MiR-520g is also strongly associated with lymph-node metastasis and tumor-differentiation degree [169]. Interestingly, serum miR-140-3p is specifically upregulated in premenopausal BC patients compared to premenopausal healthy controls [170].

The combination of mammography and blood-based miRNAs profiles can increase diagnostic accuracy and eventually anticipate treatment initiation. A 9-serum miRNA signature (miR-425, miR-365, miR-145, miR-143, miR-139-5p, miR-133a, miR-107, miR-18a, and miR-15a) discriminated early-stage ER^+^ BC from healthy controls with an AUC range of 0.61–066. This signature predicted, with 73% probability, BC development within one year in women [171,172]. Women at high risk for BC were characterized by a six-serum miRNAs signature able to predict BC onset within six months with an AUC of 0.90 [173].

In many cases, circulating miRNA panels were identified as markers for BC diagnosis and prognosis. The combination of plasma miR-451 and miR-145 has high accuracy in discriminating BC patients from both healthy controls and other types of cancers (AUC, sensitivity, specificity: 0.93–0.96, 83%–90%, and 89%–93%) [174]. A less sensitive but still accurate panel for BC diagnosis, composed of seven serum miRNAs (miR-801, miR-652, miR-409-3p, miR-376a, miR-376c, miR-148b, miR-127-3p) and three plasma miRNAs gave AUC, sensitivity, and specificity values of 0.81 and 0.69, 80% and 70%, and 72% and 55%, respectively [175,176]. Combinations of four miRNAs (miR-133a, miR-133b, miR-92a, miR-1), upregulated in both BC tissue and serum compared to their nontumor counterparts, gave AUC values of 0.90 (miR-133b + miR-92a), 0.91 (miR-133a + miR-92a), and 0.90 (miR-92a + miR-1) [177]. Very high diagnostic performances were obtained with the ratio of serum miR-382, miR-155, and miR-145 (AUC, sensitivity, specificity: 0.99, 98%, 100%) [178], as well as with serum miR-451 + miR-148a + miR-30b + miR-27a and plasma miR-148b + miR-133a panels (AUC, sensitivity, specificity: 0.95, 95%, 83%, and AUC = 0.86, respectively) [179,180]. An additional 8-miRNA plasma panel (let-7d, let-7i, miR-148a, miR-107, miR-103, miR-22*, miR-19b, and miR-16) accurately discriminated BC patients from healthy controls, although with low specificity (AUC, sensitivity, specificity: 0.81, 91%, 49%). Sensitivity dropped to 80% in the case of metastatic BC (mBC) [181]. The combination of miR-6861-5p, miR-1307-3p, miR-1246 (upregulated in BC), and miR-6875-5p, miR-4634 (downregulated in BC) accurately discriminated BC patients and healthy controls with AUC = 0.97, 97% sensitivity, and 83% specificity [182]. A serum panel composed of miR-574-5p, miR-155, let-7a, and metastasis-associated lung adenocarcinoma transcript 1 (MALAT1), had high discriminatory potential between BC patients and healthy controls: AUC ranged from 0.96 to 0.97, sensitivity and specificity were 90%–94% and 97%–99%, respectively. Of these four ncRNAs, miR-155 was downregulated after chemotherapy [183]. Considering the miR-106a–363 cluster, a 4-plasma (miR-106a-3p + miR-106a-5p + miR-92a-2-5p + miR-20b-5p) and 4-serum (miR-106a-5p + miR-92a-3p + miR-20b-5p + miR-19b-3p) miRNA panels, all upregulated in BC, accurately discriminated cases from controls with AUC, sensitivity and, specificity values of 0.85–0.90, 82%–83%, and 79%–80% (plasma panel), and 0.91–0.97, 87%–94%, and 87%–94% (serum panel). In addition, the plasma levels of miR-106a-3p, miR-106a-5p, miR-92a-2-5p, and miR-20b-5p were correlated with tumor stage and hormone-receptor status [184]. Finally, the combination of serum miR-195 and let-7a, both downregulated in BC, gave AUC of 0.75 and 0.72 in discriminating cases from either healthy controls or patients with benign lesions, respectively [185].

#### 2.2.2. Circulating miRNAs as Markers for Metastatic BC, Tumor Recurrence, and Response to Adjuvant Therapy

Based on ER, PR, and HER2 receptor expression, BC can be grouped under different types. The triple-negative BC (TNBC: ER^−^, PR^−^, HER2^−^) accounts for approximately 15%–20% of all BC subtypes and is associated with poor prognosis and early tumor recurrence. A 4-serum miRNA signature (miR-652, miR-107, miR-103, miR-18b, all upregulated in BC) is strongly associated with tumor recurrence and worse survival in TNBC patients; thus, it can be considered as a useful discriminator index for tumor relapse [186]. Other plasma miRNAs accurately discriminate TNBC from non-TNBC: downregulated miR-199a-5p, miR-21, miR-16 (AUC: 0.88, 0.80, 0.87, respectively) and upregulated miR-489, miR-200b, miR-193b, miR-125b, miR-105, miR-93-3p (AUC: 0.99, 0.88, 0.91, 0.97, 0.93, 0.66, respectively). In addition, miR-199a-5p is correlated with hormone-receptor expression levels [187,188,189]. More recently, a 7-serum miRNA panel (miR-489-3p + miR-199a-3p + miR-195-5p + miR-15a-5p + miR-7-5p + let-7c-5p + let-7i-5p) was identified as a valuable diagnostic marker for TNBC patients with an AUC of 0.93 [190].

HER2 overexpression is associated with more aggressive tumor behavior and poor prognosis for mBC patients. The most successful therapy in HER2-positive patients is the use of trastuzumab, an anti-HER2 monoclonal antibody. In mBC patients, trastuzumab combined with lapatinib, an inhibitor of HER2-associated intracellular tyrosine kinases, improves therapeutic effectiveness. Non-mBC HER2^+^ patients have higher serum miR-21 expression levels than non-mBC HER2^-^ (AUC = 0.71), whereas mBC HER2^+^ have higher serum miR-10b compared to mBC HER2^-^ (AUC = 0.75). In addition, elevated serum miR-19a discriminates inflammatory mBC from non-inflammatory mBC (AUC = 0.75), and is associated with better prognosis [191]. The miR-940 +miR-451a + miR-17-3p + miR-16-5p serum signature effectively predicts one- and two-year OS in HER2^+^ mBC treated with trastuzumab (AUC 0.78–0.80 and 0.74–0.77, respectively, 73% specificity, and 75% sensitivity) [192].

The BC luminal-A subtype is ER^+^ and/or PR^+^, HER2^−^ and Ki67 < 14%. These patients are characterized by better prognosis but are less responsive to chemotherapy than other BC subtypes. Elevated serum levels of miR-205 and miR-19a in luminal-A BC patients treated with NCT (epirubicin and paclitaxel) are correlated with chemotherapy-resistance onset, and the combination of these two miRNAs can effectively predict chemosensitivity and discriminate resistant patients from the sensitive group [193]. A 5-serum miRNA panel (miR-328-3p + miR-199a-3p + miR-195-5p + miR-25-3p + let-7i-5p) diagnosed luminal-A BC with an AUC of 0.94 [190]. Plasma miR-155, miR-21, miR-10b (upregulated compared to healthy controls) and let-7a (downregulated) turned out to be potential biomarkers for luminal-A BC monitoring. In fact, after surgery, chemotherapy, and radiotherapy, their levels normalized [194].

In BC patients treated with combined chemotherapy, serum miR-125b was higher in non-responsive than in responsive patients. In addition, miR-125b was directly correlated with the number of proliferating BC cells, and inversely with the proportion of apoptotic cells in tissue samples after chemotherapy [195]. In another independent cohort, serum miR-125b and miR-21 during NCT were associated with positive chemotherapy response and DFS. Considered alone or in combination, miR-125b and miR-21 gave elevated accuracy in discriminating responder from nonresponder BC patients (AUC: 0.77–0.78, 0.87–0.93, and 0.96, respectively) [196]. Serum miR-3200 and miR-451 levels were significantly decreased in BC patients during NCT and are correlated with a better prognosis [197]. Based on miRNA profiling results, miR-222, miR-451, and miR-20a plasma-level changes in HR^+^/HER2^-^ BC patients have been associated with chemoresistance. In particular, pre-NCT miR-222 upregulation, post-NCT miR-20a upregulation, and miR-451 downregulation are strongly associated with unresponsiveness to NCT in HR^+^/HER2^-^ BC patients (AUC: 0.80, 0.79, and 0.71). Moreover, miR-34a is lower in both TNBC and HR^+/^HER2^-^ NCT-unresponsive BC subjects compared to responsive patients (AUC = 0.59) [198]. In Stage II–III BC, high serum miR-122 is associated with metastatic recurrence after chemotherapy [199]. In early-stage BC patients after tumor removal and therapy, the serum levels of miR-181b, miR-155, miR-24, and miR-19a were significantly upregulated in a high-risk recurrence group compared to low-risk patients, thus indicating their prognostic behavior [200,201]. MiR-106b expression was induced in both tissue and plasma from BC patients that experienced metastatic tumor recurrence compared to patients without tumor relapse (AUC, sensitivity, specificity: 0.86, 88%, 60% for plasma miR-106, and 0.79, 88%, 58% for tissue miR-106b). In addition, this is correlated with tumor size [202]. Starting from global screening, a 7-serum miRNA panel (miR-375, miR-205-5p, miR-194-5p, miR-21-5p upregulated, and miR-382-5p, miR-376c-3p, miR-411-5p downregulated) has been associated with BC recurrence (AUC, specificity, sensitivity: 0.91, 93%, 77%) [203]. Recurrent BC is associated with higher plasma miR-200c, miR-23b, miR-21 levels, and lower miR-190 levels compared to non-recurrent BC. Circulating levels of miR-200c and miR-21, either alone or combined with miR-190, effectively predict tumor recurrence (AUC, sensitivity, specificity: 0.68, 76%, 61%, 0.69, 71%, 64%, and 0.77, 80%, 65%). The prediction accuracy of the 3-plasma miRNA panel increased when combined with axillary lymph-node infiltration and tumor grade (AUC, sensitivity, specificity: 0.87, 89%, 76%). Moreover, elevated levels of miR-21 are associated with poor DFS and OS, and high miR-200c is associated with reduced DFS [204]. Serum miR-4530 levels are associated with sensitivity to NCT: in sensitive patients, miR-4530 is higher than in resistant subjects (AUC = 0.66, 98% sensitivity, and 86% specificity) [205].

Efforts have also been made to validate circulating miRNAs discriminating mBC from non-mBC. Upregulation of serum miR-10b and miR-34a, and downregulation of miR-155 are indeed effective. Of these, miR-10b and miR-34a identify mBC cases from healthy controls, and miR-34a is correlated with advanced-tumor grade in BC patients [206]. MiR-155 has been identified in another mBC cohort (AUC = 0.78) and, together with miR-17, downregulated as well, had elevated discriminatory potential for mBC patients (AUC = 0.68), and is associated with ER^+^/PR^+^ hormone-receptor status [207]. Downregulated serum miR-411 and miR-299-5p, found in both BC tissue and serum, strongly discriminated healthy controls from mBC patients [208]. A 6-plasma miRNA panel (miR-486-5p + miR-215 + miR-210 + miR-200a + miR-200b + miR-200c) identified BC patients that developed metastasis early within two years prior to clinical diagnosis compared to those that remained metastasis-free (AUC, sensitivity, specificity: 0.82, 77%, 75%). They are also correlated with OS and DFS [209]. Recently, serum let-7 downregulation and plasma miR-24-3p upregulation have been associated with metastasis development, and effectively discriminated mBC patients from both non-mBC and healthy subjects [210,211]. Plasma miR-10b and miR-373, either alone or combined, effectively discriminated lymph-node mBC from non-mBC with AUC, specificity, sensitivity values of 0.80, 71%, 72% (miR-10b), 0.84, 68%, 89% (miR-373), 0.88, 72%, 94.3% (miR-10b + miR-373) [212]. The downregulation of plasma miR-146a, miR-130a, and miR-16 is also correlated with lymph-node mBC [213] as well as serum miR-455-3p and miR-1915-3p, which are down- and upregulated, respectively, in patients with lymph-node mBC, with AUC values of 0.80 and 0.89 [153]. Considering mBC with distant metastasis (bone, liver, and lung), serum miR-205, and miR-155 levels are higher than non-mBC and show moderate discriminatory ability with AUC, sensitivity, specificity of 0.66, 61%, 78% and 0.67, 50%, 85%, respectively [214]. Bone metastasis (bmBC) is a main complication of BC and occurs in 65%–75% of patients with advanced BC. MiR-10b is upregulated in BC patients compared to heathy controls and discriminates bmBC patients from non-mBC (AUC, sensitivity, specificity: 0.77, 65%, 70%) [215].

Circulating-tumor-cell (CTC) count emerged as a promising biomarker in mBC, but its accuracy is still under evaluation. The combination of CTCs with circulating miRNAs may give a prognostic advantage. Plasma miR-200b (upregulated) accurately discriminated CTC + mBC from CTC cases (AUC, sensitivity, specificity: 0.88, 80%, 83%). In addition, a 5-miRNA (miR-768-3p + miR-210 + miR-200b + miR-200c + miR-141) and a 3-miRNA (miR-768-3p + miR-210 + miR-200c) panel effectively distinguished CTC+ and CTC- cases from heathy controls, respectively (AUC, sensitivity, specificity: 0.95, 90%, 91%, and 0.78, 80%, 65%) [216]. Using the serum-direct multiplex qRT-PCR (SdM-qRT-PCR) approach, a novel 3-serum miRNA panel with elevated BC diagnostic potential was identified in the combination of upregulated miR-424, miR-199a, and miR-29c (AUC, sensitivity, specificity: 0.90–0.91, 77%–78%, and 85%–89%) [217]. The diagnostic specificity of CTC for mBC patients improved to 100% when combined with miR-21, usually upregulated in BC [218].

#### 2.2.3. BC miRNome

The BC blood miRNome analysis can be a future powerful approach for early identification of BC. A prospective study provided complete serum miRNA screening from women that later developed BC or remained cancerfree, and identified a panel of 21 differently expressed miRNAs [219]. Another study identified a 41-tissue miRNA signature able to predict BC with an accuracy of 63%–83%. Of these, 20 miRNAs were detected and validated in serum, and turned out to predict BC occurrence within 18 months from blood collection with an accuracy of 53% [220].

Among all the aforementioned miRNAs, the miR-145, miR-155, and miR-382 ratio [178], miR-6875-5p + miR-6861-5p + miR-4634 + miR-1307-3p + miR-1246 panel [182], and miR-574-5p + miR-155 + let-7a + MALAT1 panel [183] discriminate with the highest power BC patients from healthy women. In the case of metastasis development, miR-10b accurately identified BC patients with bone metastasis [215], and miR-1915-3p BC patients with lymph-node metastasis [153]. The miR-486-5p + miR-215 + miR-210 + miR-200a + miR-200b + miR-200c panel prognostically identified BC patients that would develop metastasis within two years of BC onset [209]. The miR-411-5p + miR-382-5p + miR-376c-3p + miR-375 + miR-205-5p + miR-194-5p + miR-21-5p panel accurately predicted BC patients who would develop recurrence after tumor removal and treatment [203], while the miR-940 + miR-451a + miR-17-3p + miR-16-5p panel predicted one- and two-year OS of HER^+^ mBC patients treated with trastuzumab [192]. After chemotherapy treatment, the miR-125b + miR-21 panel efficiently identified responder BC patients [196], miR-4530 sensitive patients [205], and the miR-205 + miR-19a panel resistant luminal A BC patients [193]. Finally, miR-200b discriminated CTC-positive from CTC-negative cases [216], miR-125b identified TNBC patients [189], and the miR-425 + miR-365 + miR-145 + miR-143 + miR-139-5p + miR-133a + miR-107 + miR-18a + miR-15a panel ER^+^ BC patients [189].

In the case of BC, one of the most studied circulating miRNAs as diagnostic and prognostic biomarker is miR-21. All BC studies that evaluated the circulating levels of miR-21 showed increased levels of this miRNA in BC patients vs. control groups, with an AUC from 0.69 to 0.93, despite the different employed preanalytical protocols (matrix: plasma vs. serum; centrifugation speed and length) and analytical methods (RT-qPCR, microarray followed by RT-qPCR).

All information related to the above-described circulating miRNAs is reported in Table 2.

### 2.3. Circulating miRNAs and Prostate Cancer (PC)

One of the most prevalent malignancies in males is prostate cancer (PC). Currently, the combination of prostate-specific antigen (PSA) quantification and digital rectal examination are considered the gold standard for the early detection of PC. Unfortunately, the diagnostic ability of PSA is limited by low specificity that frequently gives false-positive results in patients with benign prostatic hyperplasia (BPH). Based on this consideration, novel biomarkers with higher specificity and sensibility are needed.

#### 2.3.1. Circulating miRNAs as Biomarkers for PC Diagnosis and Prognosis

The serum levels of miR-195, miR-26a, and miR-let7i are higher in PC patients compared to BPH, but only miR-26a discriminates PC from BPH (89% sensitivity, 56% specificity) [221]. MiR-221 and miR-21 have the highest discriminatory ability for PC patients from healthy controls (AUC: 0.83 and 0.89, respectively) [222]. A 5-plasma miRNA signature was identified by microarray analysis: let-7c, let-7e, and miR-30c are downregulated in PC patients compared to healthy subjects, whereas miR-1285 and miR-622 are upregulated. ROC analysis revealed elevated diagnostic potential, and further principal component analysis (PCA) indicated that this 5-miRNA panel could differentiate PC from both BPH and healthy controls with an AUC of 0.92 and 0.86, respectively [223]. MiR-628-5p, miR-101, and miR-25 downregulation in serum from PC have very high discriminatory potential between PC and healthy subjects (AUC = 0.94 for miR-628-5p, AUC = 0.80 for miR-101, AUC = 0.66 for miR-25); of these, miR-25 and miR-101 have been previously associated with PC pathogenesis [224]. Starting their identification in urine as potential biomarkers for PC, serum miR-1825, miR-484, and miR-205 have been identified as downregulated in PC, whereas serum miR-141 and let-7b as upregulated. Each of these miRNAs accurately and independently diagnosed PC with AUC, sensitivity, specificity values of 0.96, 93%, 91% (miR-1825), 0.79, 88%, 69% (miR-484), 0.91, 78%, 100% (miR-205), 0.93, 88%, 100% (miR-141), and 0.85, 72%, 88% (let-7b). In addition, higher levels of miR-1825 are correlated with advanced tumor grade, lowered after treatment. Serum let-7b and miR-205 downregulation is correlated with advanced tumor stage, hormone resistance, and metastasis occurrence [225]. Using an innovative approach based on NGS and in situ hybridization (ISH), miR-148a-3p was found upregulated in both serum samples and formalin-fixed/paraffin-embedded (FFPE) PC tissue [226].

Evidence suggests that single circulating miRNAs may have poorer diagnostic power than PSA in PC; hence, combined panels or specific ratios have been studied. The serum miR-519c-5p + miR-345 + miR-19a/b miRNA signature effectively discriminated high- and low-risk PC (AUC = 0.94) [227], while the combination of WB miR-155 + miR-145 + miR-141 + let-7a gave an AUC of 0.78. Compared to BPH, PC patients displayed an upregulation of oncomirs miR-155, miR-145, and miR-141, and a downregulation of tumor-suppressor let-7a; their tissue deregulation is associated with increased risk of PC development [228]. The combination of plasma let-7c, miR-375, miR-141, miR-30c (downregulated in PC), and PSA better discriminated PC from BPH and healthy controls (AUC: 0.78 and 0.88, respectively) than PSA alone [229]. Plasma miR-375 and miR-21, which are upregulated in PC compared to BPH, when combined with PSA have higher diagnostic potential (AUC, sensitivity, specificity: 0.88, 88%, 75%) than those given by each single miRNA (0.76, 75%, 75% for miR-375; 0.80, 88%, 75% for miR-21). Of these, miR-21 is correlated with tumor stage and metastasis [230]. Interestingly, in these two studies, the levels of miR-375 were found both upregulated and downregulated in PC vs. BPH patients despite their evaluation being performed in the same matrix (plasma) and with similar analytical and postanalytical protocols.

The same plasma miR-21 upregulation has been associated with miR-106 and miR-20a upregulation, and miR-223 downregulation. MiR-106a/miR-130b and miR-106a/miR-223 ratios, alone or in combination, effectively discriminate between lPC and BPH patients (AUC: of 0.81, 0.77, and 0.84, respectively) with specificity higher than that of PSA [231]. A subsequent study identified another plasma miRNA panel for PC diagnosis, composed of miR-4289, miR-326, miR-152-3p, and miR-98-5p, all upregulated in PC patients compared to healthy controls. ROC analysis revealed that each miRNA of this panel has diagnostic potential, but accuracy is highly increased for the whole panel (AUC = 0.82–0.95) [232]. However, according to the Cancer Genome Atlas (TCGA) dataset, miR-152-3p expression in PC tissue may be downregulated compared to adjacent normal tissue. Serum miR-326 upregulation, in accordance with its tissue expression, marks the risk for tumor recurrence, together with miR-185-5p and miR-221-3p, after radical prostatectomy [233].

Specific circulating miRNAs associated with other cancer types or deregulated in PC tissue have been tested in PC patients in order to identify potential diagnostic and prognostic circulating biomarkers. MiR-139-5p was upregulated in WB from PC patients compared to BPH, accurately discriminated PC patients from BPH and healthy subjects (AUC: 0.94 and 0.92, respectively), and is correlated with advanced and more aggressive tumor stages [234]. Serum miR-410-5p levels are higher in PC patients compared to healthy controls or non-PC patients, and are correlated with high-risk PC development. ROC analysis revealed that elevated miR-410-5p accurately discriminated PC from non-PC/healthy subjects (AUC = 0.81), and high-risk from low-risk subjects (AUC = 0.71) [235]. WB miR-18a, a previously identified oncomir, discriminated PC patients from both BPH and healthy groups (AUC = 0.81), and its high circulating levels are correlated with more aggressive tumor behavior and advanced clinical stage [236]. Another set of serum miRNAs (let-7c, let-7e, let-7i, miR-26a-5p, miR-26b-5p, miR-25-3p, miR-18b-5p), found altered in PC tissue, discriminate PC patients from BPH. Of these, miR-25-3p, miR-18b-5p, and their combination gave the highest sensitivities and specificities in predicting PC (AUC: 0.79, 0.87, and 0.92, respectively) [237]. Based on the TCGA database, circulating miR-200b and miR-200c have been validated as potential diagnostic markers for PC (AUC: 0.57 and 0.62, respectively). Compared to healthy controls, in plasma from PC patients, miR-200b was downregulated and is correlated with bone metastasis, whereas miR-200c was upregulated and is associated with more aggressive behavior [238]. Serum upregulation of miR-375, miR-141-3p, miR-106b, miR-34a-5p, and miR-21 also discriminated between PC patients and healthy subjects. Among these, miR-375 and the combination miR-375 + miR-141-3p + miR-21 have been identified as the most accurate diagnostic tools for PC (AUC, sensitivity, specificity: 0.91, 100%, 75% for miR-375 alone, and 0.86, 93%, 63% for the combination) [239].

A combination of interacting plasma miRNAs (miR-17/miR-192) and of three other independent miRNAs (miR-181a, miR-150a, and miR-22) discriminated between aggressive and nonaggressive PC. Interestingly, low miR-192 and high miR-17 circulating levels are the most likely to present aggressive PC due to a synergic effect of the inhibition of tumor-suppressing target genes (by miR-17) and induction of oncogenes (by miR-192) [240]. Later, it was highlighted that both up- and downregulation of circulating miR-17 are correlated with more aggressive PC, thus underlining the importance of this miRNA in PC behavior [241].

#### 2.3.2. Circulating miRNAs as Biomarkers for Metastatic PC, Tumor Recurrence, and Response to Adjuvant Therapy

The first attempt to identify circulating miRNAs as diagnostic biomarkers for PC has been performed by Mitchell et al., who discovered that serum miR-141 upregulation effectively discriminates metastatic PC (mPC) patients from healthy individuals with 60% sensitivity and 100% specificity [32]. These findings were confirmed by the discovery of miR-141 and miR-375-altered regulation in both PC tissue and serum [242]. MiR-141 effectively identifies mPC patients compared to local PC (lPC) ones (AUC = 0.76) [222]. High sensitivity and 100% specificity were determined for miR-562 + miR-551b + miR-501-3p + miR-375 + miR-210 panel that identified 84% of PC patients, let-7a*+ miR-616 + miR-56 2+ miR-210 panel that discriminated 80% of mPC patients from BPH, and miR-1203 + miR-708 + miR-375 + miR-200a that distinguished 75% of mPC from lPC patients [243]. In mPC, miR-375, miR-17-3p, miR-27a-3p, miR-200a-3p, and miR-376b-3p are highly expressed (AUC = 0.895) in the metastatic tissue of PC patients after radical prostatectomy compared to those who remained cancerfree [244]. MiR-93 tissue and plasma expression levels are significantly correlated, and upregulation is associated with mPC. Furthermore, plasma miR-93 decreases after lPC treatment. In parallel, tumor suppressor let-7b and oncogenic miR-21 are respectively down- and upregulated in both PC tissue and plasma [245].

Several papers have focused on circulating miRNAs as prognostic biomarkers for PC recurrence. Early PC stage is defined as hormone-sensitive PC (HSPC) because its growth depends on androgen. When HSPC regresses after radical prostatectomy (RP), it is possible that recurrence follows evolving in castration-resistant PC (CRPC) [246]. Serum levels of mir-378*, miR-375, and miR-141 gradually increase over PC progression (from low- to high-risk and CRPC), and miR-375 and miR-141 are similarly coregulated in PC tissue, thus indicating the potential effectiveness of these two circulating miRNAs to monitor PC evolution [246]. Upregulation of serum miR-194 and miR-146b-3p moderately discriminated PC recurrent patients from non-recurrent ones (AUC: 0.65 and 0.62, respectively), and miR-146b-3p levels are correlated with reduced recurrence-free interval [247]. Microarray screening identified 10 other serum miRNAs differently regulated in metastatic CRPC (mCRPC) compared to lPC: miR-423-3p, miR-375, miR-200c, miR-152, miR-151-3p, miR-141, miR-126, and miR-21 are upregulated, whereas miR-205 and miR-16 are downregulated. Among these, miR-423-3p, miR-205, miR-152, miR-151-3p, and miR-141 have been associated with adverse clinicopathological features. mMiR-375, miR-205 miR-151-3p, miR-141, miR-126, miR-21, and miR-16 display moderately accurate diagnostic ability for mCRPC, but only the miR-151-3p + miR-141 + miR-16 combination revealed discriminatory potential comparable to PSA (AUC: 0.94, 84% sensitivity, 96% specificity) [248].

The standard chemotherapy for CRPC is based on docetaxel; serum miR-21 is elevated in docetaxel-resistant CRPC patients [249]. A 6-serum/plasma miRNA panel (miR-301b, miR-222, miR-200c, miR-200b, miR-146a, miR-20a) is correlated with the response to docetaxel in CRPC. Particularly, increased serum levels of miR-200 family members before docetaxel treatment or lower/unchanged levels of miR-17 family members after docetaxel treatment are associated with poor response to chemotherapy and reduced OS [250]. After the validation stage, high plasma levels of miR-429, miR-375, miR-200a, miR-200b, miR-200c, and miR-132 before chemotherapy have been correlated with shorter OS without any association with the response [251].

In the past few years, active surveillance for an early detection of PC patients has increased, but conventional markers do not effectively differentiate PC stages. A 3-serum miRNA panel (miR-375, miR-223, miR-24,) discriminates indolent from aggressive PC (AUC: 0.69 increased to 0.70 when combined with PSA) [252].

In conclusion, miR-1825 reveals the highest diagnostic ability in discriminating PC patients from healthy subjects [225], while miR-139-5p from BPH subjects [234]. The miRNA panel composed of miR-519c-5p, miR-345, and miR-19a/b shows elevated capacity to distinguish high-risk PC from low-risk PC [228]. The miR-1203 + miR-708 + miR-375 + miR-200a and miR-151-3p + miR-141+ miR-16 panels strongly differentiate mPC and mCRPC, respectively, from lPC patients [243,248]. MiR-200b pre-docetaxel levels efficiently predict death within 12 months [250], while miR-194 and miR-146b-3p predict PC recurrence [247].

All information about circulating miRNAs in PC is reported in Table 3.

### 2.4. Circulating miRNAs and Osteosarcoma

The most common human primary malignant bone tumor in young adults and children is osteosarcoma [253].

#### 2.4.1. Circulating miRNAs as Biomarkers for Osteosarcoma Diagnosis and Prognosis

Several studies, aimed at evaluating novel circulating miRNAs as potential biomarkers for the diagnosis, prognosis, and treatment of osteosarcoma patients, started from the validation of miRNAs that were previously demonstrated to play a crucial role in the cell differentiation, proliferation, and tumorigenesis of various cancer types. MiR-21 is one of the most studied miRNAs due to its association with several types of cancer. MiR-21 is upregulated in serum samples from osteosarcoma patients compared to their healthy counterparts, and it is also correlated with advanced Enneking stages, reduced tumor response to chemotherapy, and short OS [254]. This finding is in accordance with increased plasma levels of miR-21 in another osteosarcoma cohort. The same study also identified miR-199a-3p and miR-143 as downregulated in osteosarcoma subjects compared to healthy subjects, and ROC analysis revealed that the combination of these three plasma miRNA signatures successfully discriminated between osteosarcoma and healthy subjects with AUC of 0.95, 91% sensitivity, and 94%, specificity [255]. Significantly higher serum levels of miR-199a are also found in preoperative osteosarcoma patients compared to healthy subjects, which returned to normal values after tumor resection. MiR-199a shows high diagnostic accuracy for osteosarcoma (AUC: 0.86, 88% sensitivity, 77% specificity) [256]. A case-control study identified that miR-34b plasma levels in osteosarcoma patients was significantly decreased in comparison to healthy subjects, as well as in metastatic vs. non-metastatic patients; therefore, its expression is associated with increased osteosarcoma risk [257]. Serum miR-9 was also identified as a potential diagnostic and prognostic marker, since it is upregulated in osteosarcoma compared to healthy subjects and is correlated with aggressiveness features, advanced tumor stage, size, and metastasis presence [258]. Members of the miR-29 family are aberrantly expressed in several types of human cancer. MiR-29a, miR-29b, and miR-29c are upregulated in osteosarcoma patients’ sera compared to normal control levels, and circulating miR-29a and miR-29b expression is associated with tumor grade, recurrence, metastasis presence, and OS, thus indicating their possible use as biomarkers for human OS prognosis, development, and progression [259]. Two other miRNAs are correlated with clinicopathological features and unfavorable prognosis: miR-196a and miR-196b, which belong to the human miR-196 cluster. The serum levels of these miRNAs are higher in osteosarcoma patients’ sera compared to healthy subjects, and are associated with tumor grade, metastasis development, and recurrence, reduced OS, and DFS. Furthermore, their coexpression is a potentially independent biomarker for patient survival and prognosis [260]. ROC curve analysis identified plasma miR-148a as a circulating biomarker upregulated in osteosarcoma patients compared to healthy subjects (AUC, sensitivity, specificity: 0.78, 70%, 83%), and is correlated with reduced OS and DFS, tumor size, and metastasis [261]. MiR-195, another potential diagnostic and prognostic serum biomarker for OS, with 88% sensitivity and 83% specificity, is associated with an advanced clinical stage, distant metastasis formation, and reduced OS and DSF [262]. Using a different approach, a panel of four upregulated miRNAs (miR-374a-5p, miR-320a, miR-199a-3p, miR-195–5p), out of 739 screened, was identified in the plasma samples of osteosarcoma patients (AUC, sensitivity, specificity: 0.96, 91%, 94% in distinguishing osteosarcoma patients from healthy controls). MiR-199a-3p and miR-195-5p levels are also associated with metastasis and decrease after tumor removal, whereas miR-320a and miR-199a-3p are related with osteoblastic subtypes [263]. Results about miR-199a-3p and miR-195–5p are in contrast with two previous studies that observed their downregulation in osteosarcoma patients’ blood samples [255,262]. This discordance can be due to the different analytical (RT-qPCR alone vs. microarray + RT-qPCR) and postanalytical (normalization: cel-miR-39 or snRNA U6) approaches used for miRNA evaluation and data elaboration. The profile of 752 plasma miRNAs in osteosarcoma patients revealed the downregulation of miR-205-5p and the upregulation of miR-574-3p, miR-335-5p, and miR-214, compared to healthy subjects, with good discrimination potential (AUC: 0.70, 0.88, 0.78, and 0.80). Furthermore, plasma miR-214 levels are associated with the presence of metastasis [264]. MiR-27a aberrant expression is associated with several types of cancer. Osteosarcoma patients’ serum samples show increased expression levels of this miRNA that is also correlated with aggressive clinicopathological features: advanced clinical stage, distant metastasis, and poor response to chemotherapy. Moreover, serum miR-27a levels can distinguish osteosarcoma patients from control subjects with an AUC of 0.87, 70% sensitivity, and 98% specificity [265]. Wang and colleagues identified miR-191 as able to distinguish between healthy and osteosarcoma subjects with an AUC of 0.86, 74% sensitivity, and 100% specificity. MiR-191 is significantly overexpressed in osteosarcoma patients’ serum compared to healthy subjects, and it is associated with negative clinicopathological features and prognosis in osteosarcoma patients: advanced stage, large tumor size, metastasis presence, as well as reduced OS and DFS [266].

Several miRNAs aberrantly expressed in other cancer types have been evaluated as potential markers in osteosarcoma diagnosis and for identifying individuals with negative prognosis. MiR-152, for instance, gives AUC of 0.96, 93% sensitivity, and 97% specificity. It is downregulated in serum samples from osteosarcoma patients compared to healthy subjects, and is correlated with distant metastasis and shorter survival [267]. MiR-221 plays a crucial role in osteosarcoma occurrence and progression, and it is significantly upregulated in osteosarcoma patients’ serum compared to healthy subjects. High miR-221 serum levels are correlated with stage, metastasis, and shorter survival. Moreover, it reveals osteosarcoma presence with elevated AUC, sensitivity, and specificity (0.84, 66%, and 100%) [268]. Circulating miR-451a, miR-425-5p, miR-139-5p, miR-106a-5p, miR-25-3p, miR-20a-5p, and miR-16-5p are all downregulated in OS patient samples (AUC: 0.80, 0.78, 0.71, 0.73, 0.80, 0.85, and 0.77). Unfortunately, these data are less robust than those of previous studies due to missing information about their correlation with clinicopathological features [269]. The potential oncogenic role of miR-421, miR-17, miR-542-3p, miR-300, miR-222, and the tumor-suppression role of miR-223, miR-326, miR-95-3p, miR-491-5p, miR-375, miR-101, and miR-let-7a, whose circulating expression levels are associated with several types of cancer, have been studied in osteosarcoma patients’ samples. Plasma miR-421, upregulated in osteosarcoma, is associated with poor OS [270], similarly to serum miR-17, which is also associated with worse OS prognosis [271]. Plasma miR-542-3p displays high discriminatory capacity between osteosarcoma patients and healthy subjects (AUC, sensitivity, specificity: 0.84, 78%, 94%), and survival analysis also revealed strong correlation between miR-542-3p levels and DFS and tumor stage [272]. Oncogenic behavior was also revealed by circulating miR-300, whose serum levels are increased in osteosarcoma patients; it effectively identifies osteosarcoma (AUC, sensitivity, specificity: 0.89, 84%, 89%; positive–negative predictability: 90% and 85%). Furthermore, it is associated with advanced clinical stage and metastasis, and independently predicts poor prognosis (reduced OS and DFS) [273]. Another unfavorable independent predictor marker for osteosarcoma OS and DFS has been identified in serum miR-222. It is strongly upregulated in osteosarcoma patients compared to healthy subjects and is associates with metastasis, tumor size, and clinical stage (AUC, sensitivity, specificity: 0.81, 67%, 84) [274]. The downregulation of serum miR-223 in osteosarcoma patients compared to healthy subjects is inversely correlated with clinical stage, metastasis development, and directly with OS (AUC, sensitivity, specificity: 0.92, 86%, 97%) [275]. Serum miR-326 downregulation in osteosarcoma patients compared to healthy subjects (AUC = 0.90, 84% specificity, 95% sensitivity) is associated with negative clinicopathological features (distant metastasis and advanced clinical stage) and poor prognosis [276]. Analogously, serum miR-497 downregulation in osteosarcoma (AUC = 0.85) is associated with reduced OS, metastasis presence, advanced tumor clinical stage, and poor response to chemotherapy [277]. Circulating miR-95-3p can also discriminate between osteosarcoma patients and healthy subjects with an AUC of 0.86. In addition, low serum levels of miR-95-3p independently predict OS and are associated with osteosarcoma development and progression [278]. Serum miR-491-5p upregulation discriminates osteosarcoma and healthy subjects with high accuracy (AUC, sensitivity, specificity: 0.83, 72%, 86%). Moreover, serum miR-491-5p levels are strongly associated with clinical stage and distant metastasis. Survival analysis reveals that reduced serum miR-491-5p expression is significantly associated with negative prognosis and independently predicts osteosarcoma [279]. Convincing evidence reveals that circulating miR-101 may be a moderately accurate biomarker for osteosarcoma diagnosis and prognosis. It is dramatically downregulated in serum samples from osteosarcoma patients and is powerfully associated with negative clinicopathological features (advanced tumor stage and metastasis) and worse prognosis (shorter OS and DFS). After tumor removal and therapeutic treatment, patients without metastasis showed higher miR-101 values, thus indicating that this circulating miRNA can be useful for monitoring patients’ chemotherapy response. The diagnostic potential of serum miR-101 for osteosarcoma is underlined by ROC analysis (AUC = 0.85, 76% sensitivity, and 83% specificity), while survival analysis revealed that miR-101 may be an independent and negative prognostic factor [280]. MiR-let-7a, quantified in total blood of osteosarcoma patients, is considered a potential diagnostic (AUC = 0.90) and negative prognostic biomarker for osteosarcoma [281]. In a small Mexican cohort, microarray analysis of 648 serum miRNAs revealed that only miR-642a-5p and miR-215-5p are upregulated in osteosarcoma, with AUC of 0.84 and 0.87 [282].

#### 2.4.2. Circulating miRNAs as Markers for Metastatic Osteosarcoma, Tumor Recurrence, and Response to Adjuvant Therapies

The expression pattern of miR-21 in osteosarcoma patients has been deeply investigated before and after chemotherapy in order to explore its correlation with chemosensitivity [283]. Consistent with other studies [254,255], miR-21 is remarkably upregulated in serum from osteosarcoma patients in comparison with healthy subjects and, after tumor removal, its serum level before and after chemotherapy is significantly lower in nonresponders than in responders. Moreover, PDCD4, a miR-21 target, is upregulated in serum from patients after chemotherapy [283]. The combined downregulation of miR-133b and miR-206 is associated with aggressive tumor progression, metastasis formation and recurrence, and impaired response to chemotherapy [284]. Low serum miR-375 expression level in osteosarcoma patients is associated with advanced clinical stage, important tumor size, metastasis formation, and poor response to chemotherapy. It discriminates osteosarcoma patients from control subjects (AUC, sensitivity, specificity: 0.89, 82%, 75%), and poor from good tumor drug response (AUC, sensitivity, specificity: 0.83, 84%, 84%) as well as a negative prognostic index [285].

#### 2.4.3. Osteosarcoma miRNome

Fujiwara et al. screened the serum miRNome in osteosarcoma patients in comparison to nonosteosarcoma patients with other benign tumors and healthy volunteers. The serum concentration of miR-25-3p in osteosarcoma patients is higher than in non-osteosarcoma and healthy groups, while serum concentrations of miR-17-5p was increased only in relation to healthy volunteers [286], which was consistent with a previous observation [271]. ROC analysis revealed that both serum miR-25-3p and miR-17-5p levels strongly discriminated osteosarcoma patients from healthy individuals (AUC, sensitivity, specificity: 0.87, 71, 92% and 0.72, 64%, 85%, respectively). Elevated miR-25-3p levels predict negative prognosis and OS, and are correlated with metastasis at diagnosis [286].

Genome-wide screening revealed that miR-221, miR-106a, and miR-21 are significantly upregulated in osteosarcoma patients compared to healthy subjects, with good discriminatory power (AUC: 0.83, 0.96, and 0.85), but not in determining prognosis [287]. MiR-124 also discriminated osteosarcoma patients from healthy subjects (AUC, sensitivity, specificity: 0.85, 80%, 86%): low levels are associated with advanced tumor stage, metastasis presence, and worse prognosis [288].

Among all the aforementioned miRNAs, the plasma panel composted of miR-374a-5p, miR-320a, miR-199a-3p, and miR-195–5p showed the highest diagnostic ability for discriminating osteosarcoma patients from healthy subjects, with an AUC of 0.961 [263], while miR-375 was the only circulating miRNA able to discriminate between good and poor response to chemotherapy (AUC, sensitivity, specificity: 0.83, 84%, 84.4%) [285].

All the evidence on circulating miRNAs and osteosarcoma reported above is summarized in Table 4.

## 3. Conclusions

In tumor biology, miRNAs can act as oncogenes or tumor suppressors and their altered tissue expression, as well as their altered circulating levels, has been correlated with the occurrence of several cancers. Therefore, miRNAs are considered novel potential molecular diagnostic and/or prognostic non-invasive tools. However, besides the (sometimes confliting) evidences discussed above, this correlation has been reported for multiple but not for all tumors. Indeed, the level of diasease- and tumor-associated circulating miRNAs is affected by several factors including, but not limited to, age, gender, ethnicity, diseases and comorbidities, drugs, smoking, diet, physical activity, and lifestyle [36,289,290]. For instance, miR-122 is elevated in 70% of hepatocellular carcinoma (HCC) cases, but it is also increased in case of hepatitis B infection or liver injury [291] and it fluctuates in response to therapy in chronic hepatitis C [292]. Similarly, platelets significantly contribute to the circulating miRNA profile; therefore, any intervention therapy that targets platelets, including common antiplatelet therapies for cardiovascular disorders, can alter the circulating miRNA profile [47]. On the same basis, given the biological variability (as both intra- and inter-individual variability) of circulating miRNA and the plethora of variables affecting their levels (including the stage of the treatment, commonly used drugs as NSAIDs corticosteiroids), the current evidence suggests that miRNAs may not yet be usable as universal markers for response to chemotherapy [293]. Another main issue is the lack of their cancer type-specificity, i.e., the same miRNA is altered in multiple cancers. Circulating miR-21 is altered in several cancers (e.g., head and neck, breast, colorectal, hepatocellular, pancreatic, other than in those mentioned in this review) however, miR-21 levels are also associated with inflammatory and wound-healing responses. miR-92 levels are altered in both ovarian and colorectal cancers and it also marks the surgical response [87]. Furthermore, both miR-21, miR-92a behave differently in different cancers. Only, the detailed definition of their biological role in the neoplastic transformation process will lead to the identification of a putative marker, selected throughout a rigorous literature search taking into account as many aspects as possible, as a definitive biomarker [293]. These concerns must not discourage the path towards the search for tumor-associated circulating miRNAs. Indeed, as reported in the conclusive parts of each section of this review, promising markers are available at least for breast, lung, prostate cancers, and osteosarcoma. Their priceless value resiedes in their association with fine molecular mechanisms taking place in a group of cells or in a tissue assayable without the need of any invasive procedure and this is of particular interest for those tissues in which biopsy can be difficult (e.g., bone [294], pituitary gland [295]).

On the other hand, there are fields in which research investigating miRNA-based circulating biomarkers is strongly needed. Pituitary adenomas, for instance, are among the most frequent intracranial tumors with an incidence rate of 10–15%. Although mostly begnign, they manifest through several morbidities mainly associated with the compression produced on the adjacent structures and/or hormone disturbances. Although the expression profile and role of tissue miRNAs are quite well investigated, information about the diagnostic potential of pituitary adenoma-associated circulating miRNAs are still lacking. Indeed, if hormone monitoring represents the best way to follow tumor growth and function and, therefore, the role of diagnostic miRNAs could be less important, in the case of non-functional pituitary adenomas a blood-based miRNA biomarker could help the diagnosis and patient follow-up after surgery. Potential diagnostic markers such as miR-26a, miR-126, miR-300, miR-329, miR-381, miR-665 in GH-secerning adenomas, miR-26a in corticotropic adenomas, miR-432 and miR-410 in prolactinomas, miR-106b in non-functioning adenomas and metastatic pituitary carcinomas, are currently the most promising [296].

Another important point not addressed by this review that deserves attention is the role of extracellular vesicle (EV)-associated miRNAs (exosomes and microvesicle-associated miRNAs). Rather than biomarkers, miRNAs contained into EVs act as hormone-like mediators since they target more or less specifically cells/tissues different from those of origin, thanks to the interaction between surface proteins expressed on the EV and the target cell. Once fused with (or endocytosed by) the target cell, the EV releases its content into the cell. Consequently, the EV-associated miRNA species act in this cell by modulating the gene expression and, hence, the cell function. By contrast with free extracellular miRNAs which, generally, mirror the cell content since being passively released by the cell (e.g., membrane leaking, autophagy, cell death), EV-associated miRNAs are actively and specifically loaded into an EV in order to be addressed to a target cell and, hence, their intra-vesicle concentration does not necessarily mirror the intracellular concentration [297]. The pathogenic role of EVs is, currently, a hot topic in biomedical sciences. Indeed, as carriers of information (e.g., miRNAs), EVs are fundamental modifiers of the evolution of a pathological condition as demonstrated in different fields such as heart disease [298], osteoporosis [294] and sarcopenia [299], diabetes [300], and cancer [297]. The role of EVs in cancer have been established since the demonstration of their implication in virtually every aspect of tumorigenesis. EVs can, indeed, mediate the “transfer” of a phenotype and, for instance, vesicles generated by highly aggressive cancer cells can stimulate growth, survival, and migration ability to less malignant cancer cells. This phenomenon is particularly relevant in primary tumors that often consist of heterogeneous populations of cells with different levels of aggressiveness. EVs are also key determinants of the tumor microenvironment and of the promotion of invasion in a two-way fashion that sees tumor cells and surrounding normal cells exchanging information that finally drives the process. Similarly, the recipient tissue of a metastasis is “prepared” by the primary tumor and circulating cancer cells also throughout EVs [297]. In this process, EV-associated miRNAs play a pivotal role. Yu and colleagues have recently demonstrated that HCC cells under hypoxic conditions (i.e., those found into a tumor mass) showed an increased exosomal production and these exosomes, in turn, enhanced proliferation, migration, and invasiveness and induced the epithelial-to-mesenchymal transition in HCC cells under normoxic conditions. EV-derived miR-1273f, an activator of Wnt/β-catenin signaling pathway, was identified as the main mediator of this intercellular information transfer [301]. Exosome-associated miRNAs also drive the complex modulatory effects exerted by mesenchymal stem cells (MSCs) on tumor growth and progression, as recently investigated by Che and coworkers. They demonstrated that MSC-derived exosomal miR-143 suppressed proliferation, migration, invasion, and tumor growth and induced apoptosis in prostate cancer cells by targeting trefoil factor 3 (TFF3) [302].

In conclusion, considering all the evidence reported in this review, altered circulating miRNA levels may be correlated with cancer development and progression (tumor stage and metastasis presence), thus underlining their potential as diagnostic and prognostic biomarkers. MiRNAs may also be useful tools for monitoring patients during chemotherapy treatment as well as after surgery, and can be considered as prognostic markers for patient survival. In some cases, circulating miRNA panels, miRNA ratios, or a combination of circulating miRNAs levels with currently used cancer biomarkers strongly increase diagnostic and prognostic performance compared with single miRNAs or standard markers alone. Obviously, the presence of conflicting evidences about the regulation of specific circulating miRNAs in the studied type of cancer, together with the bias affecting the miRNA validation process, underline the necessity to standardize all preanalytical, analytical, and postanalytical protocols in order to obtain reliable biomarkers for cancer diagnosis and as potential tools for personalized treatments.

## Figures and Tables

**Table 1 jcm-08-01661-t001:** Circulating miRNAs associated with lung cancer (LC).

Cohort	Pre-Analytical Variables	Analytical Method		Identified miRNAs (AUC-Sensitivity%-Specificity%)	Study
		Screening	Validation		
Discovery: 11 NSCLC, 11 HC Validation: 22 NSCLC, 3 mesothelioma, 31 HS (all smokers)	**Plasma, serum**	880 miRNAs, 473 pre-miRNAs GenoExplorer microRNA Expression System Normalization: PC-U6B, U6-337, 5S-rRNA, PC-HU5S	Targets: miR-1268, miR-1254, miR-1228*, and miR-574-5p Method: RT-qPCR Normalization: RNU6, cel-miR-39	↑ miR-1254, miR-574-5p, miR-1254 + miR-574-5p (0.75-73%-71%) in NSCLC vs. HS	[54]
88 NSCLC, 17 age-/sex-matched HC	**Serum** Storage: −80 °C	427 miRNAs microarray chip	Target: miR-21 Method: RT-qPCR Normalization: snRNA U6	↑ miR-21 in NSCLC vs. HC	[55]
70 NSCLC, 44 HC	**Serum**	723 miRNAs microarray	Targets: miR-200c, miR-141, and miR-21 Method: RT-qPCR Normalization: snRNA U6	↑ miR-21 in NSCLC vs. HC	[56]
54 NSCLC, 46 HC	**EDTA-plasma** Processing within 2 h Centrifugation: (i) 20 min, 3000 rpm; (ii) 5 min, 14,000 rpm Storage: −80 °C	Targets: miR-486, miR-21 Method: RT-qPCR Normalization: miR-16		↑ miR-21 (0.74-50%-92%), ↓ miR-486 (0.86-70%-90%) in NSCLC vs. HC miRNA-21/miRNA-486 ratio (0.90-87%-87%) for NSCLC vs. HC	[57]
70 NSCLC (36 SCC, 34 ADC), 44 HC	**Serum**	Targets: miR-183, miR-182, miR-96 Method: RT-qPCR Normalization: snRNA U6, U48		↑ miR-183, miR-182, miR-96 in NSCLC vs. HC ↑ miR-96 in SCC vs. ADC	[59]
Discovery: 30 NSCLC, 20 HS Validation: 55 NSCLC, 75 HS	**Serum** Storage: −80 °C	328 miRNAs Method: RT-qPCR		↑ miR-15b/↓ miR-27b (0.98-100%-84%), ↑ miR-15a/↓ miR-27b in NSCLC vs. HS	[60]
97 NSCLC, 20 BPD, 30 HC	**Serum**	1158 miRNAs microarray on microfluid biochips Normalization: miR-1233	Targets: miR-625*, miR-361-3p Method: RT-qPCR Normalization: miR-1233	↓ miR-361-3p (0.86) in NSCLC vs. HC and BPD ↑ miR-625* in benign lung disease vs. HC and NSCLC ↓ miR-625* (0.77) in NSCLC vs. benign lung disease and HC	[61]
Discovery: 25 early stage NSCLC, 25 age-/sex-matched HC Validation: 126 early stage NSCLC, 50 age-/sex-matched HC	**K2-EDTA-plasma** Centrifugation: (i) 3000 rpm, 20 min; (ii) 12,000 rpm, 10 min, 4 °C Storage: −80 °C	Targets: miR-146, miR-204, miR-124, miR-106a Method: RT-qPCR Normalization: snRNA U6	Target: miR-204 Method: RT-qPCR Normalization: snRNA U6	↓ miR-204 (0.81-76%-82%) in early stage NSCLC vs. HC	[62]
18 NSCLC (9 ADC, 9 SCC), 18 HC	**Serum** Storage: −80 °C	NGS	Targets: miR-486-5p, miR-181b-5p, miR-103a-3p, miR-21-5p Method: RT-Qpcr Normalization: snRNA U6	↑ miR-181b-5p, miR-21-5p in SCC vs. HC ↑ miR-103a-3p, miR-21-5p, ↓ miR-486-5p in ADC vs. HC	[63]
11 NSCLC, 11 HC	**EDTA-plasma** Centrifugation: 1000 *g*, 30 min, 4 °C Storage: −80 °C	Targets: miR-486, miR-451, miR-210, miR-205, miR-155, miR-150, miR-126, miR-34a, miR-26b, miR-21 Method: RT-qPCR Normalization: cel-miR-39		↑ miR-486 (0.93-91%-82%), miR-150 (0.75-82%-82%) in NSCLC vs. HC	[64]
99 LC, 20 BPD, 76 HC	**EDTA-plasma** Centrifugation: 2000 rpm, 8 min Storage: −80 °C	MiRCURY LNA™ microRNA Arrays Normalization: median normalization method	Target: miR-30a Method: RT-qPCR Normalization: snRNA U6	↑ miR-30a (0.73-61%-84%) in NSCLC vs. benign lesions and HC miR-30a (0.73-55%-94%) for benign lesions vs. HC	[65]
54 NSCLC, 38 SPNs, 15 HC	**Serum** (left to clot 30-40 min) Centrifugation: 1000 g, 15 min Storage: −80 °C	754 miRNAs TaqMan Array Human MicroRNA A + B Cards Set V3.0 Normalization: cel-miR-39	Targets: miR-1244, miR-301a-3p Method: RT-qPCR Normalization: cel-miR-39	↑ miR-1244 in NSCLC vs. HC and SPNs miR-1244 (0.83-82%-80% and 0.80-54%-100%) for all stages and early stage NSCLC vs. HC miR-1244 (0.86-78%-92% and 0.85-73%-92%) for all stages and early stage NSCLC vs. SPNS	[66]
I stage: 28 NSCLC, 28 HS II stage: 58 NSCLC, 29 HS	**K2-EDTA-plasma** Processing within 2 h Centrifugation: 10 min, 1300 *g*, 4 °C Storage: −80 °C	Targets: miR-708, miR-486-5p, miR-429, miR-375, miR-210, miR-205, miR-200b, miR-182, miR-145, miR-139, miR-126, miR-21 Method: RT-qPCR Normalization: miR-16		↑miR-182 (0.66-52%-76%), miR-210 (0.75-74%-69%), miR-21 (0.82-79%-66%), ↓ miR-486-5p (0.88-85%-69%), miR-126 (0.76-69%-83%) in NSCLC vs. HS miR-486-5p + miR-210 + miR-126 + miR-21 (0.93-86%-97%) for NSCLC vs. HS	[67]
Training: 32 malignant SPNs, 33 SPNs, 29 HS Testing: 76 malignant SPNs, 80 SPNs	**K2-EDTA-plasma** Processing within 2 h Centrifugation: 10 min, 1300 *g*, 4 °C Storage: −80 °C	Targets: miR-486-5p, miR-375, miR-210, miR-126, miR-21 Method: RT-qPCR Normalization: miR-16		↑ miR-21 (0.59-56%-64%), miR-210 (0.69-56%-73%), ↓ miR-486-5p (0.63-72%-67%) in LC vs. SPNs and HS miR-486-5p + miR-210 + miR-21 (0.86-75%/76%-85%) for LC vs. SPNs and HS	[68]
74 LC (23 SCC, 18 ADC, 17 SCLC, 7 NSCLC, and 9 with carcinoid or mixed tumor), 68 age-matched HC	**EDTA-plasma** Centrifugation: 1600 *g*, 10 min, RT Storage: −80 °C	Targets: miR-221, miR-210, miR-205, miR-203, miR-199b, miR-197, miR-183, miR-182, miR-155, miR-128, miR-125b, miR-106a, miR-24, miR-21, miR-17 Method: RT-qPCR Normalization: miRNA expressed as absolute concentration		↑ miR-197 (0.88), miR-182 (0.71), miR-155 (0.87) in LC vs. HC miR-197 + miR-182 + miR-155 (0.90-81%-87%) for LC vs. HC	[69]
35 NSCLC, 30 HC	**Whole blood**	Target: let-7a Method: RT-qPCR Normalization: snRNA U6		↓ let-7a (0.95-90%-90%) in NSCLC vs. HC	[70]
82 LC, 50 HC	**Serum** Centrifugation: 15 min, 10,000 *g* Storage: −80 °C	Targets: miR-205, miR-30d, miR-24, and miR-21 Method: RT-qPCR Normalization: snRNA U6		↑ miR-205, miR-30d, miR-24, miR-21 in pre-operative vs. HC↓ miR-24, miR-21 in 10 days post-operative vs. paired pre-operative ↑ miR-205, miR-30d, miR-24 in 10 days post-operative vs. HC miR-205 (0.78-85%-72% and 0.81-100%-64%), miR-30d (0.74-76%-80% and 0.76-83%-64%), miR-24 (0.83-76%-64% and 0.86-83%-80%), miR-21 (0.69-46%-92% and 0.70-48%-88%), for early and advanced LC stage vs. HC miR-205 + CEA (0.66), miR-30d + CEA (0.83), miR-24 + CEA (0.75), miR-21 + CEA (0.70), for LC vs. HC	[71]
80 NSCLC, 60 HC	**Serum** Storage: −80 °C	Target: miR-21 Method: RT-qPCR Normalization: snRNA U6		↑ miR-21 (0.81-74%-72%) in NSCLC patients vs. HC	[72]
193 NSCLC, 110 HC	**Serum** Centrifugation: 4000 rpm, 10 min Storage: −80 °C	Target: miR-125b Method: RT-qPCR		↑ miR-125b (0.79-78%-66%) in NSCLC vs. HC ↑ miR-125b (0.66-96%-38%) in NSCLC stages I–II vs. HC ↑ miR-125b (0.84-93%-66%) in NSCLC stage III vs. HC ↑ miR-125b (0.90-95%-67%) in NSCLC stage IV vs. HC ↑ miR-125b (0.78-82%-68%) in NSCLC stage I–II vs. stage III ↑ miR-125b (0.67-41%-96%) in NSCLC stage III vs. stage IV	[73]
60 NSCLC (36 cisplatin-based chemotherapy, 24 fully resectable tumors), 30 HC	**Serum**, left to clot 60 min at RT Centrifugation: 1000 *g*, 10 min, 4 °C Storage: −80 °C	Target: miR-210 Method: RT-qPCR Normalization: miR-16		↑ miR-210 (0.78-79%-74%) in NSCLC vs. HC	[74]
24 HC, 86 NSCLC	**PAXgene-whole blood** Storage: −80 °C	Targets: miR-1224, miR-423, miR-361, miR-339, miR-328, miR-155, miR-140, miR-126, miR-93, miR-21, miR-22, miR-19a, miR-18a, let-7 Method: RT-qPCR Normalization: RNU38B, RNU58A		↑ miR-339 (0.76-0.74-0.80), miR-328 (0.79-0.82-0.75), miR-140 (0.71-0.69-0.74), miR-18a (0.76-0.77-0.76) in all stages, I-II stages, and III-IV stages NSCLC vs. in HC, respectively miR-361 + miR-328 (0.84). miR-140 + miR-339 (0.82) for late NSCLC stage vs. HC	[75]
Testing: 25 early stage NSCLC, 25 HC Validation: 126 early stage NSCLC, 60 HC, 42 non-cancerous pulmonary disease (25 COPD, 17 BPD)	**K2-EDTA-plasma** Centrifugation: 2000 *g*, 10 min, 4 °C Storage: −80 °C	Targets: miR-320, miR-221, miR-210, miR-182, miR-155, miR-145, miR-126, miR-30d, miR-25, miR-21, miR-20a Method: RT–qPCR Normalization: miR-16	Targets: miR-223, miR-155, miR-145, miR-21, miR-20a Method: RT–qPCR Normalization: miR-16	↑ miR-223 (0.94-0.96 for early stage NSCLC), miR-155 (0.92-0.94 for early stage NSCLC), miR-145 (0.77-0.82 for early stage NSCLC), miR-21 (0.77-0.79 for early stage NSCLC), miR-20a (0.89-0.91 for early stage NSCLC) in NSCLC vs. both non-cancerous pulmonary disease and HC ↑ miR-223 in non-cancerous pulmonary disease vs. HC	[76]
70 NSCLC, 70 HC	**Serum** Centrifugation: 3500 *g*, 5 min, RT Storage: −80 °C	Targets: miR-146a, miR-145, miR-125a-5p Method: RT-qPCR Normalization: miR-39		↑ miR-146a (0.78-84%-59%), miR-145 (0.84-93%-61%), miR-125a-5p (0.71-74%-56%) in NSCLC vs. HC	[77]
36 ADC, 32 age-/ sex-matched HC	**Serum** Centrifugation: 2000 rpm, 10 min, RT Storage: −80 °C	Targets: miR-155, miR-21 Method: RT-qPCR Normalization: snRNA U6		↑ miR-155 (0.76-72%-69%), miR-21 in ADC vs. HC CEA + CA-125 (89%-75%), CEA + miR-155 (83%-72%), CA-125 + miR-155 (89%-69%), CEA + CA-125 + miR-155 (92%-66%) for ADC vs. HC	[78]
Testing: 12 NSCLC, 12 HC Validation: 514 NSCLC, 54 age-/sex-matched HC	**Serum**	Target: miR-499 Method: RT-qPCR Normalization: cel-miR-39		↓ miR-499 (0.91-74%-93%) in NSCLC vs. HC ↓ miR-499 in stage III-IV NSCLC vs. stage I-II	[79]
120 NSCLC, 360 HC	**Plasma**	Targets: miR-152, let-7c Method: RT-qPCR Normalization: snRNA U6		↓ miR-152 (0.85-86%-81%), let-7c (0.71-72%-78%) in NSCLC vs. HC	[80]
100 NSCLC, 100 HC	**Plasma** Processing within 12 h Centrifugation: (i) 1500 rpm, 10 min; (ii) 12,000 rpm, 2 min Storage: −80 °C	Target: miR-195 Method: RT-qPCR Normalization: cel-miR-39		↓ miRNA-195 (0.89-78%-86%) in NSCLC vs. HC	[81]
153 NSCLC, 75 HC	**Serum** Processing within 30 min Centrifugation: 1500 *g*, 10 min, 4 °C Storage: −80 °C	Target: miR-411 Method: RT-qPCR Normalization: snRNA U6		↑ miR-411 (0.84) in NSCLC vs. HC	[82]
127 NSCLC, 60 HC	**Serum** Processing within 30 min Centrifugation: 2800 *g*, 10 min Storage: −80 °C	Target: miR-98 Method: RT-qPCR Normalization: snRNA U6		↓ miR-98 (0.86-80%-82%) in NSCLC vs. HC	[83]
91 NSCLC, 50 HC	**Serum** Centrifugation: 2000 *g*, 10 min, RT Storage: −80 °C	Target: miR-494 Method: RT-qPCR Normalization: cel-miR-39		↑ miR-494 (0.85-71%-92%) in NSCLC vs. HC	[84]
196 NSCLC, 77 HC	**Serum**	Target: miR-770 Method: RT-qPCR Normalization: GAPDH		↓ miR-770 (0.84-68%-89%) in NSCLC vs. HC	[85]
Training: 34 NSCLC, 20 HC Testing: 72 NSCLC, 50 HC	**EDTA-plasma** Processing within 2 h Storage: −80 °C	44 miRNAs miRCURY LNA™ universal RT microRNA PCR panel Normalization: miR-24-3p	Targets: miR-375, miR-328-3p, miR-141, miR-21 Method: RT-qPCR Normalization: miR-24-3p	↑ miR-141 (0.92-83%-98%), miR-21 and ↓ miR-375, miR-328-3p in NSCLC vs. HC miR-375 + mir-328 + miR-141 + miR-21 (0.62-62%-80%) for early stage NSCLC vs. HC	[86]
90 NSCLC, 90 HC	**EDTA-whole blood** Storage: −80 °C	Targets: miR-128-3p, miR-33a-5p Method: RT-qPCR Normalization: snRNA U6		↓ miR-128-3p (0.93-93%-80% all stages LC; 0.93-91%-86% early stage LC), miR-33a-5p (0.87-87%-73% all stages LC; 0.85-85%-76% early stage LC) in LC vs. HC miR-33a-5p + miR-128-3p (0.95-97%-83% all stages LC; 0.96-90%-93% early stage LC) for LC vs. HC	[87]
201 NSCLC, 103 age-/sex-matched HC	**Serum**	Target: miR-19a Method: RT-qPCR Normalization: snRNA U6		↑ miR-19a in LC vs. HC	[88]
221 LC, 54 age-/sex-matched HC	**Serum**	Target: miR-17-5p Method: RT-qPCR Normalization: snRNA U6		↑ miR-17-5p in LC vs. HC	[89]
Discovery: 217 NSCLC, 217 HC Validation: 53 NSCLC, 53 HC	**EDTA-plasma** Processing within 2 h Centrifugation: (i) 1500 rpm, 30 min; (ii) 3000 rpm, 5 min; (iii) 4500 rpm, 5 min Storage: −80 °C	Targets: miR-375, miR-31, miR-20a Method: RT-qPCR Normalization: cel-miR-39, snRNA U6		↓ miR-375 in NSCLC vs. HC	[90]
100 NSCLC, age-matched 100 HC	**Serum** Centrifugation: 3000 rpm, 15 min, RT Storage: −80 °C	Target: miR-138 Method: RT-qPCR Normalization: snRNA U6		↓ miR-138 in NSCLC vs. HC	[91]
100 NSCLC, age-matched 100 HC	**Serum** Centrifugation: 3000 rpm, 15 min, RT Storage: −80 °C	Target: miR-365 Method: RT-qPCR Normalization: snRNA U6		↓ miR-365 in NSCLC vs. HC	[92]
94 advanced NSCLC, age-/sex-matched 94 HC	**Serum**	Targets: miR-223, miR-146a, miR-19b Method: RT-qPCR Normalization: cel-miR-39		↑ miR-19b, ↓ miR-146a in NSCLC vs. HC	[93]
196 NSCLC, HC	**Plasma** Storage: −80 °C	Targets: let-7b, miR-378, miR-296, miR-210, miR-130a, miR-126, miR-92a, miR-20a, miR-19a, miR-19b, miR-18a, miR-17-3p, miR-17-5p Method: RT-qPCR Normalization: snRNA U6		↓ let-7b, miR-126, miR-18a, ↑ miR-378, miR-296, miR-210, miR-130a, miR-92a, miR-20a, miR-19a in NSCLC vs. HC	[96]
52 resectable NSCLC, 30 controls (10 COPD and 20-age/sex-/smoking status-matched healthy individuals)	**EDTA-Plasma** Processing within 4 h Centrifugation: 4 °C, 3000 rpm, 10 min Storage: −80 °C	Targets: miR-516-5p, miR-373-5p, miR-320-3p, miR-296-5p, mir-223-3p, miR-199a-5p, miR-191-5p, miR-155-5p, miR-152-3p, miR-145-5p, miR-129-5p, miR-126-3p, miR-96-5p, miR-25-3p, miR-24-3p, miR-20a-5p, let-7f-5p Method: RT-qPCR Normalization: mean of miR-192-5p, miR-16-5p		↑ miR-320-3p, miR-296-5p, miR-223-3p, miR-191-5p, miR-155-5p, miR-25-3p, and miR-20a-5p ↓ miR-199a-5p, miR-152-3p, miR-145-5p, miR-126-3p, miR-24-3p, let-7f-5p in NSCLC vs. controls miR-20a-5p + miR-25-3p + miR-155-5p + miR-223-3p + miR-296-5p + let-7f-5p + miR-24-3p + miR-126-3p + miR-145-5p + miR-152-3p + miR-199a-5p panel (0.88-81%-83%) for NSCLC vs. controls miR-152-3p + miR-145-5p + miR-199a-5p + miR-24-3p + miR-20a-5p + miR-25-3p (0.94-91%-83%) for NSCLC vs. COPD	[98]
Screening: 62 LC, 60 HS Validation: 34 LC, 30 SPNs, 32 HSs	**K2-EDTA-plasma** Processing within 2 h Centrifugation: 1300 *g*, 10 min, 4 °C Storage: −80 °C	Targets: miR-155, miR-145, miR-21 Method: RT-qPCR Normalization: snRNA U6B		↑ miR-155, miR-21, ↓ miR-145 in LC vs. non-cancer miR-21 + miR-145 + miR-155 (0.86-0.87; 69%-77%; 78%-81%) LC vs. HS miR-21 + miR-145 + miR-155 (0.84-77%-80%) for LC vs. SPNs	[99]
40 ADC, 260 age-/sex-/BMI-matched HC	**Serum**	667 miRNAs TaqMan low density arrays A + B cards Normalization: mean CT	Targets: miR-566, miR-550, miR-939, miR-616*, miR-146b-3p, miR-30c-1*, miR-339-5p, miR-656 Method: RT-qPCR Normalization: mean CT	↓ miR-656 (0.60), miR-339-5p (0.60), ↑ miR-939 (0.82), miR-616* (0.81), miR-566 (0.79), miR-550 (0.72), miR-146b-3p (0.71), miR-30c-1* (0.74) in ADC vs. HC miR-339-5p + miR-656 signature (0.60), miR-939 + miR-616* + miR-566 + miR-550 + miR-146b-3p + miR-30c-1* panel (0.70) for ADC vs. HC	[100]
76 NSCLC, 44 age-/sex-matched HC	**Plasma**	723 miRNAs Microarray	Targets: let-7e, miR-125a-5p, miR-30a, miR-30e, miR-30e-3p Method: RT-qPCR Normalization: the average of snRNA U6, snRNA U48	↓ let-7e, ↑ miR-30a, miR-30e-3p in NSCLC vs. HC ↓ miR-125a-5p (0.65-54%-75%) and let-7e (0.64-50%-83%) in early stage NSCLC vs. HC miR-125a-5p + CEA (0.66), let-7e + CEA (1.00), miR-125a-5p + let-7e + CEA (1.00) for early stage NSCLC vs. HC	[101]
70 NSCLC, 48 age-/sex-matched HC	**Serum**	Microarray	Targets: miR-429, miR-93, miR-29c Method: RT-qPCR Normalization: the average of snRNA U6, snRNA U48	↑ miR-29c, ↓ miR-429 in NSCLC vs. HC miR-29c (0.68-66%-74%), miR-429 (0.71-54%-81%), miR-29c + CEA (0.71), miR-429 + CEA (0.71), miR-29c + miR-429 + CEA (0.80) for NSCLC vs. HC miR-29c (0.73-50%-96%), miR-429 (0.72-94%-42%), miR-29c + CEA (0.76), miR-429 + CEA (0.66), miR-29c + miR-429 + CEA (0.83) for stage I NSCLC vs. HC	[102]
112 NSCLC, 20 HS, 23 pneumonia, 21 gastric cancer, 40 HC	**Serum** Storage: −80 °C	Targets: miR-216, miR-210, miR-183, miR-182 Method: RT-qPCR Normalization: snRNA U6		↑ miR-210, miR-183, miR-182, ↓ miR-126 in NSCLC vs. HC miR-182 (0.73-63%-80%; 0.78-68%-85%), miR-183 (0.63-41%-83%; 0.64-41%-83%), miR-210 (0.79-61%-93%; 0.65-36%-100%), miR-126 (0.79-61%-93%; 0.85-62%-98%), miR-182 + miR-183 + miR-210 + miR-126 + CEA (0.97-81%-100%; 0.98-89%-93%) for all stages and early stage NSCLC vs. HC, respectively	[103]
Discovery: 44 NSCLC, 22 age-/ sex-matched HC	**Serum** Processing within 4 h Centrifugation: 4000 rpm, RT, 10 min Storage: −80 °C	381 miRNAs RT-qPCR Normalization: spiked-in ath-miR-159a, miR-484	Targets: miR-660, miR-652, miR-194 Method: RT-qPCR Normalization: spiked-in ath-miR-159a, miR-484	↑ miR-660 (0.71-0.74), miR-652 (0.82), miR-194 in NSCLC vs. HC ↓ miR-652, miR-194 after tumor removal vs. preoperative samples miR-652 + miR-660 (0.86–0.90), miR-652 + miR-660 + Cyfra21-1 (0.94–0.95; 87–85%; 93%) for NSCLC vs. HC	[104]
50 NSCLC, 60 HC	**Serum**	Target: miR-21 Method: RT-qPCR Normalization: snRNA U6		↑ miR-21 (0.87), CEA, NSE, CYFRA21-1 (0.84) in NSCLC vs. HC ↓ miR21, CYFRA21-1 in stage I-II NSCLC vs. stage III-IV CYFRA21-1 + miRNA-21 (0.91) for early stage NSCLC vs. HC	[105]
59 NSCLC, 59 BPD	**EDTA-plasma** Centrifugation: 1000 *g*, 10 min, 4 °C Storage: −80 °C	Targets: miR-486, miR-451, miR-210, miR-205, miR-155, miR-150, miR-126, miR-34a, miR-26b, miR-21 Method: RT-qPCR Normalization: miR-16		↑ miR-486 (0.85-83%-78%), miR-210 (0.75-75%-75%) in NSCLC vs. BPD CYFRA21-1 + miR-486 + miR-210 (0.92-85%-73%) for NSCLC vs. BPD	[106]
Discovery: 70 NSCLC, 22 age-/sex-matched HC Validation: 84 NSCLC, 23 age-/sex-matched HC	**Serum** Processing within 1 h Storage: −80 °C	TaqMan OpenArray Human microRNA panel	Targets: miR-301, miR-200b, miR-193b, miR-141 Method: RT-qPCR Normalization: snRNA U6	↑ miR-301, miR-200b, miR-193b, miR-141 in NSCLC vs. HC miR-301 + miR-200b + miR-193b + miR-141 (0.99-0.99) for all stages NSCLC vs. HC miR-301 + miR-200b + miR-193b + miR-141 (0.99-0.99) for stage I NSCLC vs. HC	[107]
Training: 24 HC,24 early stage NSCLC Validation: 94 early stage NSCLC, 48 late stage NSCLC, 111 HC	**Serum** Processing within 4 h Centrifugation: 3000 rpm, 4 °C, 15 min Storage: −80 °C	Targets: miR-486-5p, miR-205, miR-200b, miR-155, miR-126, miR-125a-5p, miR-25, miR-21, miR-2 Method: RT-qPCR Normalization: cel-miR-39		↓ miR-126, miR-125a-5p, miR-25 in early stage NSCLC vs. HC miR-126 + miR-125a-5p + miR-25 (0.93-0.94; 88-88%; 83-88%) for early stage NSCLC vs. HC	[108]
36 NSCLC, 20 BPD, 10 HC	**Serum** Centrifugation: 2000 *g*, 20 min, 4 °C Storage: −80 °C	Targets: miR-152, miR-148a, miR-148b Method: RT-qPCR Normalization: snRNA U6		↓ miR-152 (0.77-72%-90%), miR-148a (0.78-78%-80), miR-148b (0.73-69%-80%) in NSCLC vs. BPD and HC miR-148a + miR-148b + miR-152 (0.79-72%-90%) for NSCLC vs. BPD	[109]
300 NSCLC, 152 HC	**Serum** Centrifugation: (i) 3000 rpm, 10 min; (ii) 12,000 rpm, 5 min Storage: −80 °C	Targets: miR-152, miR-148a, miR-148b, miR-21 Method: RT-qPCR Normalization: snRNA U6		↓ miR-152 (0.82-75%-77%), miR-148a (0.90-85%-83%), miR-148b (0.90-83%-83%), ↑ miR-21 in NSCLC vs. HC miR-152 + miR-148a + miR-148b + miR-21 (0.81-69%-71%, 0.98-96%-91%) for NSCLC vs. HC	[110]
90 LC (60 NSCLC, 30 SCLC), 85 HC	**K2-EDTA-plasma** left at RT for 30 min Centrifugation: 1200 *g*, 12 min Storage: −80 °C	Targets: miR-3662, miR-944 Method: RT-qPCR Normalization: snRNA U6		↑ miR-944 (0.91-82%-91%), miR-3662 (0.90-72%-94%) in both LC types vs. HC miR-944 + miR-3662 (0.91-82%-92%) for LC vs. HC	[111]
56 NSCLC, 100 HC	**K2-EDTA-plasma**	Targets: pri-miRNA-3662, pri-miRNA-944 Method: RT-qPCR Normalization: GAPDH		↑ pri-miR-944, pri-miR-3662 in NSCLC vs. HC pri-miR-944 + pri-miR-3662 (0.90-76%-82%) for NSCLC vs. HC	[112]
Training: 36 NSCLC, 36 HC Validation: 120 NSCLC, 71 HC	**Serum**, left 1 h at RT Centrifugation: first at low speed and further at 16,000 g, 10 min, 4 °C Storage: −80 °C	Targets: miR-1254, miR-574-5p, miR-486, miR-485-5p, miR-197, miR-182, miR-155, miR-125b, miR-30d, miR-24, miR-21, MALAT1 Method: RT–qPCR Normalization: miR-16	Targets: miR-1254, miR-574-5p, miR-485-5p, MALAT1 Method: RT–qPCR Normalization: miR-16	↓ miR-1254, miR-574-5p, miR-485-5p, MALAT1 in NSCLC vs. HC miR-1254 + miR-574-5p + miR-485-5p + MALAT1 (0.84-0.86;86-93%;73-81%) for NSCLC vs. HC	[113]
Discovery: 38 NSCLC, 16 COPD, 16 HC I validation: 16 NSCLC, 8 COPD, 6 HC II validation: 100 NSCLC, 50 non-cancerous controls	**Serum**	754 miRNAs TaqMan Low Density Arrays	I validation Targets: miR-205, miR-200b, miR-125b, miR-34b, miR-429, miR-203, miR-186, miR-1180, miR-601, miR-378 II validation Targets: miR-205, miR-200b, miR-125b, miR-34b, miR-429, miR-203, miR-1180 Normalization: average of miR-220, miR-19b, snRNA U6 Method: RT-qPCR	↑ miR-429, miR-205, miR-203, miR-200b, miR-125b, miR-34b in NSCLC vs. COPD and HC miR-429 + miR-205 + miR-203 + miR-200b + miR-125b + miR-34b (0.89-88%-71%) for all stage NSCLC vs. non-cancerous controls miR-429 + miR-205 + miR-203 + miR-200b + miR-125b + miR-34b (0.88-85%-74%) for early stage NSCLC vs. non-cancerous controls	[115]
90 LC, 85 HC	**K2-EDTA-plasma** Centrifugation: 3000 rpm, 10 min Storage: −80 °C	Targets: miR-4478, miR-4316, miR-506, miR-448 Method: RT-qPCR Normalization: snRNA U6		↑miR-4478, miRNA-448, ↓ miR-506 in LC vs. HC miR-448 (0.89-91%-74.5%), miR-4478 (0.80-71.4%-73.7%), miR-448 + miR-4478 (0.90-89.4%-79%) for LC vs. HC miR-448 (0.90-85%-77%), miR-4478 (0.82-75%-68%), miR-448 + miR-4478 (0.90-90%-76%) for early stage NSCLC vs. HC	[116]
Screening: 5 NSCLC, 5 HC Selection: 82 NSCLC, 91 HC, 43 benign lesions Testing: 74 NSCLC, 82 HC	**K2-EDTA-plasma** Centrifugation: (i) 1500 rpm, 30 min; (ii) 3000 rpm, 5 min; (iii) 4500 rpm, 5 min; all at 4 °C torage: −80 °C	748 miRNAs TaqMan Array MiRNA Cards Normalization: cel-miR-39	20 miRNAs Method: RT-qPCR Normalization: cel-miR-39	↑ miR-628-3p, miR-339-3p, miR-425-3p, ↓ miR-532 in NSCLC vs. HC miR-628-3p + miR-532 + miR-425-3p + miR-339-3p (0.98-90%-99%), miR-628-3p + miR-532 + miR-425-3p (0.97; 92–97%; 95–98%) for early stage NSCLC vs. HC	[117]
Training: 94 NSCLC, 58 HC Validation: 70 NSCLC, 54 HC	**Serum** Centrifugation: 15 min, 3000 rpm, 4 °C Storage: −80 °C	Targets: miR-664, miR-579, miR-506, miR-146-3p, miR-106-5p, miR-92-3p, miR-28-3p, miR-20a-5p, miR-19-3p, miR-17b-5p, miR-16-5p, miR-15b-5p Method: RT-qPCR	Targets: miR-92-3p, miR-28-3p, miR-20a-5p, miR-19-3p, miR-17b-5p, miR-16-5p, miR-15b-5p Method: fluorescence quantum dots liquid bead array	↓ miR-16-5p, miR-17b-5p, miR-19-3p, miR-20a-5p, miR-92-3p, ↑ miR-15b-5p in NSCLC vs. HC miR-20a-5p + miR-16-5p + miR-15b-5p (0.93; 86-94%; 91-94%) for NSCLC vs. HC	[118]
75 LC, 50 HC	**EDTA-plasma** Processing within 4 h entrifugation: (i) 400 g, 20 min; (ii) 800 *g*, 20 min Storage: −80 °C	Targets: miR-205, miR-126, miR-125b, miR-183, miR-25, miR-19b, miR-21 Method: RT-qPCRNormalization: miR-16		↑ miR-21, miR-19b, ↓ miR-183, miR-25 in LC vs. HC miR-19b (0.83-75%-91%) for SCC vs. HC miR-183 (0.92-100%-81%) for ADC vs. HC miR-183 + miR-19b (0.99-95%-95%) for LC vs. HC	[119]
126 NSCLC, 118 HS	**Plasma** Processing within 2 h Centrifugation: 1300 *g*, 10 min, 4 °C Storage: −80 °C	Targets: miR-210, miR-205-5p, miR-145, miR-126 Method: RT-qPCR Normalization: snRNA U6		miR-126 + miR-145 + miR-210 + miR-205-5p (0.96-92%-96%) for NSCLC vs. HS	[120]
Screening: 10 LC, 10 HC Validation: 40 LC, 40 BPD, 40 HC	**Serum** Centrifugation: (i) 2000 rpm, 10 min; (ii) 12,000 rpm, 3 min, RT Storage: −80 °C	355 miRNAs Microarray	21 miRNAs Method: RT–qPCR Normalization: geometric mean–based global normalization	↑ miR-423-3p, miR-199a-3p, miR-151-5p, miR-148b, miR-126-3p, miR-221, miR-27b, miR-26a, miR-23b, let-7f in LC vs. benign pulmonary disease and HC miR-423-3p + miR-221 + miR-199a-3p + miR-151-5p + miR-148b + miR-126-3p + miR-27b + miR-26a + miR-23b + let-7f (0.89), miR-423-3p + miR-221 + miR-148b + miR-23b (0.89) for LC vs. HC	[121]
Training: 20 early stage NSCLC, 20 age-/sex-matched HC Validation: 109 early stage NSCLC, 63 HC	**EDTA-Plasma** Centrifugation: (i) 1200 rpm, 10 min; (ii) 1000 rpm, 10 min, 4 °C	Targets: miR-383, miR-223, miR-221, miR-210, miR-126, miR-20a, miR-145, miR-30d, miR-25, miR-21 Method: RT–qPCR Normalization: miR-16		↑ miR-223 (0.81-70%-84%), miR-145 (0.89-81%-89%), miR-21 (0.84-78%-86%), miR-20a (0.89-80%-88%) in early stage NSCLC vs. HC ↓ miR-223, miR-145, miR-21, miR-20a post-operative vs. pre-operative miR-223 + miR-145 + miR-21 + miR-20a (0.90-82%-90%) for early stage NSCLC vs. HC	[122]
Training: 48 NSCLC, 20 benign pulmonary nodule, 60 pulmonary inflammation Validation: 70 NSCLC, 22 SPNs, 101 pulmonary inflammation	**Serum** left at RT for 20 min Centrifugation: 15 min, 3000 rpm, 4 °C Storage: −80 °C	Several ratios of selected miRNAs Method: RT-qPCR Normalization: the ratio of miRNAs was uses in order to eliminate the experimental differences		↑ miR-15b-5p/miR-146b-3p (0.73), miR-20a-5p/miR-146b-3p (0.85), miR-19a-3p/miR-146b-3p (0.84), miR-92a-3p/miR-146b-3p (0.74), miR-16-5p/miR-146b-3p (0.83) in NSCLC vs. SPNs ↑ miR-15b-5p/miR-146b-3p (0.72), miR-92a-3p/miR-15b-5p, miR-16-5p/miR-15b-5p, miR-20a-5p/miR-146b-3p (0.77), miR-19a-3p/miR-146b-3p, miR-92a-3p/miR-146b-3p, miR-16-5p/miR-146b-3p, miR-20a-5p/miR-17-5p (0.69), miR-92a-3p/miR-20a-5p (0.54), miR-16-5p/miR-20a-5p (0.70), miR-16-5p/miR-19a-3p (0.80), miR-92a-3p/miR-17-5p, miR-16-5p/miR-17-5p in NSCLC vs. pulmonary inflammation diseases	[123]
Discovery: 20 LC, 10 HC Validation: 30 LC, 20 HC, 10 endobronchitis	**EDTA-Plasma** Processing within 4 h Centrifugation: (i) 400 *g*, 20 min; (ii) 800 *g*, 20 min Storage: −80 °C	179 miRNAs iRCURY LNA miRNA qPCR Serum/Plasma panel Normalization: global mean	19 miRNAs ratios Method: RT-qPCR Normalization: Ratio-based normalization	10 miRNA ratios panel: miR-22-3p/miR-210, miR-107/miR-222-3p, miR-19b-3p/miR-484, miR-150-5p/miR-144-5p, miR-484/miR-374a-5p, miR-484/miR-338-3p, miR-484/miR-324-5p, let-7i-5p/miR-222-3p, miR-22-5p/miR-324-5p, miR-374a-5p/miR-133b (0.98) for LC vs. HC	[124]
56 NSCLC, 26 cancer-free (HC or cataract)	VP-AS109K**-Serum**, left at RT for 30 min Centrifugation: (i) 1500 *g*, 10 min, 4 °C; (ii) 20,000 *g*, 10 min, 4 °C Storage: −80 °C	Targets: miR-223-3p, miR-145-5p, miR-21-5p, miR-20a-5p Method: RT–qPCR Normalization: cel-miR-39		↓ miR-145 in all stages NSCLC vs. HC ↓ miR-20a in stage I-II and III NSCLC vs. HC ↑ miR-21 in stage IV NSCLC vs. stage I-II ↑ miR-223 in stage IV NSCLC vs. HC miR-145 (0.83-71%-89%), miR-20a (0.658-59%-73%), miR-223 (0.69-82%-52%), miR-145 + miR-223 (0.89-86%-80%), miR-145 + miR-20a (0.82), miR-20a + miR-223 (0.79), miR-223 + miR-145 + miR-20a (0.88) for NSCLC vs. HC	[125]
437 NSCLC, 415 HC	**EDTA-plasma** Centrifugation: 3000 *g*, 10 min, 4 °C Storage: −80 °C	486 miRNAs Method: RT-qPCR Normalization: cel-miR-54		↑ miR-26a-5p, miR-126-5p, miR-139-5p, miR-3135b, ↓ miR-451a in ADC vs. HC ↑ miR-139-5p, miR-151a-3p, miR-151a-5p, miR-151b, miR-550a-3p, miR-3135b in SCC vs. HC ↑ miR-139-5p, miR-152-3p in NSCLC patients vs. HC miR-3135b + miR-451a + miR-139-5p + miR-126-5p + miR-26a-5p (0.89-98%-79%) for ADC vs. HC miR-3135b + miR-550a-3p + miR-151a-3p + miR-151a-5p + miR-151b + miR-139-5p (0.97-100%-76%) for SCC vs. HC miR-152-3p + miR-139-5p (0.80-88%-95%) for NSCLC vs. HC	[126]
20 HC, 30 stage IV NSCLC, 32 stage I-II NSCLC	**EDTA-serum^§^** Processing within 4 h Centrifugation: 2000 rpm, 20 min, Storage: −80 °C	Targets: miR-222, miR-221, miR-200, miR-183, miR-126, miR-125a-5p Method: RT-qPCR Normalization: snRNA U6		↓ miR-222, miR-183, miR-126 in stage IV NSCLC vs. HC ↓ miR-183, miR-126 in stage IV NSCLC vs. stage I-II NSCLC	[129]
Training: 14 LC with lymphatic metastasis, 12 without lymphatic metastasis, 5 BPD Validation: 40 LC with lymphatic metastasis and 11 without lymphatic metastasis	**Plasma**	Targets: miR-422a, miR-378, miR-375, miR-205, miR-200b, miR-183 Method: RT-qPCR Normalization: miR-16		↑ miR-422a (0.79) in LC with lymphatic metastasis vs. LC without lymphatic metastasis	[130]
Training set, Trial INT-IEO, 19 subjects Validation set, MILD trial, 22 subjects	**EDTA-plasma** Centrifugation: 1258 *g*, 4 °C Storage: biological bank	Megaplex Pools Protocol on microfluidic card type A	Multiplex Pools Protocol	A 16 miRNAs ratios plasma signature for LC risk (0.85-80%-90%) A 16 miRNAs ratios plasma signature for LC vs. HC (0.88-75%-10%) A 10 miRNAs ratios plasma signature for poor prognosis (80%-100%) A 10 miRNAs ratios plasma signature for aggressive LC (88%-100%)	[127]
Screening: 40 NSCLC (recurrence and non-recurrence) Validation: 114 NSCLC (recurrence vs. non-recurrence)	**Serum**	664 miRNAs TaqMan Array Human miRNA A and B Cards v2.0 Normalization: median normalization	Targets: miR-618, miR-517a, miR-486-5p, miR-380*, miR-338-3p, miR-331-3p, miR-183*, miR-142-3p, miR-29b, miR-20b Method: RT-qPCR Normalization: cel-miR-39, cel-miR-54	↑ miR-142-3p, miR-29b in the recurrence vs. the non-recurrence group miR-142-3p (0.64), miR-142-3p + tumor stage (0.78) for high risk recurrence vs. low risk recurrence	[132]
Training: 34 NSCLC Testing: 34 NSCLC	**Serum**	Targets: miR-499a-5p, miR-486-5p, miR-221-3p, miR-30d-5p, miR-1, let-7a-5p Method: RT-qPCR Normalization: cel-miR-39	Targets: miR-486, miR-221, miR-30d, miR-1 Method: RT-qPCR Normalization: cel-miR-39	miR-486 + miR-221 + miR-30d + miR-1 (0.88-94.1%-70.6%) for high risk recurrence group vs. low risk group	[133]
260 NSCLC, 260 age-/sex-matched HC	**Serum** Processing within 2–5 h Centrifugation: 4000 rpm, 10 min Storage: −80 °C	Targets: miR-155, miR-125b, miR-34a, miR-10b Method: RT-qPCR Normalization: miR-16		↑ miR-125b in chemotherapy non-responsive vs. responsive	[134]
Profiling cohort: 5 NSCLC stage IIIA Validation cohort: 21 NSCLC stage IIIA	**K2-EDTA-plasma** Centrifugation: 2000 *g*, 15 min, 4 °C Storage: −80 °C	752 miRNAs Exiqon miRCURY LNA Universal RT microRNA PCR Human Panel I + II arrays	Targets: miR-342-3p, miR-320a, miR-191-5p, miR-150-5p, miR-142-3p, miR-125b-5p, miR-101-5p, miR-30d-5p, miR-29a-3p, and miR-15b-5p Method: RT-qPCR Normalization: miR-324, miR-126, miR-16-2-3p, miR-let-7d	↓ miR-29a-3p, miR-150-5p with increasing radio therapy dose	[135]
23 NSCLC	**EDTA-plasma** Processing within 4 h Centrifugation: (i) 20 min, 400 *g*; (ii) 20 min, 1200 *g* Storage: −80 °C	Targets: miR-205, and miR-126, miR-125b, miR-25, miR-19b Method: RT-qPCR Normalization: miR-16		↓ miR-19b, ↑ miR-125b in chemotherapy responsive patients	[136]
54 NSCLC	**Plasma** Centrifugation: 3000 rpm, 10 min, 4 °C Storage: −80 °C	miRNA profile via bioinformatics	Targets: miR-1297, miR-613, miR-520a-3p, miR-520b, miR-520d-3p, miR-520e, miR-495-3p, miR-372-3p, miR-328-3p, miR-302e, miR-206, miR-203a-3p, miR-193a-3p, miR-153-3p, miR-137, miR-98-5p, miR-34c-5p miR-1-3p, let-7c-5p Method: RT–qPCR Normalization: cel-miR-39	↑ miR-613 (80.0%-69.2%), miR-495-3p (93.3%-48.7%), miR-302e (66.7%-92.3%), miR-98-5p (80.0%-76.9%) in radiotherapy sensitive NSCLC vs. non-sensitive	[137]
58 advanced LC treated with nivolumab	**Serum** Processing within 1 h Centrifugation: 2410 rpm, 12 min Storage: −80 °C	NGS	Targets: miR-548j-5p, miR-495-3p, miR-494-3p, miR-493-5p, miR-411-3p, miR-215-5p, miR-93-3p Method: RT-qPCR Normalization: miR-93-5p, miR-222-3p	miR-548j-5p + miR-495-3p + miR-494-3p + miR-493-5p + miR-411-3p + miR-215-5p + miR-93-3p (0.81-71%-90%) for overall survival >6 months in LC	[140]
100 NSCLC, 100 HC	**EDTA-plasma** Processing within 2 h Centrifugation: 2000 *g*, 10 min, RT Storage: −80 °C	754 miRNAs TaqMan Human MicroRNA Arrays A + B cards Normalization: quantile normalization		24-plasma miRNA panel: miR-1267, miR-1243, miR-720, miR-661, miR-642, miR-566, miR-543, miR-520f, miR-519a, miR-517b, miR-485-3p, miR-450b-5p, miR-411, miR-340#, miR-218, miR-206, miR-203, miR-200c, miR-193a-5p, miR-182, miR-155, miR-122, let-7, let-7b (0.92-83%-80%) for early stage NSCLC vs. HC 24-plasma miRNA panel + age, sex, smoking habits (0.94-83%-84%) for early stage NSCLC vs. HC	[145]

ADC: adenocarcinoma; BPD: benign pulmonary disease; COPD: chronic obstructive pulmonary disease; HC: healthy controls; HS: healthy smokers; LC: lung cancer; NGS: next-generation sequencing; NSCLC: non-small cell lung cancer; RT: room temperature; RT-qPCR: real-time quantitative polymerase chain reaction; SCC: squamous cell carcinoma; SCLC: small-cell lung cancer; SPNs: benign solitary pulmonary nodules. §: inconsistent report: serum is reported as obtained from ethylendiaminotetraacetate (EDTA)-anticoagulated tubes; ↑: increased levels of the considered miRNA; ↓: decreased levels of the considered miRNA.

**Table 2 jcm-08-01661-t002:** Circulating miRNAs associated with breast cancer (BC).

Cohort	Pre-Analytical Variables	Analytical Method		Identified miRNAs (AUC-Sensitivity%-Specificity%)	Study
		Screening	Validation		
50 BC (25 CA, 25 AA), 20 matched HC (10 CA, 10 AA)	**Plasma** Centrifugation: 3000 rpm, 10 min, RT Storage: phased liquid nitrogen	1145 miRNAs Illumina Plasma V2 MicroRNA expression array	Targets: let-7d*, miR-425* for AA; let-7c, miR-589 for CA Method: RT-qPCRNormalization: miR-16	↓ let-7c (0.78-0.84), ↑ miR-589 (0.62-0.85) in CA BC vs. CA HC ↓ let-7d* (0.73-0.99), ↑ miR-425* (0.79-0.83) in AA BC vs. AA HC	[147]
Screening: 48 early stage BC, 57 age-matched HC Validation: 24 BC, 24 age-matched HC	**EDTA-whole blood** Storage: −20 °C	1100 miRNAs GeniomH Biochip microarray Normalization: variance stabilising normalisation	Targets: miR-718, miR-202 Method: RT-qPCR Normalization: miR-16	↑miR-202 in BC vs. HC	[148]
Discovery: 39 BC, 10 HC Validation: 98 BC, 25 HC	**Clot activator gel free serum**, kept on ice for 15–30 min Centrifugation: 15 min, 3000 rpm, 4 °C Storage: −80 °C	754 miRNAs TaqMan Array Human MicroRNA Cards A and B v3.0 Normalization: global mean, miR-16	Targets: miR-186, miR-484, miR-29a, miR-425-5p, miR-454, miR-574-3p, miR-140-3p, miR-222, let-7b, miR-483-5p Method: RT-qPCR Normalization: global mean, miR-16	↑ miR-484 in BC vs. HC	[149]
Discovery: 83 BC, 26 HC Validation: 114 BC, 116 HC	**EDTA-plasma** Processing within 1 h Centrifugation: (i) 3000 *g*, 20 min, 10 °C, (ii) 15,500 *g*, 10 min, 10 °C Storage: −80 °C	1919 miRNAs NA-based microarray	Targets: miR-3656, miR-1273g-3p, miR-638, miR-505-5p, miR-200a-3p, miR-142-3p, miR-141-3p, miR-125b-5p, miR-96-5p, miR-21-5p, miR-183-3p Method: RT-qPCR Normalization: miR-103a-3p	↑ miR-505-5p (0.72-75%-60%), miR-125b-5p, miR-96-5p (0.72-73%-66%), miR-21-5p in BC vs. HC	[150]
46 BC, 14 HC	**EDTA-plasma and serum** Centrifugation: 2000 rpm, 15 min, RT Storage: −80 °C	Agilent Human SurePrint G3 8 × 60 k v21 miRNA microarray chip	Targets: miR-4281, miR-4270, miR-3141, miR-1290, miR-1225-5p, miR-1207-5p, miR-1202, miR-642b-3p, miR-188-5p Method: RT–qPCR	↑ miR-4270, miR-1290, miR-1225-5p, miR-1207, miR-1202 in early stage BC vs. advanced stage BC	[151]
Discovery: 6 BC, 6 age-matched HC Validation: 15 BC, 13 age-matched HC	**PAXgene-whole blood** Storage: −80 °C	3100 miRNAs miRCURY LNA Array v.18.0	Targets: miR-4717-3p, miR-942-5p, miR-374b-5p, miR-331-3p, miR-204-5p, miR-182-5p, miR-144-5p, miR-96-5p, miR-30b-5p Method: RT-qPCR Normalization: miR-16-5p	↑ miR-942-5p (0.81-67%-100%), miR-374b-5 (0.83-87%-69%), miR-182-5p (0.76-53%-92%), miR-96-5p (0.77-53%-100%), miR-30b-5p (0.93-80%-100%) in BC vs. HC	[152]
94 BC, 100 BBC, 100 age-matched HC	**Serum** Storage: −80 °C	Affymetrix miRNA profiling array	Targets: miR-8075, miR-6821-5p, miR-5001-5p, miR-4487, miR-3613-5p, miR-1915-3p, miR-1825, miR-638, miR-455-3p Method: RT-qPCR Normalization: absolute concentration	↑ miR-1915-3p in BC and BBC vs. HC ↓ miR-455-3p (0.79) in BC vs. BBC and HC ↑ miR-1915-3p (0.93 and 0.89), ↓ miR-455-3p (0.88 and 0.80) in invasive BC and lymph node mBC vs. in situ BC and lymph node non-mBC group, respectively	[153]
148 BC, 44 age-matched HC	**K3EDTA-whole blood and plasma, serum** left to clot 30 min at RT Centrifugation: 2000 rpm, 4 °C, 10 min Storage: −20 °C plasma and serum, 4 °C whole blood)	Targets: miR-195, miR-155, miR-145, miR-21, miR-16, miR-10b, let-7a Method: RT-qPCR Normalization: miR-16		↑ miR-195, let-7a in BC vs. HC	[154]
68 BC, 40 age-matched HC	**Serum** Storage: −80 °C	Targets: miR-335, miR-199a, miR-155, miR-126, miR-106a, miR-21, miR-16 Method: RT-qPCR Normalization: miR-16		↑ miR-155, miR-106a, miR-21, ↓ miR-335, miR-199a, miR-126 in BC vs. HC	[155]
103 BC, 44 age-matched HC	**Clot activator gel-Serum**, left 30 min-2 h at RT Centrifugation: (i) 1300 *g*, 15 min, (ii) 10,000 *g*, 10 min, 4 °C Storage: −80 °C	Target: miR-155 Method: RT-qPCR Normalization: cel-miR-39		↑ miR-155 (0.80-65%-81%) in BC vs. HC	[156]
Discovery: 40 BC, 10 HC Validation: 62 BC, 10 HC	**Red tiger-top gel-serum** Processing within 2-5 h Storage: −80 °C	Target: miR-21 Method: RT-qPCR Normalization: miR-16		↑ miR-21 (0.72-75%-67%) in BC vs. HC ↑ miR-21 (0.83-86%-70%) in stage IV BC vs. early stage BC	[157]
20 BC, 20 age-matched HC	**Serum** Centrifugation: (i) 1600 rpm, 5 min; (ii) 12,000 rpm, 10 min	NGS	Targets: miR-29a, miR-23a, miR-23b, miR-192, miR-21 Method: RT-qPCR Normalization: snRNA U6	↑ miR-29a, miR-21 in BC vs. HC	[158]
Screening: 13 BC, 10 HC Validation: 50 BC, 50 age-matched HC, 20 BBC, 20 OVC	**Serum**	NGS	Targets: miR-222, miR-103, miR-29a, miR-25, miR-24, miR-23a, miR-23b Method: RT-qPCR Normalization: snRNA U6	↑miR-222 (0.67-74%-60%) in BC vs. HC	[159]
100 BC, 20 age-matched HC	**Serum**	Targets: miR-17, miR-21, miR-665, miR-625, miR-558, miR-185, miR-125b, miR-106b, miR-93, miR-92a Method: RT-qPCR Normalization: miR-16		↑ miR-21 (0.93), ↓ miR-92a (0.92) in BC vs. HC	[160]
152 BC, 75 age-matched HC	**EDTA-serum**^§^ Processing within 4 h Centrifugation: (i) 1600 rpm, 4 °C, 5 min; (ii) 12,000 rpm, 15 min, 4 °C Storage: −70 °C	Target: miR-181a Method: RT-qPCR Normalization: miR-16		↓ miR-181 (0.75-71%-60%) in BC vs. HC	[161]
100 BC, 64 age-matched HC	**EDTA-plasma** Centrifugation: 3000 rpm, 10 min, RT Storage: −80 °C	Target: miR-30a Method: RT-qPCR Normalization: miR-16		↓ miR-30a (0.76-74%-66%) in BC vs. HC	[162]
210 BC, 102 age-matched HC	**EDTA-serum**^§^ Processing within 4 h Centrifugation: (i) 1600 rpm, 4 °C, 5 min; (ii) 12,000 rpm, 15 min, 4 °C Storage: −70 °C	Target: miR-195 Method: RT-qPCR Normalization: miR-16		↓ miR-195 (0.86-69%-89%) in BC vs. HC	[163]
32 BC, 22 age-matched HC	**Serum** Centrifugation: 30 min, 2650 *g* Storage: −80 °C	Target: miR-182 Method: RT-qPCR		↑ miR-182 in BC vs. HC	[164]
57 BC, 20 HC	**EDTA-whole blood** (first 5 mL discarded to avoid skin contamination) Storage: −20 °C	Targets: miR-200c, miR-141 Method: RT-qPCR Normalization: 5S rRNA, snRNA U6		↓ miR-200c (0.79-90%-70.2% for all stages, 0.82-90%-75% for early stage) in BC vs. HC	[165]
Discovery: 28 BC, 27 age-matched HC Validation: 59 BC, 35 age-matched HC	**Red stopper clot-serum** Processing within 1 h Centrifugation: 1000 *g*, 10 min, RT Storage: −80 °C	Targets: miR-10b-5p, miR-145-5p, miR-148b-3p, miR-425-5p, miR-652-3p Method: ddPCR Normalization: cel-miR-39-3p		↓ miR-652-3p (0.75), miR-425-5p, miR-148b-3p (0.70), miR-145-5p, ↑ miR-10b in BC vs. HC	[166]
100 BC, 40 age-matched HC	**Serum**	Targets: miR-1246, miR-598-3p, miR-382-3p, miR-184 Method: RT-qPCR Normalization: cel-miR-39		↑ miR-1246 (0.90-93%-75%), miR-382-3p (0.74-52%-93%), ↓ miR-598-3p (0.94-95%-85%), miR-184 (0.74-88%-71%) in BC vs. HC	[167]
24 HC, 24 BPT, 11 BAH, 85 early stage BC	**Clot activator gel-free serum**, left on ice for 15–30 min Centrifugation: 5 min, 3000 rpm, 4 °C Storage: −80 °C	88 miRNAs Method: RT-qPCR Normalization: miR-16 and miR-92a	Targets: miR-24, miR-103a Method: RT-qPCR Normalization: miR-16, miR-92a	↓ miR-24 (0.82-0.90 and 0.90-0.96), miR-103a (0.72-0.81 and 0.71-0.82) in BAH and early stage BC vs. HC	[168]
86 BC, 26 age-matched HC	**Plasma** Centrifugation: 3000 rpm, 10 min	Target: miR-520g Method: RT-qPCR Normalization: snRNA U6		↑ miR-520g in BC vs. HC	[169]
40 BC, 40 age-matched HC	**Serum** Storage: −80 °C	Target: miR-140-3p Method: RT-qPCR Normalization: miR-16		↑ miR-140-3p (0.66-50%-60%) for BC vs. HC ↑ miR-140-3p (0.69-73%-50%) for premenopausal BC vs. premenopausal HC	[170]
Discovery: 24 ER+ mBC, 24 ER+ non-mBC, 24 age-matched HC Validation: 60 ER+ BC, 51 age-matched HC	**Serum** Processing within 1 h Storage: −80 °C	174 miRNAs Exiqons LNA-based quantitative PCR (qRT-PCR) Plasma/serum Focus panel Normalization: global mean	Targets: miR-425, miR-365, miR-145, miR-143, miR-139-5p, miR-133a, miR-107, miR-18a, miR-15a Method: RT-qPCR Normalization: miR-10b, miR-30a	↑ miR-425, miR-107, miR-18a, miR-15a, ↓ miR-365, miR-145, miR-143, miR-139-5p, miR-133a in ER+ BC vs. HC miR-425 + miR-365 + miR-145 + miR-143 + miR-139-5p + miR-133a + miR-107 + miR-18a + miR-15a (0.67-73%-41%) for ER+ BC vs. HC	[171]
36 BC, 80 HC	**Serum** Processing within 1 h Centrifugation: 4 °C, 2000 *g*, 10 min Storage: −80 °C	Targets: miR-425, miR-365, miR-145, miR-143, miR-139-5p, miR-133a, miR-107, miR-18a, miR-15a Method: RT-qPCR Normalization: miR-10b-5p		↑ miR-425, miR-107, miR-18a, miR-15a, ↓ miR-364, miR-145, miR-143, miR-139-5p, miR-133a in ER+ BC vs. HC miR-425 + miR-365 + miR-145 + miR-143 + miR-139-5p + miR-133a + miR-107 + miR-18a + miR-15a (0.61) for ER+ BC vs. HC	[172]
48 high risk for BC	**Serum** Processing within 1 h Centrifugation: 1811 *g*, 10 min Storage: −80 °C	2578 miRNAs Affymetrix GeneChip v4.0 miRNA array Normalization: the average of 95 anti-genomic probesets subtracted from each RNA expression probeset in R v3.x		↓ miR-7855-5p, miR-4529-3p, miR-4423-3p, miR-3124-5p, miR-1184, ↑ miR-4446-3p in BC vs. cancer-free women miR-7855-5p + miR-4446-3p + miR-4529-3p + miR-4423-3p + miR-3124-5p + miR-1184 (0.90) in BC vs. cancer-free women	[173]
Discovery: 5 BC, 5 age-matched HC Training: 15 BC, 15 age-matched HC Validation: 240 BC, 150 age-matched HC, 95 with other cancers	**Plasma**	TaqMan Array Human Micro-RNA Panels A and B	Training targets: miR-451, miR-210, miR-200c, miR-191, miR-150, miR-145, miR-27a, miR-21, miR-16 Validation targets: miR-451, miR-145, miR-21, miR-16 Method: RT-qPCR Normalization: snRNA U6	↑ miR-451, miR-21, miR-16, ↓ miR-145 in BC vs. HC miR-145 + miR-451 (0.91-0.96; 83-90%; 89-93%) for BC vs. HC	[174]
1st cohort: 127 BC, 80 HC 2nd cohort: 30 benign BC, 120 BC, 60 HC	**EDTA-plasma** Processing within 2 h Centrifugation: (i) 1300 *g*, 20 min, 10 °C; (ii) 15,500 *g*, 10 min, 10 °C Storage: −80 °C	Targets: miR-801, miR-652, miR-409-3p, miR-376a, miR-376c, miR-148b, miR-127-3p Method: RT-qPCR Normalization: cel-miR-39		↑ miR-801, miR-652, miR-409-3p, miR-376a, miR-376c, miR-148b, miR-127-3p in BC vs. HC ↑ miR-801, miR-652, miR-148b in benign BC vs. HC ↑ miR-801, miR-652, miR-409-3p, miR-148b, miR-127-3p stage I-II in BC vs. HC miR-801 + miR-652 + miR-409-3p + miR-376a + miR-376c + miR-148b + miR-127-3p (0.81-80%-72%) for BC vs. HC	[175]
127 BC, 80 age-matched HC	**Plasma** Processing within 2 h Centrifugation: (i) 1300 *g*, 20 min, 10 °C; (ii) 15,500 *g*, 10 min, 10 °C Storage: −80 °C	667 miRNAs TaqMan low-density arrays human MicroRNA Cards A v2.1 & B v2.0	Targets: miR-801, miR-571, miR-409-3p, miR-376c, miR-206, miR-148b, miR-139-3p Method: RT-PCR Normalization: miR-16	↑ miR-801, miR-409-3p, miR-376c, miR-148b in BC vs. HC miR-801 + miR-409-3p + miR-148b (0.69-70%-55%) for BC vs. HC	[176]
Screening: 47 BC, 58 age-matched HC Validation: 132 BC, 101 age-matched HC	**Clot activator gel-serum**, allowed to clot for 30 min Centrifugation: 2200 *g* Storage: −80 °C	742 miRNAs NA RT-PCR human microRNA panels	Targets: miR-133a, miR-133b, miR-92a, miR-1 Method: RT-PCR Normalization: miR-103, miR-191	↑ miR-133a, miR-133b, miR-92a, and miR-1 in BC vs. HC miR-133b + miR-92a (0.90), miR-133a + miR-92 (0.91), and miR-92a + miR-1 (0.90) for BC vs. HC	[177]
61 BC and 10 HC	**Serum** Centrifugation: 3000 rpm, 5 min Storage: −70 °C	Targets: miR-382, miR-191, miR-155, miR-145, miR-125b, miR-21, miR-10b Method: RT-qPCR Normalization: 18S RNA		↑ miR-382, miR-191, miR-155, miR-145, miR-125b, miR-21, and miR-10b in BC vs. HC miR-145, miR-155, miR-382 ratio (0.99-98%-100%) for BC vs. HC	[178]
20 benign BC, 60 BC, 29 HC	**Serum** Storage: −80 °C	Targets: miR-666-5p, miR-451, miR-374, miR-148a, miR-30b, miR-27a Method: RT-qPCR Normalization: cel-miR-67		↓ miR-451, miR-148a, miR-30b, miR-27a in BC vs. HC ↓ miR-451, miR-30b, miR-27a in benign BC vs. HC miR-451 + miR-148a + miR-30b + miR-27a (0.95-95%-83%) for BC vs. HC	[179]
Discovery: 52 early stage invasive BC, 35 local BC, 35 age-matched HC Validation: 50 early stage invasive BC, 50 age-matched HC	**EDTA-plasma** Centrifugation: 3000 rpm, 10 min, RT Storage: phased liquid nitrogen	168 miRNAs Serum/Plasma Focus microRNA PCR Panel	Targets: miR-409-3p, miR-148b, miR-133a Method: RT-qPCR Normalization: miR-93	↑ miR-148b, miR-133a in BC vs. HC miR-148b + miR-133a (0.86) for BC vs. HC	[180]
149 BC, 35 BC in remission, 31 mBC, 30 gynecologic tumors	**EDTA-plasma** Processing within 1 h Centrifugation: (i) 10 min, 815 *g*, 4 °C, (ii) 10 min, 2500 *g*, 4 °C Storage: −80 °C	188 miRNAs Method: RT-qPCR Normalization: mean Cq of the most expressed 50 miRNAs		8 miRNAs panel: let-7d, let-7i, miR-148a, miR-107, miR-103, miR-22*, miR-19b, miR-16 (0.81-91%-49%) for BC vs. HC	[181]
Discovery: 74 BC, 1493 HC, 514 other cancers or benign diseases Validation: 1206 BC, 54 benign BC, 1343 HC	**Serum** Storage: −20 °C or −80 °C	2555 miRNAs 3D-Gene Human miRNA Oligo Chip Normalization: miR-149-3p, miR-2861, miR-4463	Targets: miR-1246, miR-1307-3p, miR-4634, miR-6861-5p, miR-6875-5p Method: RT-qPCR Normalization: miR-149-3p	↑ miR-6861-5p, miR-1307-3p, miR-1246, ↓ miR-6875-5p, miR-4634 in BC vs. HC miR-6875-5p + miR-6861-5p + miR-4634 + miR-1307-3p + miR-1246 (0.97-97%-83%) for BC vs. HC	[182]
Discovery: 30 BC, 30 age-matched HC Validation: 128 BC, 77 HC age-matched	**Serum** left to clot 1 h at RT Centrifugation: 10 min, 16,000 *g*, 4 °C Storage: −80 °C	Targets: let-7a, miR-1254, miR-574-5p, miR-205, miR-196a, miR-195, miR-181b, miR-155, miR-21, miR-10b Method: RT-qPCR Normalization: cel-miR-39	Targets: let-7a, miR-574-5p, miR-155 Method: RT-qPCR Normalization: cel-miR-39	↓ let-7a (0.78), ↑ miR-574-5p (0.87), miR-155 (0.82), MALAT1 (0.65) in BC vs. HC miR-574-5p + miR-155 + let-7a + MALAT1 (0.96-0.97; 90-94%; 97-99%) for BC vs. HC	[183]
Training: 24 BC, 24 HC (serum, plasma) Testing: 146 BC, 146 HC (plasma); 150 BC, 148 HC (serum) External validation: 30 BC, 30 HC (plasma and serum)	**EDTA-plasma and clot activator gel-serum** Processing within 6 h Centrifugation plasma: (i) 350 RCF, 10 min; (ii) 20,000 RCF, 10 min Centrifugation serum: (i) 1500 RCF, 10 min; (ii) 12,000 RCF, 2 min Storage: −80 °C	RT-qPCR	Target: miR-106a–363 cluster Method: RT-qPCR Normalization: plasma, cel-miR-39 + miR-16; serum, cel-miR-39 + miR-1228	↑ miR-106a-3p, miR-106a-5p, miR-92a-2-5p, miR-20b-5p in BC vs. HC (plasma) ↑ miR-106a-5p, miR-92a-3p, miR-20b-5p, miR-19b-3p in BC vs. HC (serum) Plasma miR-106a-3p + miR-106a-5p + miR-92a-2-5p + miR-20b-5p (0.85-0.90; 82-83%; 79-80%) for BC vs. HC Serum miR-106a-5p + miR-92a-3p + miR-20b-5p + miR-19b-3p (0.91-0.97; 87-94%; 87-94%) for BC vs. HC	[184]
72 BC, 56 BBL, 72 age-matched HC	**Clot activator gel-serum** Processing within 2 h Storage: −80 °C	Targets: miR-195, let-7a Method: RT-qPCR Normalization: snRNA U6		↓ miR-195 (0.75), let-7a (0.72) in BC vs. BBL and HC	[185]
Discovery: 20 TNBC Validation stage I: 40 TNBC Validation stage II: 110 TNBC 30 HC, 33 ER + BC	**Serum** Processing within 1 h	742 miRNAs microRNA Ready-to-Use PCR, Human panel I and panel II v2 Normalization: global mean	Targets: miR-652-3p, miR-223-3p, miR-107, miR-103a-3p, miR-101-3p, miR-32-5p, miR-30d-5p, miR-20a-5p, miR-18b-5p Method: RT-qPCR Normalization: snRNA U6	↑ miR-652 + miR-107 + miR-103 + miR-18b (0.81) in early recurrence vs. non-recurrence	[186]
Discovery: 5 TNBC, 5 non-TNBC, 5 age-matched HC Validation: 67 TNBC, 95 non-TNBC, 90 age-matched HC	**Plasma** Centrifugation: (i) 1600 *g*, 10 min, 4 °C; (ii) 1600 *g*, 10 min, 4 °C	752 miRNAs miRCURY LNA Array V3.R	Targets: miR-199a-5p, miR-21, miR-16 Method: RT-qPCR Normalization: miR-484	↓ miR-199a-5p (0.88), miR-21 (0.87), miR-16 (0.80) in TNBC vs. non-TNBC	[187]
12 HC, 74 TNBC, 44 non-TNBC	**Plasma** Storage: −80 °C	Targets: miR-105, miR-93-3p Method: RT-qPCR Normalization: snRNA U6		↑ miR-105 (0.93-85%), miR-93-3p (0.66-56%) in TNBC vs. non-TNBC	[188]
Discovery: 20 TNBC, 14 DPBC, 11 HC Validation: 24 TNBC, 24 DPBC, 28 HC	**EDTA-plasma** Centrifugation: 300 rpm, 5 min Storage: −80 °C	84 miRNAs Human Breast Cancer miScript miRNA PCR Array Normalization: cel-miR-39, SNORD68, SNORD95, SNORD96A, RUN6–2	Targets: miR-489, miR-200b, miR-193b, miR-125b, miR-10a Method: RT-qPCR Normalization: snRNA U6, snRNA U48, miR-16	↑ miR-489 (0.99), miR-200b (0.88), miR-193b (0.91), miR-125b (0.97), miR-10a in TNBC and DPBC vs. HC	[189]
57 luminal BC, 36 TNBC, 34 age-matched HC	**EDTA-plasma**, kept at 4 °C Processing within 2 h Centrifugation: (i) 1500 *g*, 4 °C, 15 min; (ii) 2500 *g*, 4 °C, 15 min Storage: −80 °C	18 miRNAs Method: RT-qPCR Normalization: cel-miR-39		↑ miR-210-3p, miR-195-5p, miR-29c-3p, miR-22-3p, miR-19a-3p, miR-19b-3p, miR-15a-5p, miR-7-5p in TNBC vs. HC ↑ let-7c-5p, miR-489-3p, miR-328-3p, ↓ miR-340-5p, miR-199a-5p in luminal BC vs. HC ↑let-7i-5p, miR-25-3p, miR-16-5p, let-7b-5p, ↓ miR-199a-3p in luminal BC and TNBC vs. HC miR-489-3p + miR-199a-3p + miR-195-5p + miR-15a-5p + miR-7-5p + let-7c-5p + let-7i-5p (0.93) for TNBC vs. HC miR-328-3p + miR-199a-3p + miR-195-5p + miR-25-3p + let-7i-5p (0.94) for luminal BC vs. HC	[190]
113 BC, 30 age-matched HC	**Serum**, left to clot 30 min at RT Storage: −80 °C	Targets: miR-21, miR-19a, miR-10b Method: RT-qPCR Normalization: miR-192		↑ miR-21 (0.71) in non-mBC HER2+ vs. non-mBC HER2- and HC ↑ miR-10b (0.75) in mBC HER2+ vs. mBC HER2- and HC ↑ miR-19a (0.75) metastatic IBC vs. metastatic non-IBC	[191]
386 HER2+ mBC treated with chemotherapy + trastuzumab (divided in training and 2 validation cohorts), 179 HER2+ mBC treated with chemotherapy only, 55 HC	**Serum** Processing according to Early Detection Research Network serum operating standard procedure	Microarray Normalization: quantile normalization and RMA algorithm	Targets: miR-4716-5p, miR-4310, miR-940, miR-720, miR-494, miR-451a, miR-451b, miR-30b-3p, miR-29a-5p, miR-22-3p, miR-17-3p, miR-16-5p, miR-10b-3p Method: RT-qPCR Normalization: cel-miR-39	↑ miR-451a, miR-17-3p, miR-16-5p, ↓ miR-940 in trastuzumab sensitive vs. non-sensitive ↓ miR-451a, miR-17-3p, miR-16-5p in trastuzumab resistant vs. HC ↑ miR-940 in trastuzumab sensitive vs. HC (further increased in trastuzumab resistant) miR-940+miR-451a+miR-17-3p+miR-16-5p panel (0.78-0.80%; 0.74-0.77%) for 1- and 2-years OS of patients under trastuzumab	[192]
68 BC luminal A subtypes treated with adjuvant chemotherapy	**Serum**, left to clot for 20 min Centrifugation: 3000 rpm, 5 min Storage: −80 °C	762 miRNAs TaqMan Real-time PCR microRNA Arrays A+B cards Normalization: miR-484	Targets: miR-375, miR-205, miR-19a, let-7b Method: RT-qPCR Normalization: miR-484	↑ miR-205, miR-19a in resistant BC vs. sensitive BC miR-19a+miR-205 (0.90-81%-75%) for resistant BC vs. sensitive BC	[193]
30 luminal A BC, 10 HC	**EDTA-plasma** Centrifugation: 1500 *g*, 20 min, 4 °C Storage: −80 °C	Targets: miR-155, miR-21, miR-10b, let-7a Method: RT-qPCR Normalization: SNORD		↑ miR-155, miR-21, miR-10b, ↓ let-7a in luminal A BC vs. HC miR-155, miR-21, miR-10b, ↑ let-7a in luminal A BC after the surgery, chemotherapy and radiotherapy vs. before treatments	[194]
56 BC, 10 age-matched HC	**Serum**	Targets: miR-155, miR-125b, miR-34a, miR-10b Method: RT-qPCR Normalization: miR-16		↑ miR-10b, miR-155 in BC vs. HC miR-155 advanced stage BC vs. early stage BC miR-125b advanced stage BC vs. early stage BC and HC miR-125b in non-responsive vs. responsive	[195]
118 BC NCT treated, 30 age-matched HC	**Serum**, left to clot 2-3 h at RT Centrifugation: 20 min, 1900 *g* Storage: −80 °C	Targets: miR-451, miR-373, miR-205, miR-155, miR-125b, miR-122, miR-21, miR-19a Method: RT-qPCR Normalization: miR-16		↑ miR-125b, ↓ miR-21 before NCT vs. after NTC in responders miR-125b (0.77-0.78), miR-21 (0.87-0.93), miR-125b+miR-21 (0.96) for responders vs. non-responders	[196]
27 BC treated with adjuvant chemotherapy	**Serum**, left to clot at least 1 h at RT Centrifugation: 2000 *g*, 10 min Storage: −80 °C	Affymetrix1 GeneChip1 2.0 miRNA array	Targets: miR-3200, miR-451, miR-205, miR-21 Method: RT-qPCR Normalization: miR-191	↓ miR-451, miR-3200 during chemotherapy treatment	[197]
109 BC underwent NCT	**Plasma** Processing within 1 h Centrifugation: (i) 1200 *g*, 10 min, 4 °C; (ii) 12,000 *g*, 10 min, 4 °C Storage: −80 °C	TaqMan miRNA microarray	Targets: miR-222, miR-451, miR-20a, miR-9 Method: RT-qPCR Normalization: cel-miR-39	↓ miR-34a in TNBC and HR+/HER2- insensitive vs. sensitive after NCT ↑ miR-222 in HR+/HER2- insensitive vs. sensitive before NCT ↑ miR-20a, ↓ miR-451 in HR+/HER2- insensitive vs. sensitive after NCT miR-34 (0.59) for TNBC and HR+/HER2- insensitive vs. Sensitive miR-20a (0.80), miR-451 (0.79) and miR-222 (0.71) for HR+/HER2- insensitive vs. sensitive	[198]
Training: 42 BC treated with chemotherapy Validation: 26 BC treated with chemotherapy	**Serum** Processing according to Institutional Review Board-approved protocols Storage: −80 °C	Solexa sequencing	42 miRNAs Method: RT-qPCR Normalization: miR-16	↑ miR-122 in relapse vs. non-relapse	[199]
63 early stage BC	**Clot activator gel-serum**, left to clot 30–60 min at RT Centrifugation: 3000 rpm, 10 minStorage: −80 °C	Targets: miR-155, miR-181b, miR-24, miR-19a Method: RT-qPCR Normalization: let-7a		↓ miR-181b, miR-155, miR-24 after tumor removal ↓ miR-19a after therapy ↑ miR-181b, miR-155, miR-24, miR-19a in high-risk BC vs. low-risk	[200]
133 BC, 21 age-matched HC	**Clot activator gel-serum** left to clot 30–60 min at RT Centrifugation 3000 rpm, 10 min Storage: −80 °C	Targets: miR-155, miR-181b, miR-24, miR-19a Method: RT-qPCR Normalization: let-7a		↑ miR-155, miR-24 in BC patients with recurrence vs. non-recurrence	[201]
187 early stage BC, 50 fibroadenoma, 20 HC	**EDTA-plasma** Processing within 1 h Centrifugation: 2000 *g*, 15 min, 4 °C Storage: −80 °C	Target: miR-106b Method: RT-qPCR Normalization: let-7a		↑ miR-106b in BC vs. fibroadenoma ↑ miR-106b (0.86-88%-60%) in BC recurrence vs. non-recurrence	[202]
Discovery: 40 BC without recurrence, 8 BC with recurrence, 31 HC Validation: 22 BC without recurrence and 20 BC with recurrence	**Red/gray tap gel-serum**, left to clot 30 min at RT Centrifugation: 4 °C, 2500 rpm, 10 min Storage: −80 °C	752 miRNAs miRCURY LNA Universal RT microRNA Ready-to-Use PCR Human panels I+II V3.M Normalization: global mean	20 miRNAs Method: RT-qPCR Normalization: miR-361-5p, miR-186-5p	↑ miR-375, miR-205-5p, miR-194-5p, miR-21-5p, ↓ miR-411-5p, miR-382-5p, miR-376c-3p in BC with recurrence vs. BC without recurrence miR-411-5p + miR-382-5p + miR-376c-3p + miR-375 + miR-205-5p + miR-194-5p + miR-21-5p (0.91-93%-77%) for BC with recurrence vs. BC without recurrence	[203]
209 early BC who underwent surgery followed by NCT and 23 HC	**EDTA-plasma** Processing within 2 h Centrifugation: (i) 2500 rpm, 15 min, 4 °C; (ii) 2000g, 15 min, 4 °C torage: −80 °C	Targets: miR-200b, miR-200c, miR-190, miR-23b, miR-21 Method: RT-qPCR Normalization: miR-23a		↑ miR-200c (0.68-76%-61%), miR-23b, miR-21 (0.69-71%-64%) and ↓ miR-190 in relapsed vs. non-relapsed miR-190 + miR-23b + miR-21 (0.77-80%-65%), miR-21 + miR-23b + miR-190 + axillary lymph node infiltration and tumor grade (0.87-89%-76%), miR-200c + axillary lymph node infiltration + tumor grade + ER status (0.89-75%-89%) for relapsed vs. non-relapsed	[204]
78 BC patients underwent NCT	**Serum** Processing within 2 h Centrifugation: 1600 *g*, 10 min Storage: −80 °C	2549 miRNA Agilent Human miRNA microarray 21.0 Normalization: quantile normalization procedure	Target: miR-4530 Method: RT-qPCR Normalization: cel-miR-39	↑ miR-4530 (0.66-98%-86%) in the sensitive group vs. resistant group	[205]
59 BC, 30 mBC, 29 HC	**Serum**	Targets: miR-155, miR-141, miR-34a, miR-10b Method: RT-qPCR Normalization: miR-16		↑ miR-10b, miR34a, ↓ miR155 in mBC vs. BC ↑ miR-155 in both mBC and BC vs. HC ↑ miR-10b, miR-34 in mBC vs. HC	[206]
120 BC without metastasis, 32 mBC, 40 HC	**Serum**	Targets: miR-373, miR-155, miR-93, miR-34a, miR-17, miR-10b Method: RT-qPCR Normalization: miR-16		↑ miR-373 (0.88), miR-93 (0.70), miR-34a (0.64) in BC with and without metastasis vs. HC ↑ miR-373 in mBC vs. HC ↓ miR-155, miR-17 in mBC vs. both HC and BC without metastasis miR-17 (0.68), miR-155 (0.78) for mBC vs. non metastatic BC	[207]
75 BC and 20 HC	**EDTA-whole blood and plasma, serum** Centrifugation: 2000 *g*, 10 min, RT (serum) (i) 150 *g*, 20 min, 4 °C; (ii) 1650 *g*, 20 min, RT (plasma) Storage: −80 °C	Targets: miR-452, miR-411, miR-215, miR-299-5p Method: RT-qPCR Normalization: miR-16		↓ miR-411, miR-299-5p in BC and mBC vs. HC	[208]
Discovery: 40 mBC Validation I: 354 mBC Validation II: 332 BC	**EDTA-plasma** Processing within 2 h Centrifugation: (i) 1300 *g*, 20 min, 10 °C; (ii) 15,500 *g*, 10 min, 10 °C Storage: −80 °C	677 miRNAs TaqMan^®^ Human microRNA cards v3.0	20 miRNAs Method: RT-qPCR Normalization: cel-miR-39, miR-29a, miR-139-5p	↑miR-200a, miR-200b, miR-200c, miR-210, miR-486-5p, ↓miR-215 in mBC (metastasis developed within 2 years) vs. non-mBC miR-486-5p + miR-215 + miR-210 + miR-200a + miR-200b + miR-200c panel (0.82-77%-75%) in mBC (metastasis developed within 2 years) vs. non-mBC	[209]
50 BC, 25 BBL, 50 age-matched HC	**Serum** Processing within 1-2 h Centrifugation: 3000 *g*, 20 min, 10 °C	Targets: miR-21, let-7 Method: RT-qPCR Normalization: miR-16		↑ miR-21, ↓ let-7 in BC vs. BBL and HC ↓ let-7 in mBC vs. non-mBC	[210]
115 BC, 115 age-matched HC	**Plasma** Centrifugation: (i) 1200 *g*, 10 min, 4 °C; (ii) 12,000 *g*, 10 min, 4 °C Storage: −80 °C	800 miRNAs Nanostring nCounter Human v3 miRNA Expression Assay Normalization: spike-ins		↑ miR-24-3p in mBC vs. non-mBC	[211]
35 lymph node mBC, 25 non-mBC, 10 HC	**Plasma**	Targets: miR-373, miR-10b Method: RT-qPCR Normalization: miR-16		↑ miR-373 (0.84-68%-89%), miR-10b (0.80-71%-72%) in mBC vs. non-mBC miR-10b + miR-373 (0.88-72%-94.3%) for mBC vs. non-mBC	[212]
111 BC, 46 age-matched HC	**Plasma**	1300 miRNAs Microarray Validation: RT-qPCR	Targets: miR-146a, miR-132, miR-130a, miR-107, miR-27a, miR-16 Method: RT-qPCR Normalization: miR-1207	↓ miR-146a, miR-130a, miR-16 in lymph node mBC vs. non-mBC	[213]
20 mBC (bone, liver and lung metastasis), 80 non-mBC, and 30 HC	**Serum** Storage: −80 °C	Targets: miR-205, miR-197, miR-155, miR-29b-2 Method: RT-qPCR Normalization: SNORD		↑ miR-205, miR-197, miR-155, miR-29b-2 in BC vs. HC ↑ miR-205 (0.66-61%-78%), miR-155 (0.67-50%-85%) in mBC vs. non-mBC	[214]
22 bmBC, 100 non-mBC, 59 age-matched HC	**Serum** Centrifugation: 1300 *g*, 10 min, RT Storage: −80 °C	Target: miR-10b Method: RT-qPCR Normalization: miR-16		↑ miR-10b in BC vs. HC ↑ miR-10b (0.77-65%-70%) in bmBC vs. non-bmBC	[215]
Screening: 30 mBC (11 CTC+, 9 CTC-), 10 HC Validation: 133 mBC (61 CTC+, 72 CTC-), 76 HC	**EDTA-plasma** Processing within 2 h Centrifugation: (i) 1300 *g*, 20 min, 10 °C; (ii)15,500 *g*, 10 min, 10 °C Storage: −80 °C	TaqMan Human MicroRNA array Card Set v2.0	Targets: miR-801, miR-768-3p, miR-630, miR-571, miR-375, miR-210, miR-206, miR-203, miR-200a, miR-200b, miR-200c, miR-193b*, miR-142-3p, miR-141, miR-139-3p, miR-133b, miR-99a Method: RT-qPCR Normalization: cel-miR-39	↑ miR-801, miR-375, miR-210, miR-203, miR-200a, miR-200b (0.88-80%-83%), miR-200c, miR-141 in CTC+ mBC vs. CTC- mBC and HC ↓ miR-768-3p in CTC+ mBC and CTC- mBC vs. HC miR-768-3p+mmiR-210+miR-200b+miR-200c+miR-141 (0.95-90%-91%) for CTC+ mBC vs. HC miR-768-3p+miR-210+miR-200c (0.78-80%-65%) CTC- mBC vs. HC	[216]
Discovery: 25 BC, 20 HC Validation: 76 BC, 52 HC	**Serum**, left to clot30–60 min at RT Centrifugation: 2000 rpm, 15 min, RT Storage: −80 °C	96 miRNAs Method: SdM-qRT-PCR Normalization: miR-103a+miR-132	Targets: miR-424, miR-199a, miR-29c Method: SdM-qRT-PCR Normalization: miR-103a+miR-132	↑ miR-424, miR-199a, miR-29c in BC vs. HC miR-424+miR-199a+miR-29 (0.90-0.91; 77-78%; 85-89%) for BC vs. HC	[217]

AA: African American; BAH: breast atypical hyperplasia; BBC: benign BC; BC: breast cancer; CA: Caucasian; ER: estrogen receptor; HC: healthy controls; HER2: human epidermal growth factor receptor 2; mBC: metastatic BC; NGS: next-generation sequencing; PR: progesterone receptor; RT: room temperature; RT-qPCR: real-time quantitative polymerase chain reaction; TNBC: triple-negative BC. §: inconsistent report: serum is reported as obtained from EDTA-anticoagulated tubes; ↑: increased levels of the considered miRNA; ↓: decreased levels of the considered miRNA.

**Table 3 jcm-08-01661-t003:** Circulating miRNAs associated with prostate cancer (PC).

Cohort	Pre-Analytical Variables	Analytical Method		Identified miRNAs AUC-Sensitivity%-Specificity%)	Study
		Screening	Validation		
25 mPC, 25 age-matched HC	**Clot activator gel-serum**	Targets: miR-296, miR-205, miR-143, miR-141, miR-125b, miR-100 Method: RT–qPCR Normalization: average of cel-miR-39, cel-miR-54, cel-miR-238		↑miR-141 (60%–100%) in PC vs. HC	[32]
Screening: 14 lPC + 7 mPC 1st validation: 42 lPC + 3 mPC 2st validation: 71 PC	**Serum**, left to clot 30 min at RT Centrifugation: 2000 *g*, 10 min Storage: −80 °C	667 miRNAs Low-density Taqman arrays	Targets: miR-375, miR-516a-3p, miR-200b, miR-141, miR-9* Method: RT–qPCR Normalization: cel-miR-39, cel-miR-54, cel-miR-238	↑ miR-375, miR-141 in mPC vs. lPC	[242]
37 lPC after radical prostatectomy, 18 BPH, 8 mPC, 20 HC	**Serum**, left to clot at least 60 min Processing within 2 h Centrifugation: 1800 *g*, 10 min Storage: −80 °C	Targets: miR-16, miR-32, miR-26a, miR-let7i, miR-195 Method: RT–qPCR Normalization: cel-miR-39		↑ miR-195, miR-26a (89%-56%), miR-let7i in PC vs. BPH ↓miR-195, miR-26a, miR-16 after vs. before radical prostatectomy miR-32+miR-26a+miR-let7i+miR-195 (78%-67%) for PC vs. BPH	[221]
26 lPC, 25 mPC, 20 HC	**Plasma**	Targets: miR-221, miR-141, miR-21 Method: RT–qPCR Normalization: miRNA/RNU1A ratio		↑ miR-221 (0.89), miR-21 (0.89) in PC s HC ↑ miR-221, miR-141 (0.76), miR-21 in mPC vs. lPC	[222]
Screening: 25 PC, 17 BPH Validation: 80 PC, 44 BPH, 54 HC	**EDTA-plasma** Centrifugation: 4 °C, 1500 rpm, 10 min Storage: −80 °C	1146 miRNAs Illumina’s miRNA expression platform v2	Targets: let-7c, let-7e, miR-1285, miR-940, miR-622, miR-346, miR-30c, miR-25 Method: RT–qPCR Normalization: snRNA U6	↓ let-7e, let-7c, miR-30c, ↑ miR-1285, miR-622 in PC vs. HC let-7e + let-7c + miR-30c + miR-622 + miR-1285 (0.92) for PC vs. BPH let-7e + let-7c + miR-30c + miR-622 + miR-1285 (0.86) for PC vs. HC	[223]
40 PC (16 CA, 24 AA), 32 age-matched HC (20 CA, 12 AA)	**Serum**	667 miRNAs TaqMan^®^ Array Human MicroRNA Cards A + B v2.0	Targets: miR-628-5p, miR-101, miR-25 Method: RT–qPCR Normalization: miR-223	↓ miR-628-5p (0.94), miR-101 (0.80), miR-25 (0.66) in PC vs. HC	[224]
72 PC, 34 HC	**Serum** Centrifugation: 3000 *g*, 10 min Storage: −80 °C	Targets: miR-1825, miR-484, miR-205, miR-141, let-7b Method: RT–qPCR		↓ miR-484 (0.79-88%-69%), miR-205 (0.91-78%-100%), let-7b (0.85-72%-88%), ↑ miR-1825 (0.96-93%-91%), miR-141 (0.93-88%-100%) in PC vs. HC	[225]
19 PC, 19 HC	**Serum** Storage: −80 °C	NGS Normalization: rimmed mean of M-values method based on log-fold and absolute gene-wise changes in expression levels between samples (TMM normalization)	Targets: miR-379-5p, miR-150-3p, miR-148a-3p, miR-134-3p, miR-127-3p Method: RT–qPCR Normalization: global mean	↑ miR-148a-3p in PC vs. HC	[226]
Screening: 21 PC, 25 HC Validation: 72 PC, 77 HC	**EDTA-plasma** Processing within 2 h Centrifugation: 3000 *g*, 15 min twice Storage: −80 °C	Targets: miR-375, miR-221, miR-145, miR-143, miR-141, miR-126, miR-125b, miR-93, miR-34a, miR-25, miR-21, let-7b Method: Customed TaqMan Array MicroRNA Cards Normalization: mean Ct of mi-17 + mean Ct of miR-191		↓ let-7b and ↑ miR-21, miR-125b, miR-126, miR-141, miR-143, miR-221, miR-375 in mPC vs. HC↓ miR-93 after intervention vs. baseline level at diagnosis	[245]
13 BPH, 11 lPC, 9 mPC (lymph node or distant), 11 CRPC	**Serum** Storage: −80 °C	732 miRNAs Exiqon’s microRNA Ready-to-Use PCR, Human panel I + II, V2.M Normalization: miR-320a		let-7a* + miR-616 + miR-562 + miR-210 (0.90-80%-100%) for PC miR-562 + miR-551b + miR-501-3p + miR-375 + miR-210 (0.92-84%-100%) for mPC vs. BPH miR-1203 + miR-708 + miR-375 + miR-200a (0.88-75%-100%) for mPC vs. lPC	[243]
Screening: 48 high risk PC, 48 low risk PC Validation: 25 high risk PC, 35 low risk PC	**Serum**	672 miRNAs Microfluidic-based multiplex qRT-PCR method Normalization: gobal mean		↑ miR-19b, miR-19a in PC vs. control miR-519c-5p + miR-345 + miR-19a/b (0.94) for high risk PC vs. low risk PC	[227]
75 PC, 27 BPH	**EDTA-whole blood**	Targets: miR-425, miR-375, miR-155, miR-145, miR-143, miR-141, miR-125b, miR-221, miR-34a, miR-21, miR-16, let7a Method: RT–qPCR Normalization: miR-16, miR-425		↑ miR-155 (0.62), miR-145 (0.63), miR-141 (0.66), ↓ let7a (0.69) in PC vs. BPH miR-155 + miR-145 + miR-141 + let-7a (0.78) for PC vs. BPH	[228]
59 PC, 16 BPH, 11 HC (young asymptomatic men)	**EDTA-plasma** Processing within 1 h Storage: −80 °C	RT–qPCR	Targets: miR-375, miR-141, miR-30c, let-7c Method: RT-qPCR Normalization: RNU6B	↓ miR-375 (0.81), miR-30, let-7c (0.76-75%-61%) in PC vs. BPH miR-375 + miR-141 + miR-30c + let-7c + PSA (0.78-64%-73%) for PC vs. BPH miR-375 + miR-141 + miR-30c + let-7c + PSA (0.88-87%-82%) for PC vs. HC	[229]
57 PC, 28 BPH	**EDTA-plasma** Centrifugation: 4 °C, 1000 *g*, 20 min Storage: −80 °C	Targets: miR-375, miR-21 Method: RT-qPCR Normalization: snRNA U6		↑ miR-375 (0.76-75%-75%), miR-21 (0.80-88%-75%) in PC vs. BPH miR-375 + miR-21 + PSA (0.88-88%-75%) for PC vs. BPH	[230]
36 PC, 31 BPH	**EDTA-plasma** Processing within 2h Centrifugation: (i) 580 *g*, 30 min, RT; (ii) 14,000 g, 4 °C, 15 min Storage: −80 °C	Targets: miR-2110, miR-1207-5p, miR-574-3p, miR-375, miR-223-3p, miR-141-3p, miR-130b-3p, miR-106a-5p, miR-93-5p, miR-24-3p, miR-21-5p, miR-20a-5p Method: RT-qPCR Normalization: miR-24		↑ miR-106, miR-21, miR-20a, ↓miR-223 in PC vs. BPH miR-106a/miR-130b (0.81), miR-106a/miR-223 (0.77), miR-106a/miR-130b + miR-106a/miR-223 (0.84) for PC vs. BPH	[231]
Discovery: 42 PC, 37 HC Validation: 40 PC, 37 HC	**EDTA-plasma** Processing within 1 h Storage: −80 °C	372 cancer-associated miRNAs miScript miRNA PCR array Normalization: global mean	Targets: miR-4289, miR-326, miR-152-3p, miR-98-5p Method: RT-qPCR Normalization: cel-miR-39	↑ miR-4289 (0.69–0.85), miR-326 (0.82–0.91), miR-152-3p (0.72–0.80), miR-98-5p (0.70–0.79) in PC vs. HC miR-4289 + miR-326 + miR-152-3p + miR-98-5p (0.82-0.95) in PC vs. HC	[232]
45 PC, 45 BPH, 50 HC	**EDTA-whole blood**	Target: miR-139-5p Method: RT-qPCR Normalization: snRNA U6		↑ miR-139-5p (0.94) for PC vs. BPH ↑ miR-139-5p (0.92) for PC vs. HC	[234]
149 PC, 179 non-PC (81 BPH, 56 HC, 40 other urinary diseases)	**Clot activator gel-serum** Processing within 1 h Centrifugation: 4 °C, 3000 *g*, 5 min Storage: −80 °C	Target: miR-410-5p Method: RT-qPCR Normalization: snRNA U6		↑ miR-410-5p (0.81) in PC vs. non-PC miR-410-5p (0.71) for high risk PC vs. low risk PC	[235]
24 PC, 24 BPH, 23 HC	**EDTA-whole blood** Storage: −80 °C	Target: miR-18a Method: RT-qPCR Normalization: snRNA U6		↑ miR-18a (0.81) in PC vs. BPH and HC	[236]
68 PC, 79 BPH	**Clot activator gel-serum** Centrifugation: 2000 *g*, 10 min Storage: −80 °C	Targets: let-7c, let-7e, let-7i, miR-940, miR-874-3p, miR-622, miR-497, miR-363-3p, miR-346, miR-195, miR-106a-5p, miR-30c-5p, miR-27b-3p miR-26a-5p, miR-26b-5p, miR-25-3p, miR-24-3p, miR-23b-3p, miR-20b-5p, miR-19b-2-5p, amiR-18b-5p Method: RT-qPCR Normalization: miR-191-5p, miR-425-5p		↓ let-7c, let-7e, let-7i, miR-26a-5p, miR-26b-5p, miR-25-3p (0.80-39%-88%), miR-18b-5p (0.87-24%-94%) in PC vs. BPH miR-18b-5p + miR-25-3p (0.92-44%-90%) for PC vs. BPH	[237]
102 PC, 50 HC	**EDTA-plasma** Processing within 2 h Centrifugation: (i) 700 *g*, 10 min; (ii) 2000 *g*, 10 min, 4 °C Storage: −80 °C	Targets: miR-375, miR-205-5p, miR-200c-3p, miR-183-5p, miR-143-3p, miR-133a-3p, miR-133b Method: RT-qPCR Normalization: RNU6B and RNU48		↑ miR-200c (0.62), ↓ miR-200b (0.57) in PC vs. HC	[238]
20 PC, 8 HC	**Serum**, left to clot 30 min at RT Centrifugation: 1500 *g*, 15 min, 4 °C Storage: −80 °C	Target: miR-375, miR-141-3p, miR-106b, miR-34a-5p, miR-21 ethod: RT-qPCR Normalization: snRNA U6		↑ miR-375 (0.91-100%-75%), miR-141-3p (0.83; 65-90%; 88-63%), miR-106b (0.75-95%-50%), miR-21 (0.86-90%-75%) in PC vs. HC miR-141-3p + miR-21 + miR-375 (0.86-93%-63%) for PC vs. HC	[239]
81 non-aggressive PC, 33 aggressive PC	**CPT-plasma** Storage: −80 °C	92 miRNAs miRNA Ready-to-Use PCR, CANCer Focus panel Normalization: the median polish method		↑ miR-17/↓miR-192 in aggressive PC (correlation) ↓ miR-181a, miR-150a, ↑ miR-22 in aggressive PC (independent correlation)	[240]
68 non-aggressive PC, 25 aggressive PC	**CPT-plasma** Storage: −80 °C	92 miRNAs miRNA Ready-to-Use PCR, CANCer Focus panel Normalization: the median polish method		↑miR-17 and ↓ miR-17 in aggressive PC (correlation)	[241]
26 mCRPC, 28 lPC low-risk, 30 lPC high-risk	**Serum**, left to clot 30 min at RT Processing within 15 h (storage 4 °C) Centrifugation: 1000 *g*, 4 °C, 10 min Storage: −80 °C	669 miRNAs TaqMan Low Density Array (TLDA) cards Normalization: snRNA U6	Targets: miR-409-3p, miR-378*, miR-375, miR-141 Method: RT-qPCR Normalization: snRNA U6	↑ miR-378*, miR-375, miR-141, ↓ miR-409-3p in mCRPC vs. low-risk group	[246]
Discovery: 8 early recurrent PC, 8 non-recurrent PC following radical prostatectomy Validation: 31 early recurrent PC, 39 non-recurrent PC following radical prostatectomy	**Gel-serum**, left to clot 30 min at RT Centrifugation: RT, 10 min, 2500 *g* Storage: −80 °C	TaqMan Low-Density Array (TLDA) Human MicroRNA A + B Cards Set v3.0 Normalization: (i) the interpolate normalization on U6 snRNA; (ii) all data scaled to cel-miR-39 to correct for extraction efficiency; (iii) mean Cq of miRNAs with a Cq< 30	Validation: miR-375, miR-194, miR-146b-3p, miR-141 Method: RT-qPCR Normalization: cel-miR-39	↑ miR-194 (0.65), miR-146b-3p (0.62) in recurrent PC vs. non-recurrent PC	[247]
25 lPC, 25 mCRPC	**EDTA-plasma** Processing within 4 h Storage: −80 °C	742 miRNAs Exiqon miRNA qPCR panel Normalization: miR-30e	Targets: miR-423-3p, miR-375, miR-205, miR-200c, miR-152, miR-151-3p, miR-141, miR-126, miR-21, miR-16 Method: RT-qPCR Normalization: miR-30e	↑ miR-423-3p, miR-375, miR-200c, miR-152, miR-151-3p, miR-141, miR-126, miR-21, ↓ miR-205, miR-16 in mCRPC vs. lPC miR-151-3p + miR-141 + miR-16 (0.94-84%-96%) for mCRPC vs. lPC	[248]
20 lPC, 20 mPC, 10 mCRPC, 6 BPH	**Serum**	Target: miR-21 Method: RT-qPCR Normalization: snRNA U6		↑ miR-21 in chemotherapy-resistant mCRPC vs. not resistant mCRPC and PC	[249]
97 CRPC	**K2-EDTA-plasma and clot-activator gel-serum** Processing within 30 min Centrifugation: 3000 *g*, 5 min, RT Storage: −80 °C	47 miRNAs Taqman array microRNA cards Normalization: global mean	Targets: miR-429, miR-301b, miR-222, miR-200a, miR-200b, miR-200c, miR-146a, miR-21, miR-20a, miR-20b Method: RT-qPCR Normalization: global mean	miR-200b (0.72) pre-docetaxel levels to predict death within 12 months	[250]
87 CRPC	**K2-EDTA-plasma** Processing within 30 min Centrifugation: 3000 *g*, 5 min, RT Storage: −80 °C	RT–qPCR	Targets: miR-590-5p, miR-429, miR-375, miR-301b, miR-222, miR-200a, miR-200b, miR-200c, miR-145a, miR-132, miR-25, miR-21, miR-20a, miR-20b Method: RT-qPCR Normalization: average of miR-152, miR-30c, miR-24	↑ miR-429, miR-375, miR-200a, miR-200b, miR-200c, miR-132 before chemotherapy correlate with reduced OS in CRPC	[251]

BPH: benign prostatic hyperplasia; CRPC: castration-resistant PC; HC: healthy controls; HSPC: hormone-sensitive PC; lPC: local PC; mCRPC: metastatic CRPC; mPC: metastatic PC; NGS: next-generation sequencing; PC: prostate cancer; PSA: prostate-specific antigen; RP: radical prostatectomy; RT: room temperature; RT-qPCR: real-time quantitative polymerase chain reaction. ↑: increased levels of the considered miRNA; ↓: decreased levels of the considered miRNA.

**Table 4 jcm-08-01661-t004:** Circulating miRNAs associated with osteosarcoma.

Cohort	Pre-Analytical Variables	Analytical Method		Identified miRNAs (AUC-Sensitivity%-Specificity%)	Study
		Screening	Validation		
65 osteosarcoma, 30 HC	**Serum** Centrifugation: 1200 *g*, 10 min, 4 °C Storage: −80 °C	Target: miR-21 Method: RT–PCR Normalization: snRNA U6		↑ miR-21 in osteosarcoma vs. HC	[254]
Discovery: 40 osteosarcoma, 40 HC Validation: 40 osteosarcoma, 40 HC	**Plasma** Storage: phased liquid nitrogen	Targets: miR-199a-3p, miR-143, miR-140, miR-132, miR-34, miR-21 Method: RT–PCR Normalization: cel-miR-39	Targets: miR-199a-3p, miR-143, miR-21 Method: RT–PCR Normalization: cel-miR-39	↑ miR-21 (0.86), ↓ miR-199a-3p (0.92), miR-143 (0.90) in osteosarcoma vs. HC miR-199a-3p + miR-143+ miR-21 (0.95-91%-94%) for osteosarcoma vs. HC	[255]
60 preoperative osteosarcoma, 60 matched HC. Paired serum samples before and 1 month after surgery from 28 osteosarcoma	**Serum** Processing within 1 h Centrifugation: 1200 *g*, 10 min, 4 °C Storage: −80 °C	739 miRNAs TaqMan Human MicroRNA Array A+B cards Normalization: snRNA U6	33 miRNAs Method: RT–PCR Normalization: data expressed as absolute values	↑ miR-199a in pre-operative osteosarcoma vs. HC ↓ miR-199a in post-operative osteosarcoma vs. pre-operative miR-199a-5p (0.86-88%-77%) for osteosarcoma vs. HC	[256]
1st 69 osteosarcoma 2nd 35 osteosarcoma before surgery and before and after chemotherapy	**Serum** Centrifugation: gradient centrifugation	Target: miR-21 Method: RT–PCR		↑ miR-21 in osteosarcoma vs. HC	[283]
100 osteosarcoma, 100 age-matched HC	**Serum** Storage: −80 °C	Targets: miR-206, miR-133b Method: RT–PCR Normalization: snRNA U6		↓ miR-206, miR-133b in osteosarcoma vs. HC	[284]
133 osteosarcoma, 133 age-/sex-matched HC	**Plasma** Storage: liquid nitrogen	Targets: miR-34b and miR-34c Method: RT–PCR Normalization: cel-miR-39		↓ miR-34b in osteosarcoma vs. HC	[257]
118 osteosarcoma, 60 HC	**Serum**, left to clot 60 min at RT Centrifugation: 1000 *g*, 10 min, 4 °C Storage: −80 °C	Target: miR-9 Method: RT–PCR Normalization: snRNA U6		↑ miR-9 in osteosarcoma vs. HC	[258]
80 osteosarcoma, 80 HC	**Serum**	Targets: miR-29a, miR-29b, miR-29c Method: RT–PCR Normalization: snRNA U6		↑ miR-29a, miR-29b, miR-29c in osteosarcoma vs. HC	[259]
100 osteosarcoma, 100 HC	**Serum**	Targets: miR-196a, miR-196b Method: RT–PCR Normalization: snRNA U6		↑ miR-196a, miR-196b in osteosarcoma vs. HC	[260]
89 osteosarcoma, 89 HC	**EDTA-plasma** Processing within 4 h Centrifugation: 1200 *g*, 15 min, 4 °C Storage: −80 °C	Target: miR-148a Method: RT–PCR Normalization: snRNA U6		↑ miR-148a (0.78-70%-83%) in osteosarcoma vs. HC	[261]
166 osteosarcoma, 60 HC	**Serum** Centrifugation: 1200 *g*, 15 min, 4 °C Storage: −80 °C	Target: miR-195 Method: RT–PCR Normalization: snRNA U6		↓ miR-195 (0.89-88%-83%) in osteosarcoma vs. HC	[262]
90 osteosarcoma, 50 osteosarcoma pre-surgery and 1 month post-surgery, 90 HC	**Plasma** Storage: phased liquid nitrogen	739 miRNAs TaqMan Human MicroRNA Array A + B Cards Normalization: snRNA U6	55 miRNAs Method: RT–PCR Normalization: cel-miR-39	↑ miR-374a-5p (0.92), miR-320a (0.92), miR-199a-3p (0.90), miR-195–5p (0.90) in osteosarcoma vs. HC miR-374a-5p+miR-320a+miR-199a-3p+miR-195–5p (0.96) for osteosarcoma vs. HC	[263]
40 osteosarcoma, 30 HC	**EDTA-plasma** Centrifugation: (i) 1300 *g*, 25 min, 4 °C; (ii) 1000 *g*, 5 min Storage: −80 °C	752 miRNAs Exiqon miRNome platform (human panels I+II) Normalization: miR-320a, miR-15a-5p, UniSp2		↓ miR-205-5p (0.70), ↑ miR-574-3p (0.88), miR-335-5p (0.78), miR-214 (0.80) in osteosarcoma vs. HC	[264]
166 osteosarcoma, 60 HC	**Serum** Centrifugation: 1200 *g*, 15 min, 4 °C Storage: −80 °C	Target: miR-27a Method: RT–PCR Normalization: snRNA U6		↑ miR-27a (0.87-70%-98%) in osteosarcoma vs. HC	[265]
100 osteosarcoma, 20 HC	**Serum** Centrifugation: 1200 *g*, 15 min, 4 °C Storage: −80 °C	Target: miR-191 Method: RT–PCR Normalization: snRNA U6		↑ miR-191 (0.86-74%-100%) in osteosarcoma vs. HC	[266]
80 osteosarcoma, 20 periostitis + 20 age-matched HC	**EDTA-serum**^§^ Storage: −80 °C	Target: miR-152 Method: RT–PCR Normalization: snRNA U6		↓ miR-152 (0.96-93%-96%) in osteosarcoma vs. HC	[267]
108 osteosarcoma, 50 age-/sex-matched HC	**Serum**	Target: miR-221 Method: RT–PCR Normalization: snRNA U6		↑ miR-221 (0.84-66%-100%) in osteosarcoma vs. HC	[268]
20 osteosarcoma, 20 age-/sex-matched HC	**Serum** Centrifugation: (i) RT, 60 min; (ii) 1000 *g*, 10 min, 4 °C. Storage: phased liquid nitrogen	168 miRNAs Exiqon Serum/Plasma Focus microRNA PCR Panel	14 miRNAs GeneCopoeia All-in-One™ miRNA RT-qPCR Normalization: cel-miR-39	↓ miR-451a (0.80), miR-425-5p (0.78), miR-139-5p (0.71), miR-106a-5p (0.73), miR-25-3p (0.80), miR-20a-5p (0.85), miR-16-5p (0.77) in osteosarcoma vs. HC	[269]
40 osteosarcoma, 40 age-/sex- matched HC	**Plasma** Centrifugation: (i) 2000 rpm, 30 min; (ii) 3000 rpm, 4 min; (iii) 5000 rpm, 3 min Storage: −80 °C	Target: miR-421 Method: RT–PCR Normalization: snRNA U6		↑ miR-421 in osteosarcoma vs. HC	[270]
46 osteosarcoma, 46 matched HC	**Serum** Centrifugation: 978 *g*, 10 min, RT Storage: liquid nitrogen	Target: miR-17 Method: RT–PCR Normalization: snRNA U6		↑ miR-17 in osteosarcoma vs. HC	[271]
76 osteosarcoma, 76 HC	**EDTA-plasma** Processing within 20 min Centrifugation: 4000 *g*, 15 min; supernatant further centrifuged at 12,000 *g*, 5 min Storage: −80 °C	Target: miR-542-3p Method: RT–PCR Normalization: snRNA U6		↑ miR-542-3p (0.84-78%-94%) in osteosarcoma vs. HC	[272]
114 osteosarcoma, 114 HC	**Serum**	Target: miR-300 Method: RT–PCR Normalization: snRNA U6		↑ miR-300 (0.89-84%-89%) in osteosarcoma vs. HC	[273]
57 osteosarcoma, age-/sex-/matched HC	**Serum**	Target: miR-222 Method: RT–PCR Normalization: snRNA U6		↑ miR-222 (0.81-67%-84%) in osteosarcoma vs. HC	[274]
112 osteosarcoma, 50 HC	**Serum** Storage: −80 °C	Target: miR-223 Method: RT–PCR Normalization: snRNA U6		↓ miR-223 (0.93-90%-97%) in osteosarcoma vs. HC	[275]
60 osteosarcoma, 30 HC	**Serum** Storage: −80 °C	Target: miR-326 Method: RT–PCR Normalization: RNU48		↓ miR-326 (0.90-84%-95%) in osteosarcoma vs. HC	[276]
185 osteosarcoma, 130 HC	**Serum** Storage: −80 °C	Target: miR-497 Method: RT–PCR Normalization: snRNA U6		↓ miR-497 (0.85) in osteosarcoma vs. HC	[277]
133 osteosarcoma, 133 age-/sex-matched HC	**Serum**, left to clot 1 h at 37 °C Centrifugation: 1500 *g*, 15 min, 4 °C Storage: −80 °C	Target: miR-95-3p Method: RT–PCR Normalization: snRNA U6		↓ miR-95-3p (0.86) in osteosarcoma vs. HC	[278]
72 osteosarcoma, 40 HC	**Serum** Centrifugation: 1100 *g*, 20 min, RT Storage: −80 °C	Target: miR-491-5p Method: RT–PCR Normalization: snRNA U6		↓miR-491-5p (0.83-72%-86%) in osteosarcoma vs. HC	[279]
95 osteosarcoma, 95 age-/sex-matched HC	**Serum** Centrifugation: 1500 *g*, 15 min, 4 °C Storage: −80 °C	Target: miR-375 Method: RT–PCR Normalization: snRNA U6		↓ miR-375 (0.89-82%-75%) in osteosarcoma vs. HC miR-375 (0.83-84%-84%) for good vs. poor drug response	[285]
152 osteosarcoma, 70 age-/sex-matched HC	**Serum** Storage: −80 °C	Target: miR-101 Method: RT–PCR Normalization: snRNA U6		↓ miR-101 (0.85-79%-83%) in osteosarcoma vs. HC	[280]
39 osteosarcoma, 19 age-/sex-matched HC	**Whole blood** Storage: −80 °C	Target: let-7a Method: RT–PCR Normalization: miR-191		↓ miR-let-7a (0.90) in osteosarcoma vs. HC	[281]
Discovery: 10 osteosarcoma, 10 age-matched non-osteosarcoma patients with other benign tumors, 10 HC Validation: 14 osteosarcoma, 14 age-matched non-osteosarcoma patients with other benign tumors, 8 HC	**Serum** Storage: −80 °C	Agilent miRNA microarray	Targets: miR-17-5p and miR-25-3p Method: RT–PCR Normalization: cel-miR-39	↑ miR-17-5p (0.72-64%-85%) and miR-25-3p (0.87-71%-92%) in osteosarcoma vs. non-osteosarcoma and HC	[286]
Screening: 32 osteosarcoma, 8 HC Validation: 29 osteosarcoma, 17 HC	**EDTA-plasma** Storage: −80 °C	752 miRNAs miRCURY LNA™ Universal RT miRNA PCR, Ready-to-Use Human Panel I V2 Normalization: miR-320a+miR-15a-5p	Targets: miR-221, miR-106a, and miR-21 Method: RT–PCR Normalization: the sum of miR-320a, miR-15a-5p, UniSp2	↑ miR-221 (0.83), miR-106a (0.96), miR-21 (0.85) in osteosarcoma vs. HC	[287]
114 osteosarcoma, 40 periostitis, 50 HC	**Serum** Processing within 30 min Centrifugation: 1500 *g*, 20 min Storage: −80 °C	Target: miR-124 Method: RT–PCR Normalization: miR-191		↓ miR-124 (0.85-80%-86%) in osteosarcoma vs. HC	[288]
15 osteosarcoma, 15 age-/sex-matched HC	**Clot activator gel-serum** Processing within 30 min Centrifugation: 5000 *g*, 10 min Storage: −80 °C	678 miRNAs TaqMan Array Human MicroRNA Panel v2.0 (A+B Cards) Normalization: global mean	Targets: miR-99b-5p, and miR-655-3p, miR-642a-5p, miR-376c-3p, miR-320a, miR-215-5p Method: RT–PCR Normalization: snRNA U6	↑ miR-642a-5p (0.84), miR-215-5p (0.87) in osteosarcoma vs. HC	[282]

HC: healthy controls; NGS: next-generation sequencing; RT: room temperature; RT-qPCR: real time quantitative polymerase chain reaction. §: inconsistent report: serum is reported as obtained from EDTA-anticoagulated tubes; ↑: increased levels of the considered miRNA; ↓: decreased levels of the considered miRNA.

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
