# Peer review of "Circulating miRNAs as Diagnostic and Prognostic Biomarkers in Common Solid Tumors: Focus on Lung, Breast, Prostate Cancers, and Osteosarcoma"

_jcm, 2019, doi:10.3390/jcm8101661_

Round 1

Reviewer 1 Report

In the manuscript by Michela Bottani and colleagues (ms# jcm-593833), the authors review the role of circulating miRNAs as potential diagnostic and prognostic biomarkers in human common cancers. The authors focus mostly on lung cancer, breast cancer, prostate cancer, and osteosarcoma.  The manuscript is well written and easy to understand. The article is very long, consists of 121 pages, including 4 large tables and a long list of references. The manuscript is a comprehensive collection and description of the current literature on the subject (summary of published studies and references), rather than the type of review that critically review the subject and proposes new concepts or solutions in the field. The authors well/honestly describe the advantages and limitations of circulating miRNA as biomarkers. The aim and the scope of the review are well/precisely defined  (including exclusions) in the Introduction section and consequently fulfilled in the subsequent parts of the manuscript. Although the review is bit hermetic and difficult to read as a whole from cover to cover, it may be a useful compendium of knowledge and source of information for researches focussed on specific cancer type, specific biomarker application, or specific miRNA. Although there are numerous review articles about circulating miRNAs as biomarkers in different types of cancer the form of the manuscript is quite unique and distinct from most other reviews what in my opinion justify its publication.

Please find below several comments/suggestions that may help to improve the manuscript. None of these comments is critical.

Title: I would suggest indicating in the title that review is focussed on lung cancer, breast cancer, prostate cancer, and osteosarcoma. E.g., “Circulating miRNAs as diagnostic and prognostic biomarkers in common solid tumors – focus on lung cancer, breast cancer, prostate cancer, and osteosarcoma”. Introduction, subsection 1.1. I would remove the word ‘just’ from the following sentence: ‘In addition, miRNAs expression has just been characterized in 61 tissues [6].’ It was over 3 years ago. Introduction, subsection 1.1., the last but one sentence. There are also other factors worth to be mentioned that may affect miRNA biogenesis/expression: (i) common SNPs (e.g., Saunders MA, Proc Natl Acad Sci USA 2007; Sun G, RNA 2009; Jazdzewski K, Proc Natl Acad Sci USA 2008), (ii) germline (e.g., Hughes AE, Am J Hum Genet 2011; Conte I, Proc Natl Acad Sci USA 2015; Mencia A, Nat Genet 2009) and (iii) somatic mutations (e.g., Oak N, Hum Mutat 2018; Galka-Marciniak P, Cancers 2019), and (iv) copy number variants (CNVs) (e.g., Calin GA, Proc Natl Acad Sci USA 2004; Czubak K, Oncotarget 2015). Please doublecheck the suggested references. Introduction, subsection 1.1., the last sentence. At least one more miRNA database, i.e., MiRGeneDB (Fromm B, Annu Rev Genet 2015) should be mentioned. Introduction, subsection 1.2. “... with RT-qPCR considered as the gold standard.” Is RT-qPCR really considered as “the gold standard” or is just most frequently used technique. Please doublecheck whether it should be gold or golden. All AUC, sensitivity and specificity values should be presented (if possible) in a uniform way across the manuscript. I would suggest the sensitivity and specificity values present as full integers, and all the AUC values to round-up to two digits after the decimal. As probably most readers will not read the article from cover to cover the better structuralization of the article may facilitate navigation in the text, e.g., adding subsections in chapters dedicated to particular cancers. It would be good if subsections would be consistent between subsequent parts dedicated to different cancer types. In case of breast cancer, are there differences in circulating miRNAs in patients with mutations in BRCA1 and BRCA2 or other breast cancer predisposition genes? If yes, such information would be interesting to add. It would be interesting on an example of one or two most extensively studied circulating miRNAs to see consistency between studies, between different methodologies. It would show reliability of results obtained in individual studies. Even if the consistency would not be perfect such comparison would better illustrate potential and limitation of circulating miRNA as biomarkers. The article is accompanied by 4 large (multi-page) tables. I would suggest publishing these tables as supplementary materials. It would substantially reduce the size of the article and those facilitate its printing and reading. The references present in Tables that are not cited in text may be listed in sentences referring to the particular Tables. Please note that some affiliations are differentially presented in the title page (english version) and in Acknowledgments (polish version).

Reviewer 2 Report

The authors provide with this review an important collection of miRNA literature and comprehensively describe the circulating miRNAs in lung -, breast-, and prostate cancer, and osteosarcoma.

In line 96 the ROC curves are mentioned, but the term AUC is never described in detail. Please include two sentences of explanation. For example: The area under the curve (AUC) represents the degree of separability. An AUC of 1 refers to a perfect separability whereas a value of 0.5 shows no capacity to discriminate the datasets.

Reviewer 3 Report

The ms by Bottani et al. "Current and future application of circulating miRNAs as biomarkers in the diagnosis and prognosis of common solid tumors" represents a huge amount of work done in summing up the current information about the use of circulating miRs as neoplastic biomarkers. Definitely the strengths of the article is the subject (with the highlighted perspective) and meticously gathered data. However, as for a review article, especially with the subject as above, one could expect a leading plot guiding through that enormous amount of information. At present it is just compilation of known facts.

Specifically, I would suggest to rethink the following points:

Abstract - for now it is a compilation of statements, should be reedited to present the vision of the paper. Introduction
- the authors state that biopsy is an expensive examination, miRNA estimation is not. I am afraid it's still not the case, rather the opposite. It would be a good argument, but some data should be given then.
- 1.1: against the subtitle, nothing is mentioned about miR biological functions (those could be presented in the table)
- 1.2: the contents of that paragraph suits rather towards the ending of the ms rather than beginning; one should avoid mentioning unpublished data unless they're groundbreaking
- 1.3: this section should be rather presented in the beginning of the introduction Next section is titled "2. miRNAs as biomarker for tumors", the subsections starting with "Circulating". Also in the paragraph the contents is mixed up when mentioning markers related to tumors themselves and to the circulating markers. Maybe the contents of the paragraph should be shortened / reformulated to the most promising biomarkers, while most of the information is doubled in tables. The authors stress the diagnostic / prognostic / predictive function of the biomarkers - maybe that would be a good key for selecting and organizing the species to be described in detail. The tables are extremely overloaded, lots of repetitions in columns (eg. instead of putting "xx NSCLC and xx HC" in every line, maybe o column description should contain the information "Cohort NSCLC/HC"), unnecessary information (e.g. storage conditions - can be find in referenced paper). What I find lacking is the analysis of isolation and detection methods for miRs in use now - if we are to use miRs in clinical practice, that's main issue. The English used is understandable, but the ms should be once carefully checked for grammatical, punctuation, and style errors (eg. using ";" and a lot of passive, line 24: highly invasiveness, ine 41: tissues biopsies, line 89: high incidence tumors, just to mention a few).

Round 2

Reviewer 3 Report

As mentioned previously, the review by Bottani et al. "Current and future application of circulating miRNAs as biomarkers in the diagnosis and prognosis of common solid tumors" presents a comprehensive up-to-date information / data / views on the subject.However one can still argue about details, the authors were able to correct obvious mistakes while standing still at the ms philosophy. The arguments presented are fair and acceptable.

Author Response

The authors thanks the reviewer for the encouraging comment. A professional Language editing was performed in the first round of revision (a certificate is available, if needed). Other typos emerged during this revision round have been corrected.